# Gromov-Wasserstein at Scale, Beyond Squared Norms

**Guillaume Houry** [1]  **Jean Feydy** [1]  **François-Xavier Vialard** [2]

## Abstract

A fundamental challenge in data science is to match disparate point sets with each other. While optimal transport efficiently minimizes point *displacements* under a bijectivity constraint, it is inherently sensitive to rotations. Conversely, minimizing *distortions* via the Gromov-Wasserstein (GW) framework addresses this limitation but introduces a non-convex, computationally demanding optimization problem. In this work, we identify a broad class of distortion penalties that reduce to a simple alignment problem within a lifted feature space. Leveraging this insight, we introduce an iterative GW solver with a linear memory footprint and quadratic (rather than cubic) time complexity. Our method is differentiable, comes with strong theoretical guarantees, and scales to hundreds of thousands of points in minutes. This efficiency unlocks a wide range of geometric applications and enables the exploration of the GW energy landscape, whose local minima encode the symmetries of the matching problem.

## 1. Introduction

**Point Set Registration.**   From 3D point clouds to imaging datasets, many objects are best described as collections of samples in a vector space. Establishing correspondences between these sets is an important task that facilitates critical downstream operations such as cross-domain information transfer, domain adaptation, and the formulation of geometric loss functions for shape registration and generative modeling (Courty et al., 2016; Genevay et al., 2018).

Taking as input a source and a target distribution, defined up to permutations of the samples, we look for a coupling that matches corresponding points and structures with each other.

[1]Inria, Université Paris Cité, Inserm, HeKA team, F-75015 Paris, France [2]LIGM, Université Gustave Eiffel, Marne-la-Vallée, France. Correspondence to: Guillaume Houry <guillaume.houry@inria.fr>.

*Proceedings of the 43rd International Conference on Machine Learning*, Seoul, South Korea. PMLR 306, 2026. Copyright 2026 by the author(s).

When both distributions are embedded in the same vector space, endowed with a domain-specific metric, a first idea is to match points from the source distribution with their closest neighbors in the target set. Accordingly, correspondences between 3D point clouds may be computed using the standard Euclidean metric on plain $xyz$ coordinates or custom features that leverage local shape contexts (Besl & McKay, 1992; Rusu et al., 2009).

**Optimal Transport (OT).**   To avoid degenerate couplings that match all source points with a single target, a common approach is to look for a *bijective* assignment. Under this constraint, minimizing the average distance between corresponding points leads to a linear optimization problem known as the Earth Mover, Monge-Kantorovitch, Linear Assignment or Optimal Transport (OT) problem in different communities. As detailed in (Peyré et al., 2019), measure theory generalizes this framework to clouds that contain different numbers of points, weighted samples or continuous distributions. Bijectivity constraints can then be enforced exactly, or relaxed to account for outliers and variations of the point sampling density (Séjourné et al., 2019).

While exact solvers for the OT problem struggle to scale beyond a few thousand points in difficult cases, iterative methods have been designed to handle large distributions at a small cost in the accuracy of the coupling. We may cite the auction algorithm which remains ideally suited to sparse problems in operations research (Bertsekas, 1988) or semi-discrete solvers which have become invaluable for physics simulations (Lévy et al., 2021; 2025). In machine learning research, solvers based on entropic regularization have emerged as the standard approach for computing OT couplings between large distributions sampled in vector spaces. These methods are based on variations of the Sinkhorn algorithm, which streams well on massively parallel hardware and provides smooth gradients (Sinkhorn, 1967; Kosowsky & Yuille, 1994; Cuturi, 2013). Modern implementations leverage annealing and multiscale heuristics (Gerber & Maggioni, 2017; Schmitzer, 2019; Feydy, 2020), scaling to millions of points to unlock a wide range of applications (Schiebinger et al., 2019; Shen et al., 2021; Qu et al., 2023).

**Gromov-Wasserstein (GW).**   In spite of these achievements, OT still suffers from two fundamental limitations:

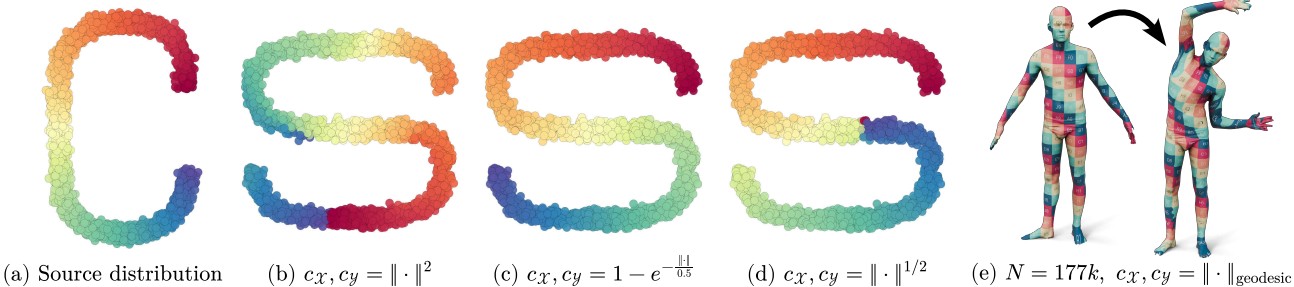

(a) Source distribution     (b) $c_{\mathcal{X}}, c_{\mathcal{Y}} = \|\cdot\|^2$     (c) $c_{\mathcal{X}}, c_{\mathcal{Y}} = 1 - e^{-\frac{\|\cdot\|}{0.5}}$     (d) $c_{\mathcal{X}}, c_{\mathcal{Y}} = \|\cdot\|^{1/2}$     (e) $N = 177k$, $c_{\mathcal{X}}, c_{\mathcal{Y}} = \|\cdot\|_{\text{geodesic}}$

*Figure 1.* (a) We optimize the GW objective of Eq. (2) to match a source distribution of points in the unit square ("C") with a target ("S"). Colors let us visualize the destination of every source point. (b) The preservation of squared distances between points has been studied extensively, but prioritizes the alignment of principal axes over smoothness. We propose a scalable GW solver for a broad class of penalties that may promote the preservation of topology (c) or find a balance between local and global structure (d). (e) This opens the door to applications on high-resolution data, such as those silhouettes sampled with 177k points each.

it requires a common embedding or at least a metric between the source and target samples, which is problematic for multimodal applications; and it is not robust to large non-uniform shifts such as global rotations.

To address these issues, an appealing strategy is to look for alignments that minimize the *distortion* induced by the coupling instead of the *displacement* of the samples. When framed in the language of measure theory, this perspective leads to the Gromov-Wasserstein (GW) optimization problem: a versatile framework capable of matching diverse data types, from graphs to continuous probability distributions (Mémoli, 2011). By focusing on intrinsic structure preservation instead of extrinsic feature engineering, GW has rapidly emerged as a convenient tool for graph machine learning (Chowdhury & Mémoli, 2019; Xu et al., 2019a; Brogat-Motte et al., 2022), image processing (Thual et al., 2022; Takeda et al., 2025; Beier & Beinert, 2025; Salmona et al., 2023; Zhang et al., 2021) and geometric data analysis (Chowdhury & Needham, 2021; Clark et al., 2025).

Despite this growing popularity, GW remains hard to solve. It is equivalent to the quadratic assignment problem, which implies that finding a global optimum is NP-hard in the general case while standard local minimization methods are prohibitively expensive in time and memory (Vayer et al., 2019a). As in the linear OT case, this context motivates the development of solvers based on the Sinkhorn iterations for an Entropy-regularized Gromov-Wasserstein (EGW) problem (Peyré et al., 2016; Solomon et al., 2016; Xu et al., 2019b). This technique is central to all state-of-the-art GW solvers (Kerdoncuff et al., 2021; Scetbon et al., 2022; Li et al., 2023) but comes at a cost: the regularization creates geometric biases that degrade both the quality of the optimal matching and the convergence speed of EGW solvers.

Meanwhile, the properties of an optimal GW coupling heavily depend on the objective used to penalize distortions. Recent works have largely focused on the preservation of

squared Euclidean distances between pairs of points, thanks to algebraic identities that simplify the optimization landscape (Vayer, 2020; Delon et al., 2022; Wang & Goldfeld, 2023; Zhang et al., 2024; Dumont et al., 2025); this was recently exploited by (Rioux et al., 2024) to derive a robust dual solver for EGW in this setting. Squared-norm EGW can also be seen as a concave-penalized OT problem, an algorithmic framework that has been thoroughly investigated in (Sebbouh et al., 2024). However, as illustrated in Fig. 1b, squared norm penalties are biased towards long-range interactions and are thus sensitive to outliers, which limits the applicability of this line of work to real-world problems.

**Contributions.** As a middle ground between general distortion penalties (that lead to complex optimization landscapes) and the preservation of squared distances between points (which is too brittle for most applications) we study the preservation of pairwise quantities which are *Conditionally of Negative Type* (CNT) (Maron & Lipman, 2018; Séjourné, 2022). As illustrated in Fig. 1, this class of GW problems is large enough to encode a broad range of behaviors, with attention paid to the preservation of long- or short-range structures depending on the target application.

We present three main contributions. First, we demonstrate that the optimal assignment in this setting decomposes into a linear map between feature spaces, followed by a standard OT coupling under the squared Euclidean distance. This structural insight reveals that the non-convex GW landscape is parameterized by a transformation matrix with dimensions determined solely by the preserved quantities. Second, we prove that the regularization biases typical of EGW solvers are either absent or easily mitigated within the CNT framework. Finally, we leverage these findings to develop a versatile, robust, and differentiable EGW solver that consistently outperforms state-of-the-art methods. Our code is publicly available at github.com/guillaumeHoury/egw-solvers.

## 2. Background and Notations

**Optimal Transport Plans.** We now introduce our main definitions and refer to (Peyré et al., 2019) for additional background or intuitions. Let the source $\alpha \in \mathcal{M}_1^+(\mathcal{X})$ and the target $\beta \in \mathcal{M}_1^+(\mathcal{Y})$ be probability measures with compact support over two topological spaces $\mathcal{X}$ and $\mathcal{Y}$. Let $\mathcal{M}(\alpha, \beta) \subset \mathcal{M}_1^+(\mathcal{X} \times \mathcal{Y})$ be the set of *couplings* (or *transport plans*) between $\alpha$ and $\beta$, i.e. the set of probability measures $\pi$ over $\mathcal{X} \times \mathcal{Y}$ that satisfy the marginal constraints:

$$\int \mathrm{d}\pi(\cdot, y) = \alpha \quad \text{and} \quad \int \mathrm{d}\pi(x, \cdot) = \beta .$$

Given a measurable cost function $c : \mathcal{X} \times \mathcal{Y} \longrightarrow \mathbb{R}$, the *optimal transport* (OT) problem between $\alpha$ and $\beta$ reads:

$$\mathrm{OT}(\alpha, \beta) := \min_{\pi \in \mathcal{M}(\alpha, \beta)} \int c(x, y) \, \mathrm{d}\pi(x, y) .$$

When the measures $\alpha$ and $\beta$ are supported on at most N points, this linear program can be solved in $\mathcal{O}(\mathrm{N}^3)$ time.

**The Sinkhorn Algorithm.** A common approach to accelerate optimal transport solvers is to add an entropic penalty to the objective. For any positive temperature $\varepsilon > 0$, the *entropic optimal transport* (EOT) problem reads:

$$\mathrm{OT}_\varepsilon(\alpha, \beta) := \min_{\pi \in \mathcal{M}(\alpha, \beta)} \int c \cdot \mathrm{d}\pi + \varepsilon \mathrm{KL} \left( \pi \| \alpha \otimes \beta \right)$$

$$\text{where} \quad \mathrm{KL} \left( \pi \| \alpha \otimes \beta \right) = \int \log \left( \frac{\mathrm{d}\pi}{\mathrm{d}\alpha \mathrm{d}\beta} \right) \mathrm{d}\pi .$$

The *Sinkhorn algorithm* reduces this problem to a maximization over the set of pairs of real-valued, continuous functions $(f, g) \in \mathcal{C}(\mathcal{X}) \times \mathcal{C}(\mathcal{Y})$. We provide details in Section A, and note that the optimal EOT coupling $\pi^*$ can then be reconstructed from the optimal pair $(f^*, g^*)$ with:

$$\frac{\mathrm{d}\pi^*}{\mathrm{d}\alpha \mathrm{d}\beta}(x, y) = \exp \left( \frac{f^*(x) + g^*(y) - c(x, y)}{\varepsilon} \right) . \quad (1)$$

**Gromov-Wasserstein (GW).** As discussed in the introduction, penalizing distortions instead of displacements is desirable in many applications. Given two *base costs* $c_\mathcal{X} : \mathcal{X} \times \mathcal{X} \longrightarrow \mathbb{R}$ and $c_\mathcal{Y} : \mathcal{Y} \times \mathcal{Y} \longrightarrow \mathbb{R}$, the *Gromov-Wasserstein* (GW) problem thus reads:

$$\mathrm{GW}(\alpha, \beta) := \min_{\pi \in \mathcal{M}(\alpha, \beta)} \mathcal{L}(\pi) , \quad \text{where} \quad (2)$$

$$\mathcal{L}(\pi) := \int \left( c_\mathcal{X}(x, x') - c_\mathcal{Y}(y, y') \right)^2 \mathrm{d}\pi(x, y) \mathrm{d}\pi(x', y') .$$

Under mild assumptions, GW quantifies how far distributions are from being isometric to each other:

**Proposition 2.1.** *Let $c_\mathcal{X}$ and $c_\mathcal{Y}$ be two symmetric functions with non-negative values such that:*

$$c_\mathcal{X}(x, x') = 0 \Leftrightarrow x = x' \quad \text{and} \quad c_\mathcal{X}(y, y') = 0 \Leftrightarrow y = y' .$$

*Then, $GW(\alpha, \beta) = 0$ if and only if $\alpha$ and $\beta$ are isometric, i.e. there exists an application $I : \mathcal{X} \longrightarrow \mathcal{Y}$ that pushes $\alpha$ onto $\beta$ such that for all $x$ and $x'$ in the support of $\alpha$:*

$$c_\mathcal{X}(x, x') = c_\mathcal{Y}(I(x), I(x')) .$$

**Entropic Regularization.** As in the OT case, the *Entropic Gromov-Wasserstein (EGW)* problem corresponds to:

$$\mathrm{GW}_\varepsilon(\alpha, \beta) := \min_{\pi \in \mathcal{M}(\alpha, \beta)} \mathcal{L}(\pi) + \varepsilon \mathrm{KL} \left( \pi \| \alpha \otimes \beta \right). \quad (3)$$

This regularization enables the use of scalable methods based on the Sinkhorn algorithm, but also introduces numerical biases and instabilities that we discuss in Section D.1.

**Embeddable Costs.** A non-negative, continuous cost $c : \mathcal{X} \times \mathcal{X} \longrightarrow \mathbb{R}^+$ is *conditionally of negative type* (CNT) if it is symmetric, satisfies $c(x, x) = 0$ for all $x \in \mathcal{X}$[1] and is such that for any finite collection of points $x_1, \ldots, x_n \in \mathcal{X}$ and coefficients $\lambda_1, \ldots, \lambda_n \in \mathbb{R}$:

$$\sum_{i=1}^n \lambda_i = 0 \implies \sum_{i=1}^n \lambda_i \lambda_j \cdot c(x_i, x_j) \leq 0 .$$

Moreover, we say that a cost $c$ is *definite* CNT if $c(x, x') = 0$ if and only if $x = x'$. We refer to (Bekka et al., 2007, Appendix C) for an introduction to this class of pairwise functions that includes tree distances, hyperbolic geodesic distances, spherical distances and all $p$-powered Euclidean distances $c(x, x') = \|x - x'\|^p$ for $0 < p \leq 2$.

CNT costs are relevant in machine learning because they are to squared Euclidean norms what reproducing kernels are to scalar products (Berlinet & Thomas-Agnan, 2011). This is made clear in the following characterizations:

**Theorem 2.2** (Schoenberg (1938)). *Let $c$ be a symmetric function on $\mathcal{X} \times \mathcal{X}$ satisfying $c(x, x) = 0$. The following statements are equivalent: (i) the cost $c$ is CNT, (ii) there exists a real Hilbert space $\mathcal{H}$ and a continuous mapping $\varphi : \mathcal{X} \longrightarrow \mathcal{H}$ such that for all $x$ and $x'$ in $\mathcal{X}$,*

$$c(x, x') = \|\varphi(x) - \varphi(x')\|_\mathcal{H}^2 , \quad \text{and} \quad (4)$$

*(iii) for every $\lambda > 0$, $e^{-\lambda c(\cdot, \cdot)}$ is a positive definite kernel.*

Note that the embedding $\varphi$ is not uniquely defined, but is necessarily injective if $c$ is a definite CNT cost.

**Proposition 2.3** (Schoenberg (1938)). *Let $x_0 \in \mathcal{X}$. A cost $c$ is CNT if and only if it induces a positive definite kernel:*

$$k(x, x') = \left( c(x, x_0) + c(x_0, x') - c(x, x') \right) / 2 .$$

---

[1]This condition can be relaxed by considering the cost $\tilde{c}(x, y) = c(x, y) - \frac{1}{2}(c(x, x) + c(y, y))$.

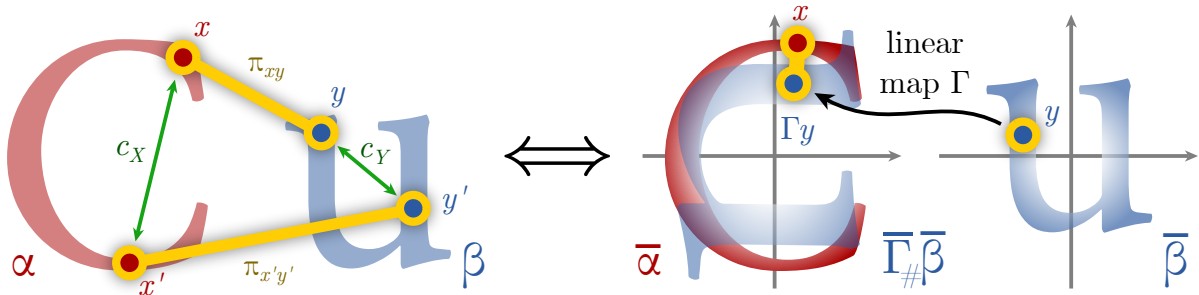

*Figure 2.* (left) In order to match a source distribution $\alpha$ (red "C") with a target distribution $\beta$ (blue "u"), the GW objective of Eq. (2) penalizes distortions between corresponding pairs of points. (right) Theorem 3.4 shows that up to known embeddings in Hilbert spaces and the optimization of a linear alignment $\Gamma$, we can reduce this problem to the computation of an OT plan for the squared norm cost.

**Hilbert-Schmidt Operators.** A linear map between Hilbert spaces $\Gamma : \mathcal{H}_{\mathcal{Y}} \longrightarrow \mathcal{H}_{\mathcal{X}}$ is a *Hilbert-Schmidt (HS) operator* if it is the limit of a sum of rank-1 operators:

$$uv^{\perp} : w \longmapsto \langle v, w \rangle \, u \quad \text{for } u \in \mathcal{H}_{\mathcal{X}} \text{ and } v \in \mathcal{H}_{\mathcal{Y}}.$$

These operators form a Hilbert space $\mathbf{H} = \mathrm{HS}(\mathcal{H}_{\mathcal{Y}}, \mathcal{H}_{\mathcal{X}})$ that generalizes the notion of matrices to infinite dimensions: in particular, when $\mathcal{H}_{\mathcal{X}}$ and $\mathcal{H}_{\mathcal{Y}}$ are Euclidean spaces, the HS norm $\|\cdot\|_{\mathrm{HS}}$ is identical to the Frobenius norm.

## 3. Gromov-Wasserstein with CNT Costs

**GW features.** We introduce a non-linear embedding that best encodes the geometry of the GW problem.

**Definition 3.1** (GW-embeddings). Let $\alpha$ be a probability measure on $\mathcal{X}$, $c_{\mathcal{X}} : \mathcal{X} \times \mathcal{X} \longrightarrow \mathbb{R}^{+}$ be a definite CNT cost and $\varphi : \mathcal{X} \longrightarrow \mathcal{H}_{\mathcal{X}}$ be an injective embedding provided by Theorem 2.2 to represent $c_{\mathcal{X}}$. Since Eq. (4) is invariant by translations, we can assume that $\int \varphi(x) \, d\alpha(x) = 0_{\mathcal{H}_{\mathcal{X}}}$. Then, we say that the injective map:

$$\Phi : x \in \mathcal{X} \mapsto \left( \varphi(x), \tfrac{1}{2}\|\varphi(x)\|^2 \right) \in \mathcal{H}_{\mathcal{X}} \times \mathbb{R}$$

is a GW-embedding of the distribution $\alpha$ with respect to $c_{\mathcal{X}}$. Similarly, we extend any Hilbert-Schmidt operator $\Gamma \in \mathbf{H}$ to our product feature space with:

$$\overline{\Gamma} : (y, s) \in \mathcal{H}_{\mathcal{Y}} \times \mathbb{R} \mapsto (\Gamma y, s) \in \mathcal{H}_{\mathcal{X}} \times \mathbb{R} \, .$$

**Example.** When $c_{\mathcal{X}}$ and $c_{\mathcal{Y}}$ are squared Euclidean norms on the finite-dimensional spaces $\mathcal{X} = \mathbb{R}^{\mathrm{D}}$ and $\mathcal{Y} = \mathbb{R}^{\mathrm{E}}$, the CNT embedding spaces coincide with the base spaces as $\mathcal{H}_{\mathcal{X}} = \mathbb{R}^{\mathrm{D}}$ and $\mathcal{H}_{\mathcal{Y}} = \mathbb{R}^{\mathrm{E}}$. The GW-embeddings of two distributions $\alpha \in \mathcal{M}_1^+(\mathcal{X})$ and $\beta \in \mathcal{M}_1^+(\mathcal{Y})$ read:

$$\Phi : x \in \mathbb{R}^{\mathrm{D}} \mapsto \left( x - x_\alpha, \tfrac{1}{2}\|x - x_\alpha\|^2 \right) \in \mathbb{R}^{\mathrm{D}+1} \quad \text{and}$$
$$\Psi : y \in \mathbb{R}^{\mathrm{E}} \mapsto \left( y - y_\beta, \tfrac{1}{2}\|y - y_\beta\|^2 \right) \in \mathbb{R}^{\mathrm{E}+1} \, ,$$

where $x_\alpha = \int z \, d\alpha(z)$ and $y_\beta = \int z \, d\beta(z)$ denote the centers of mass of both distributions. We can identify any

linear map $\Gamma : \mathcal{H}_{\mathcal{Y}} \longrightarrow \mathcal{H}_{\mathcal{X}}$ with a D × E matrix and write:

$$\overline{\Gamma} = \left( \begin{array}{c|c} \Gamma & 0 \\ \hline 0 & 1 \end{array} \right) \text{ as a } (\mathrm{D}+1) \times (\mathrm{E}+1) \text{ matrix.}$$

**Dual Formulation.** We can now state our main theorem:

**Theorem 3.2.** *Let $\alpha \in \mathcal{M}_1^+(\mathcal{X})$ and $\beta \in \mathcal{M}_1^+(\mathcal{Y})$ be two probability distributions with compact supports on topological spaces $\mathcal{X}$ and $\mathcal{Y}$, endowed with definite CNT costs $c_{\mathcal{X}}$ and $c_{\mathcal{Y}}$. Let $\Phi(x) = (\varphi(x), \tfrac{1}{2}\|\varphi(x)\|^2)$ and $\Psi(y) = (\psi(y), \tfrac{1}{2}\|\psi(y)\|^2)$ denote their respective GW-embeddings, as in Definition 3.1. Then, for any temperature $\varepsilon \geq 0$, the EGW problem of Eq. (3) is equivalent to:*

$$GW_\varepsilon(\alpha, \beta) = C(\alpha, \beta) + 8 \min_{\Gamma \in \mathbf{H}} \min_{\pi \in \mathcal{M}(\alpha,\beta)} \mathcal{F}(\Gamma, \pi) \, ,$$

*where $C(\alpha, \beta)$ is an additive constant and:*

$$\mathcal{F}(\Gamma, \pi) := \|\Gamma\|_{\mathrm{HS}}^2 + (\varepsilon/8) \cdot KL\left( \pi \| \alpha \otimes \beta \right)$$
$$- 2 \int \left\langle \overline{\Gamma}, \Phi(x)\Psi(y)^\top \right\rangle_{\mathrm{HS}} d\pi(x, y) \, .$$

Our proof builds upon similar dual formulations proposed recently for squared Euclidean costs in finite dimensions (Rioux et al., 2024; Zhang et al., 2024; Sebbouh et al., 2024; Pal et al., 2025). The objective function $\mathcal{F}(\Gamma, \pi)$ is not *jointly* convex, but has the following properties:

**Theorem 3.3.** *$\mathcal{F}$ is convex with respect to $\Gamma$ and:*

$$\Gamma^\star(\pi) := \underset{\Gamma \in \mathbf{H}}{\operatorname{argmin}} \, \mathcal{F}(\Gamma, \pi) = \int \varphi(x)\psi(y)^\top \, d\pi(x, y) \, .$$

**Theorem 3.4.** *Let $\overline{\alpha} = \Phi_\# \alpha$ and $\overline{\beta} = \Psi_\# \beta$ denote the pushforward images of the distributions $\alpha$ and $\beta$ by their respective GW-embeddings. Since $\Phi$ and $\Psi$ are injective, we can identify the supports of $\alpha$ and $\beta$ with those of $\overline{\alpha}$ and $\overline{\beta}$ and remark that $\mathcal{F}$ is convex with respect to $\pi$, with:*

$$\pi^\star(\Gamma) := \underset{\pi \in \mathcal{M}(\alpha,\beta)}{\operatorname{argmin}} \, \mathcal{F}(\Gamma, \pi) = \arg_{\pi \in \mathcal{M}(\alpha,\beta)} \mathrm{OT}_{\varepsilon/8}(\overline{\alpha}, \overline{\beta})$$

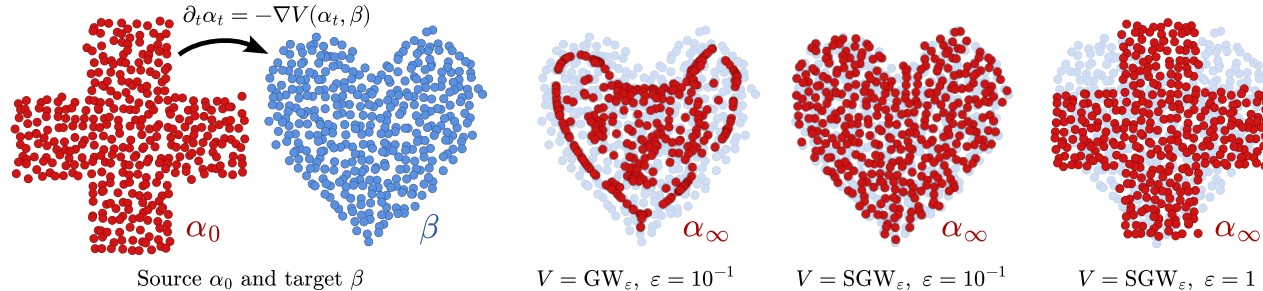

*Figure 3.* Wasserstein gradient flows $\alpha_{t+\delta t} = \alpha_t - \delta t \nabla V(\alpha_t, \beta)$ transporting a source measure $\alpha_0$ (red cross) towards a target $\beta$ (blue heart). Entropic bias causes the limit measure to collapse in small clusters with $\mathrm{GW}_\varepsilon$. The debiased Sinkhorn GW divergence $\mathrm{SGW}_\varepsilon$ fixes this issue, as the flow converges to the target. At large temperature $\varepsilon$, convergence fails and the flow remains cross-shaped.

*for the* bilinear *cost* $c_\Gamma(x, y) := -2\langle x, \overline{\Gamma}y \rangle_{\mathcal{H}_\mathcal{X} \times \mathbb{R}}$.

*Equivalently, if we identify the supports of $\overline{\beta}$ and $\overline{\alpha}$ with their pushforward images under the actions of the linear map $\overline{\Gamma}$ and of its transpose $\overline{\Gamma}^\top$, we can write that:*

$$\pi^\star(\Gamma) = \arg_{\pi \in \mathcal{M}(\alpha,\beta)} \mathrm{OT}_{\varepsilon/8}(\overline{\alpha}, \overline{\Gamma}_\# \overline{\beta})$$
$$= \arg_{\pi \in \mathcal{M}(\alpha,\beta)} \mathrm{OT}_{\varepsilon/8}(\overline{\Gamma}^\top_\# \overline{\alpha}, \overline{\beta})$$

*for the* squared norms costs *in $\mathcal{H}_\mathcal{X} \times \mathbb{R}$ and $\mathcal{H}_\mathcal{Y} \times \mathbb{R}$.*

As illustrated in Fig. 2, this remarkable result allows us to offload most of the complexity of the GW problem to the well-understood theory of entropy-regularized OT with squared Euclidean costs. In the remainder of the paper, we show that starting from an arbitrary linear map $\Gamma_0$ in **H** and iteratively applying the alternate minimization updates:

$$\pi_{t+1} \leftarrow \pi^\star(\Gamma_t), \quad \Gamma_{t+1} \leftarrow \Gamma^\star(\pi_{t+1}) \tag{5}$$

is both easy to implement at scale and enjoys strong theoretical guarantees.

## 4. Theoretical Guarantees of CNT-EGW

We now leverage Theorem 3.4 to prove two key results for EGW with CNT costs that have important practical implications: EGW can be debiased, enabling its use for variational methods; and EGW solvers are provably convergent, making them reliable in all contexts.

### 4.1. Entropic Bias and Symmetric Debiasing

**Entropic Bias.** Unlike GW, the EGW problem does not satisfy the geometric properties of Proposition 2.1: if $\varepsilon > 0$, we no longer have $\mathrm{GW}_\varepsilon(\alpha, \alpha) = 0$ and $\alpha$ does not necessarily minimize $\mathrm{GW}_\varepsilon(\alpha, \beta)$ over $\beta \in \mathcal{M}_1^+(X)$. This *entropic bias* is well known in the EOT setting and is often addressed by introducing the *Sinkhorn divergence*:

$$\mathrm{S}_\varepsilon(\alpha, \beta) = \mathrm{OT}_\varepsilon(\alpha, \beta) - \tfrac{1}{2}(\mathrm{OT}_\varepsilon(\alpha, \alpha) + \mathrm{OT}_\varepsilon(\beta, \beta)).$$

We propose a new formulation of the result of (Feydy et al., 2019) that highlights the importance of the class of CNT costs for entropy-regularized OT:

**Theorem 4.1.** *Let $c$ be a continuous cost (not necessarily vanishing on the diagonal) on a bounded domain. Then, $c$ is CNT if and only if $\mathrm{S}_\varepsilon \geq 0$ for any $\varepsilon > 0$.*

By analogy, we define the *Sinkhorn GW* divergence as:

$$\mathrm{SGW}_\varepsilon(\alpha, \beta) = \mathrm{GW}_\varepsilon(\alpha, \beta) - \tfrac{1}{2}(\mathrm{GW}_\varepsilon(\alpha, \alpha) + \mathrm{GW}_\varepsilon(\beta, \beta)).$$

The use of SGW was previously proposed in (Séjourné, 2022; Rioux et al., 2024); however, whether this approach effectively corrects the entropic bias remained an open question. In the CNT case, we provide a positive answer:

**Theorem 4.2.** *If $c_\mathcal{X}$ and $c_\mathcal{Y}$ are CNT costs, then $\mathrm{SGW}_\varepsilon(\alpha, \beta) \geq 0$ for any distributions $\alpha$, $\beta$ and temperature $\varepsilon > 0$. If $\alpha$ and $\beta$ are isometric, then $\mathrm{SGW}_\varepsilon(\alpha, \beta) = 0$.*

Yet, the nullity of $\mathrm{SGW}_\varepsilon(\alpha, \beta)$ does not imply that $\alpha$ and $\beta$ are isometric. When the regularization parameter $\varepsilon$ is too large, $\mathrm{SGW}_\varepsilon$ may fail to separate non-isometric measures: we discuss examples in Proposition C.6. In finite dimension, we provide a criterion that controls this effect based on the eigenvalues of the covariance matrices $\Sigma_\alpha = \int xx^\top \mathrm{d}\alpha(x)$ and $\Sigma_\beta = \int yy^\top \mathrm{d}\beta(y)$ of centered distributions.

**Proposition 4.3.** *Let $\varepsilon > 0$, $\alpha \in \mathcal{M}_1^+(\mathbb{R}^\mathrm{D})$, $\beta \in \mathcal{M}_1^+(\mathbb{R}^\mathrm{E})$ with supports of diameter $R$, $R'$, and $c_\mathcal{X}, c_\mathcal{Y}$ the squared norm of $\mathbb{R}^\mathrm{D}$ and $\mathbb{R}^\mathrm{E}$. Let $\lambda_\alpha$, $\lambda_\beta$ be the smallest eigenvalues of $\Sigma_\alpha$ and $\Sigma_\beta$. There are constants $C(\mathrm{D}, R)$ and $C(\mathrm{E}, R')$ such that, if:*

$$\varepsilon \max(1, \log(1/\varepsilon)) \leq \max(C(\mathrm{D}, R) \cdot \lambda_\alpha^2, \ C(\mathrm{E}, R') \cdot \lambda_\beta^2),$$

*then $\alpha$ and $\beta$ are isometric if and only if $\mathrm{SGW}_\varepsilon(\alpha, \beta) = 0$.*

In words: the measures must be spread across all ambient dimensions for EGW to accurately capture their differences. This result cannot be generalized to infinite dimensions, since the covariance operators $\Sigma_\alpha$ and $\Sigma_\beta$ are compact,

with 0 as the infimum of their singular values. As illustrated in Figure 3, keeping $\varepsilon$ as small as possible is thus essential to let $\mathrm{SGW}_\varepsilon$ separate any two non-isometric distributions.

## 4.2. Convergence of EGW Solvers

Most existing solvers for the EGW problem of Eq. (3) rely on descent schemes in the space of couplings $\mathcal{M}(\alpha, \beta)$. Illustratively, ENTROPIC-GW – the influential algorithm of (Peyré et al., 2016) – relies on the updates:

$$\pi_{t+1} = \operatorname*{argmin}_{\pi \in \mathcal{M}(\alpha, \beta)} \left( \int \nabla_{\pi_t} \mathcal{L} \cdot \mathrm{d}\pi + \varepsilon \mathrm{KL}\left( \pi \| \alpha \otimes \beta \right) \right). \quad (6)$$

The authors proved the equivalence of Eq. (6) with a projected gradient descent (PGD) for the KL geometry with a step size $\tau = 1/\varepsilon$. Although PGD is guaranteed to converge when $\tau$ is sufficiently small, this may not be compatible with the small values of $\varepsilon$ that are required to approximate the true GW problem. In practice, ENTROPIC-GW may thus oscillate between distinct transport plans without reaching convergence: we show a typical failure case in Figure 7.

Fortunately, this cannot happen with CNT costs since ENTROPIC-GW is then equivalent to our scheme:

**Proposition 4.4** (Primal descent). *Couplings $(\pi_t)$ obtained by alternate minimization from Eq. (5) also satisfy Eq. (6), with a monotonic decrease of the EGW loss at every step.*

Similarly, we draw an equivalence between alternate minimization and gradient descent on the dual space **H**:

**Proposition 4.5** (Dual descent). *Operators $\Gamma_t \in \mathbf{H}$ obtained by alternate minimization from Eq. (5) also satisfy:*

$$\Gamma_{t+1} = \Gamma_t - (1/2) \cdot \nabla \mathrm{U}_\varepsilon(\Gamma_t)$$
$$\text{where} \quad \mathrm{U}_\varepsilon : \Gamma \mapsto \min_{\pi \in \mathcal{M}(\alpha, \beta)} \mathcal{F}(\Gamma, \pi) \quad (7)$$

*derives from the objective $\mathcal{F}$ defined in Theorem 3.2.*

(Rioux et al., 2024) studied a similar dual gradient descent scheme in finite dimensions, with step sizes that must stay small ($\tau = \mathcal{O}(\varepsilon)$) to account for the reduced smoothness of $\mathrm{U}_\varepsilon$ at low temperatures $\varepsilon$. Beyond generalizing this framework to CNT costs, Proposition 4.5 makes the gradient step size independent of $\varepsilon$. Our alternate minimization point of view allows us to adapt the majorization-minimization (MM) framework to our problem (Lange, 2016), providing quantitative convergence results even with a fixed $\tau = 1/2$:

**Theorem 4.6.** *Let $(\pi_t, \Gamma_t)$ be the alternate sequence of Eq. (5). Then, $\|\Gamma_t - \Gamma_{t+1}\|_{\mathrm{HS}} \to 0$ and every subsequential limit of $(\Gamma_t)$ is a critical point of $\mathrm{U}_\varepsilon$, with:*

$$\sum_{k=0}^{t-1} \|\Gamma_k - \Gamma_{k+1}\|_{\mathrm{HS}}^2 \leq \mathrm{U}_\varepsilon(\Gamma_0) - \mathrm{U}_\varepsilon(\Gamma_t). \quad (8)$$

**Optimization in the Space of Linear Operators.** Although we established the equivalence of alternate minimization, primal descent and dual descent, these optimal schemes offer distinct perspectives. By parameterizing the problem via the operator $\Gamma$, whose dimensions reflect the geometries of the costs, we effectively recast the combinatorial matching problem as a linear registration task in a high-dimensional feature space.

This reparameterization has profound practical implications. Because estimating a global linear transform is inherently more robust than resolving a precise point-wise matching, the solver becomes tolerant to approximation errors. Consequently, we can employ cheap, coarse estimates of the Optimal Transport solutions needed in Theorem 3.4 without compromising the final convergence. We discuss this point further in Section D. Moreover, since the dual variable $\Gamma$ encodes the global geometric alignment of the measures $\alpha$ and $\beta$ without being tied to their supports, it provides a robust framework to visualize the optimization landscape (as illustrated in Section 6) and naturally supports the transfer of correspondences across different sampling resolutions.

## 5. Implementation

**Notations.** We now consider two discrete measures:

$$\alpha = \sum_{i=1}^{\mathrm{N}} a_i \delta_{x_i} \quad \text{and} \quad \beta = \sum_{j=1}^{\mathrm{M}} b_j \delta_{y_j},$$

where the samples $x_1, \ldots, x_{\mathrm{N}} \in \mathcal{X}$ and $y_1, \ldots, y_{\mathrm{M}} \in \mathcal{Y}$ belong to the base spaces $\mathcal{X}$ and $\mathcal{Y}$ while the two collections of probability weights $a_1, \ldots, a_{\mathrm{N}} \geqslant 0$ and $b_1, \ldots, b_{\mathrm{M}} \geqslant 0$ sum up to 1. We identify admissible couplings $\pi \in \mathcal{M}(\alpha, \beta)$ with matrices $(\pi_{ij}) \in \mathbb{R}^{\mathrm{N} \times \mathrm{M}}$ that satisfy the marginal constraints $\sum_j \pi_{ij} = a_i$ and $\sum_i \pi_{ij} = b_j$ for all $i$ and $j$.

**CNT-GW Solver.** Our method relies on the embeddings introduced in Section 3. Assuming that we have access to *centered* collections of embedding vectors $X_1, \ldots, X_{\mathrm{N}} \in \mathbb{R}^{\mathrm{D}}$ and $Y_1, \ldots, Y_{\mathrm{M}} \in \mathbb{R}^{\mathrm{E}}$ such that for all $i$ and $j$:

$$\sum_{i=1}^{\mathrm{N}} a_i X_i = 0, \quad c_{\mathcal{X}}(x_i, x_j) = \|X_i - X_j\|^2, \quad (9)$$
$$\sum_{j=1}^{\mathrm{M}} b_j Y_j = 0, \quad c_{\mathcal{Y}}(y_i, y_j) = \|Y_i - Y_j\|^2, \quad (10)$$

Algorithm 1 provides an efficient implementation of the alternate minimization scheme of Eq. (5). In this pseudo-code, that we detail in Section B.1, SINKHORNOT$_{\varepsilon/8}^{\|\cdot\|^2}$ refers to a solver of the EOT problem for the squared Euclidean cost $c(x, y) = \|x - y\|^2$ in $\mathbb{R}^{\mathrm{D}+1}$. Following Theorem 3.4 and Eq. (1), we use two dual vectors $(f_i) \in \mathbb{R}^{\mathrm{N}}$ and $(g_j) \in \mathbb{R}^{\mathrm{M}}$ to encode the current optimal coupling $\pi_t \in \mathbb{R}^{\mathrm{N} \times \mathrm{M}}$ as:

$$\pi_{ij} = a_i b_j \cdot \exp \tfrac{8}{\varepsilon} [f_i + g_j - \|\overline{X}_i - \overline{Z}_j\|^2]$$

without having to store a large $\mathrm{N} \times \mathrm{M}$ array in memory. In Section B.6, we explain how to compute efficiently the derivatives of the $\mathrm{GW}_\varepsilon$ and $\mathrm{SGW}_\varepsilon$ loss functions with respect to the weights and positions of the samples. Combined

**Algorithm 1** CNT-GW
**Input:** probability weights $a_1, \ldots, a_N$ and $b_1, \ldots, b_M$,
  embeddings $X_1, \ldots, X_N \in \mathbb{R}^D$ and $Y_1, \ldots, Y_M \in \mathbb{R}^E$.
**Output:** optimal dual potentials $f \in \mathbb{R}^N$ and $g \in \mathbb{R}^M$,
  optimal linear map $\Gamma \in \mathbb{R}^{D \times E}$.

1: $\Gamma_0 \leftarrow 0_{D \times E}$ or another initialization in $\mathbb{R}^{D \times E}$
2: $\overline{X}_i \leftarrow [X_i, \frac{1}{2}\|X_i\|^2] \in \mathbb{R}^{D+1}$
3: **while** $\|\Gamma_t - \Gamma_{t+1}\| > \text{tol}$ **do**
4: $\quad \overline{Z}_j \leftarrow [\Gamma_t Y_j, \frac{1}{2}\|Y_j\|^2] \in \mathbb{R}^{D+1}$
5: $\quad f, g \leftarrow \text{SINKHORNOT}_{\varepsilon/8}^{\|\cdot\|^2}(a_i, \overline{X}_i; b_j, \overline{Z}_j)$
6: $\quad \widetilde{X}_j \leftarrow \sum_{i=1}^{N} a_i X_i \cdot \exp\frac{8}{\varepsilon}[f_i + g_j - \|\overline{X}_i - \overline{Z}_j\|^2]$
7: $\quad \Gamma_{t+1} \leftarrow \sum_{j=1}^{M} b_j \widetilde{X}_j Y_j^\top \in \mathbb{R}^{D \times E}$
8: $\quad t \leftarrow t + 1$
9: **end while**

with the guarantees of Section 4, this original contribution opens the door to the use of scalable loss functions based on GW theory.

**CNT Embeddings.** While embeddings $X_i$ and $Y_j$ are easy to compute when $c_\mathcal{X}$ and $c_\mathcal{Y}$ correspond to squared Euclidean costs, simple expressions for the injective embedding maps of Theorem 3.2 may not always be available. In this context, we propose to apply Kernel PCA (Schölkopf et al., 1997) on the kernels $k_\mathcal{X}$ and $k_\mathcal{Y}$ induced by $c_\mathcal{X}$ and $c_\mathcal{Y}$ via Proposition 2.3. Selecting a small number of PCA components allows us to produce low-dimensional embeddings $X_i$ and $Y_j$ that approximately satisfy Eqs. (9-10). In the experiments below, we combine this method with other dimension reduction techniques to handle cases where $c_\mathcal{X}$ and $c_\mathcal{Y}$ are (non-squared) Euclidean norms, kernel-based formulas or even geodesic distances on 3D surfaces (Coifman et al., 2005; Lipman et al., 2010; Panozzo et al., 2013). In Section B.3, we also present an *exact* Algorithm, KERNEL-GW, that works directly with the cost functions $c_X$ and $c_Y$ but has to represent the (possibly infinite-dimensional) operator $\Gamma$ as a $N \times M$ array via the kernel trick.

**Complexity.** We can reuse efficient computational routines that were initially designed for the popular "Wasserstein-2" problem, i.e. OT with a squared Euclidean cost. In Section B, we detail how to use the KeOps library (Feydy et al., 2020; Charlier et al., 2021) to implement the iterations of CNT-GW with a $\mathcal{O}(N + M)$ memory footprint and $\mathcal{O}(NM)$ time complexity. Likewise, we compute kernel PCA embeddings with a $\mathcal{O}(ND + ME)$ memory footprint and $\mathcal{O}(N^2D + M^2E)$ time complexity. All in all, finding a local minimum of the EGW problem for CNT costs with Algorithm 1 thus requires $\mathcal{O}(ND + ME)$ storage space and $\mathcal{O}(N^2D + M^2E + NMn_{\text{inner}}n_{\text{outer}})$ time, where $n_{\text{inner}}$ and $n_{\text{outer}}$ stand for the number of iterates in the SINKHORNOT

and CNT-GW solvers, respectively. These two integers depend on the geometry of the input distributions and usually stay in the 20-200 and 10-50 ranges, respectively. In Section B.5, we also introduce a multiscale MsGW solver that implements the first outer iterations of CNT-GW on coarse sub-samplings of the input distributions.

To conclude, we stress that our Algorithms remain local optimization methods. While our fast solvers can be used on random initializations $\Gamma_0$ to quickly explore the basins of the optimization landscape, computing the global optimum of the EGW problem in full generality remains difficult.

# 6. Experiments

**Setup.** We now present some benchmarks and illustrative experiments, performed on a Dell laptop powered by an Intel i7-12700H CPU and a relatively modest Nvidia RTX 3050 GPU with 4Gb of VRAM. When not specified otherwise, we use embeddings in dimension $D = E = 20$ (that explain $80\%$ to $90\%$ of the variance of the kernel matrices), normalize the input distributions to have a radius of 1 and use a temperature $\varepsilon = 10^{-3}$. For the sake of simplicity, we run the SINKHORNOT solver with a fixed number of 100 iterations and stop all GW solvers when the objective decreases by less than $10^{-5}$ between two consecutive steps. For our methods, this usually corresponds to 10 to 50 "outer" steps depending on the geometry of the problem. We provide full details for our experiments in Section E and leave a thorough exploration of these parameters to future works.

**Benchmarks.** As detailed in Section A.2, the ENTROPIC-GW (EGW) (Peyré et al., 2016) and QUADRATIC-LOWRANK-GW (QLRGW) (Scetbon et al., 2022, Algorithm 2) solvers provide strong baselines for the estimation of optimal GW couplings at scale. In Table 1, we compare our Algorithm 1 (CNTGW), Algorithm 5 (KGW) and Algorithm 7 (MsGW) to these state-of-the-art solvers on pairs of shapes from the CAPOD (Papadakis, 2014), SMAL (Zuffi et al., 2017), KIDS (Rodola et al., 2014), FAUST (Bogo et al., 2014) and MedShapeNet (Li et al., 2025) datasets. We supplement these results by extensive convergence plots, ablation studies and visualizations in Section D. We observe a significant speed-up for our new solvers KGW and CNTGW over their corresponding baselines EGW and QLRGW, while achieving similar GW objectives: Figure 4 confirms that the output of these methods are all qualitatively equivalent. Most importantly, our multiscale MsGW solver consistently outperforms competitors by one to two orders of magnitude. Although heuristic accelerations exist, like LOWRANK-GW (Scetbon et al., 2022, Section 5), they generate important artifacts that strongly degrade the transport plan quality (Figure 4g and h). As a result, our multiscale algorithm is the only GW solver capable of scaling

*Table 1.* Evaluation of different solvers on shape data (solving time and GW objective attained after convergence). We use $c_\mathcal{X}, c_\mathcal{Y} = \|\cdot\|_{\text{geodesic}}$ for KIDS, FAUST, HIPS and VESSELS; $c_\mathcal{X}, c_\mathcal{Y} = \|\cdot\|_{\mathbb{R}^3}$ otherwise. For FAUST, only MSGW was evaluated as it is the only solver converging in less than one hour. True objectives $GW_\varepsilon$ are only reported on HORSES and HANDS; for larger datasets, these could not be computed without memory overflow, and approximate values are given instead (in italic).

| SHAPES | N, M | EGW | | KGW | | QLRGW | | CNTGW | | MSGW | |
|---|---|---|---|---|---|---|---|---|---|---|---|
| | | TIME | $GW_\varepsilon$ | TIME | $GW_\varepsilon$ | TIME | $GW_\varepsilon$ | TIME | $GW_\varepsilon$ | TIME | $GW_\varepsilon$ |
| HORSES | $4k$ | $81s$ | 1.4e-2 | $16s$ | 1.3e-2 | $15s$ | 1.4e-2 | $3s$ | 1.3e-2 | **2s** | 1.3e-2 |
| HANDS | $10k$ | $426s$ | 1.3e-2 | $256s$ | 1.3e-2 | $45s$ | 1.3e-2 | $21s$ | 1.3e-2 | **5s** | 1.3e-2 |
| FEMURS | $25k$ | MEM. | MEM. | MEM. | MEM. | $54s$ | *6.3e-3* | $21s$ | *6.5e-3* | **10s** | *6.6e-3* |
| DOGS | $36k$ | MEM. | MEM. | MEM. | MEM. | $205s$ | *1.2e-2* | $145s$ | *1.2e-2* | **26s** | *1.2e-2* |
| KIDS | $60k$ | MEM. | MEM. | MEM. | MEM. | $2,025s$ | *2.5e-2* | $279s$ | *2.0e-2* | **83s** | *2.0e-2* |
| HIPS | $60k$ | MEM. | MEM. | MEM. | MEM. | $479s$ | *1.0e-2* | $342s$ | *9.4e-3* | **89s** | *9.4e-3* |
| VESSELS | $100k$ | MEM. | MEM. | MEM. | MEM. | $2,444s$ | *7.7e-3* | $891s$ | *7.5e-3* | **196s** | *7.4e-3* |
| FAUST | $177k$ | MEM. | MEM. | MEM. | MEM. | $> 1h$ | - | $> 1h$ | - | **355s** | *1.4e-2* |

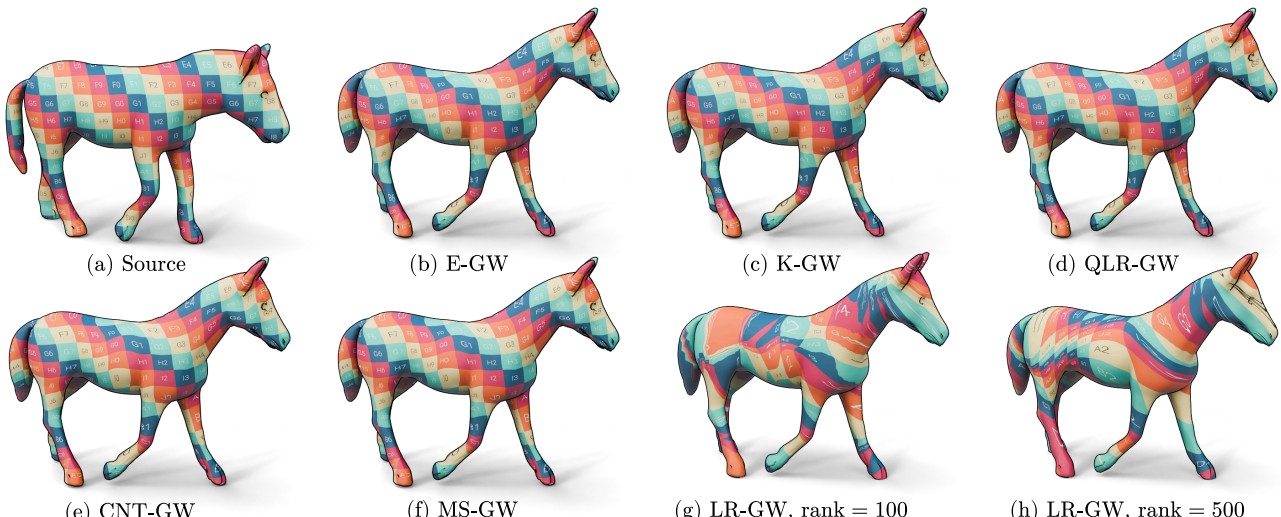

|  |  |  |  |
|---|---|---|---|
| (a) Source | (b) E-GW | (c) K-GW | (d) QLR-GW |
| (e) CNT-GW | (f) MS-GW | (g) LR-GW, rank = 100 | (h) LR-GW, rank = 500 |

*Figure 4.* GW-based texture transfer between horse shapes, using transport plans computed with different solvers (as detailed in Section E). We also illustrate the limitations of heuristic solvers by plotting transport plans obtained with LOWRANK-GW.

to more than N = $100k$ samples in minutes while providing transport maps of sufficient quality for shape matching applications.

**Optimization Landscape.** In Figure 1e, we use MSGW with geodesic costs to transfer a texture from one FAUST surface to another. We guide the registration by specifying 5 pairs of landmarks $(x_1, y_1), \ldots, (x_5, y_5)$ that should be matched with each other at the top of the skull and at the tips of the arms and legs. They induce a covariance matrix $\Gamma_0 = \sum_{k=1}^5 X_k Y_k^\top$ in feature space that we use to initialize the alternate minimization: this ensures that we fall in the "correct", user-defined local minimum of the GW objective.

Going further, we explore the optimization landscape of the GW problem as we match two poses from the KIDS dataset, with cost functions that correspond to geodesic distances embedded in spaces of dimension D = E = 20 (Panozzo et al., 2013). We sample the coefficients of 500 initializations $\Gamma_0 \in \mathbb{R}^{D \times E}$ uniformly at random (with a

relevant scale factor), and run the "coarse" phase of MSGW with this diverse collection of seeds. We identify 25 distinct local minima, that we refine in the second step of MSGW to obtain high-resolution transport plans $\pi$. As illustrated in Figure 5, the local minima of the GW problem correspond to body part permutations, with the 8 best candidates attracting 82% of our random seeds $\Gamma_0$. In the lower right corner, the global optimum corresponds to the widest basin of attraction and represents the correct alignment of limbs.

We perform a Principal Component Analysis on the 25 linear maps $\Gamma \in \mathbb{R}^{D \times E}$ that encode these local minima, weighted by the number of seeds that they attracted. This allows us to represent them in the most relevant 2-dimensional plane of the full operator space, with contour lines of the objective function $U_\varepsilon$ of Eq. (7) in the background. This figure reveals the structure of the matching problem, as a left-right symmetry dominates the visualization.

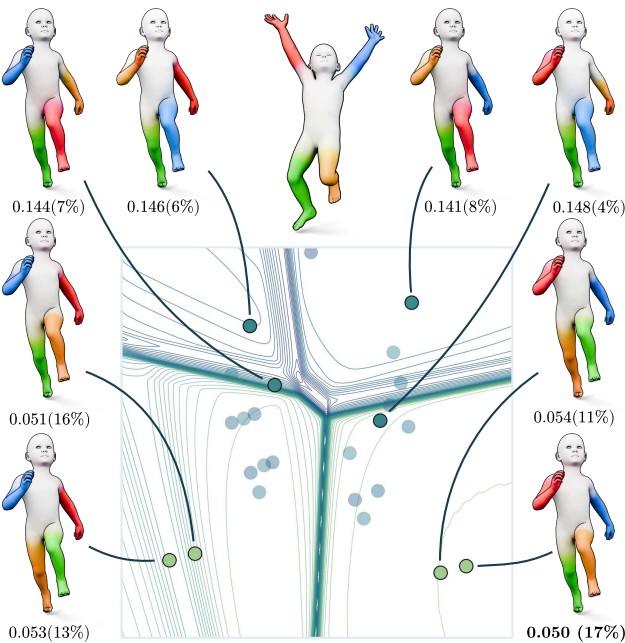

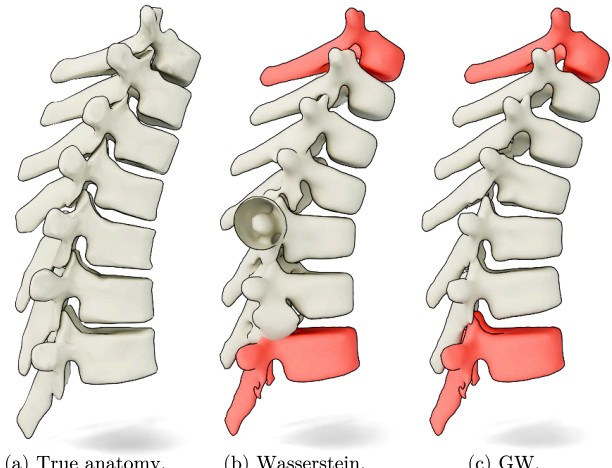

(a) True anatomy.    (b) Wasserstein.    (c) GW.

*Figure 6.* (a) Interpolating between the first (top) and seventh (bottom) thoracic vertebrae. We compute barycenters on normalized data, and align them with true anatomical positions for visualization purposes. (b) Wasserstein barycenters are now affordable, but create many topological artifacts (Agueh & Carlier, 2011). (c) We compute GW barycenters for the Euclidean cost in $\mathbb{R}^3$. The GW metric puts more emphasis on topology preservation and could become a versatile baseline for 3D shape analysis.

*Figure 5.* Optimization landscape of the EGW problem, visualized in a dual plane. We match a source point cloud (with arms raised) to a target pose (running). We highlight the 8 best minima, the corresponding EGW losses and the proportions of random seeds $\Gamma_0$ that fell in their attraction basins.

**GW Barycenters.** Since our solvers provide gradients for the $\mathrm{GW}_\varepsilon$ objective and the debiased $\mathrm{SGW}_\varepsilon$ divergence, we can easily implement gradient flows and other variational schemes that involve the GW metric. To illustrate this, we compute GW barycenters between anatomical shapes from the BodyParts3D[2] dataset in Figure 6. Specifically, we sample $\mathrm{N} = 3{,}000$ points uniformly on the source and target surfaces to create two distributions of points $\alpha$ and $\beta$ in $\mathbb{R}^3$. Then, we perform gradient descent on a point cloud $z_1, \ldots, z_\mathrm{N}$ that parameterizes a measure $\gamma_z = \frac{1}{\mathrm{N}} \sum_i \delta_{z_i}$ in order to minimize the weighted objective:

$$(1 - \lambda) \cdot \mathrm{SGW}_\varepsilon(\gamma_z, \alpha) + \lambda \cdot \mathrm{SGW}_\varepsilon(\gamma_z, \beta) \,,$$

where $\lambda \in [0, 1]$ is an interpolation slider. We turn the resulting point clouds into surface meshes using Poisson surface reconstruction (Kazhdan et al., 2006).

## 7. Conclusion

While these experiments validate our theoretical findings, further development is required to turn these illustrations into competitive state-of-the-art methods for shape registration or domain adaptation. Specifically, future works should adapt our approach to the unbalanced (Séjourné et al., 2021) or fused-GW settings (Vayer et al., 2020; Thual et al.,

---

[2](c) The Database Center for Life Science licensed under CC Attribution-Share Alike 2.1, Japan.

2022). Furthermore, the impact of cost functions, embedding dimensions or annealing schedules for the temperature $\varepsilon$ (Chizat, 2024) remains to be fully characterized.

Despite these necessary extensions, we believe our work offers significant conceptual value to the field. By establishing a surprising connection between a broad class of Quadratic Assignment Problems (GW with CNT costs) and the linear registration of point clouds, we provide a new geometric perspective on structured data alignment. This perspective closely relates to Procrustes–Wasserstein (PW) approaches (Grave et al., 2019; Alvarez-Melis et al., 2019), which combine rigid alignment with optimal transport–based matching. A deeper comparison between the GW and PW frameworks would certainly provide valuable insights into both problems, from both geometric and algorithmic viewpoints. This bridge also invites further investigation into the functional maps framework, where analogous problems have been explored in spectral or data-driven feature spaces (Ovsjanikov et al., 2012; Ren et al., 2020; Pai et al., 2021), and may pave the way for the development of provably approximate global optimizers (Jubran et al., 2021).

## Acknowledgements

This work was supported by the French "Agence Nationale de la Recherche" via the "PR[AI]RIE-PSAI" project (ANR-23-IACL-0008). We would also like to thank Robin Magnet for relevant comments and his invaluable help with the rendering of the 3D figures.

## Impact Statement

This paper presents work whose goal is to advance the field of machine learning. There are many potential societal consequences of our work, none of which we feel must be specifically highlighted here.

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

# A. Additional Background

## A.1. The Sinkhorn Algorithm

By convex duality, the EOT problem is equivalent to the maximization of the following objective:

$$\mathrm{OT}_\varepsilon(\alpha, \beta) = \max_{f \in \mathcal{C}(X)} \max_{g \in \mathcal{C}(Y)} \int f(x)\mathrm{d}\alpha(x) + \int g(y)\mathrm{d}\beta(y) - \varepsilon \int \left( \exp\left( \frac{f(x) + g(y) - c(x, y)}{\varepsilon} \right) - 1 \right) \mathrm{d}\alpha(x)\mathrm{d}\beta(y).$$

The Sinkhorn algorithm solves this dual problem by block coordinate ascent. Starting from an initial potential $g_0 \in \mathcal{C}(Y)$, it computes a sequence of functions $(f_s, g_s) \in \mathcal{C}(X) \times \mathcal{C}(Y)$ through alternating updates:

$$f_{s+1} = \mathcal{S}_{\varepsilon, \beta}(g_s) \quad \text{and} \quad g_{s+1} = \mathcal{S}_{\varepsilon, \alpha}(f_{s+1}),$$

where the Sinkhorn operators $\mathcal{S}_{\varepsilon, \beta}$ and $\mathcal{S}_\alpha$ are defined as:

$$\begin{aligned} \mathcal{S}_{\varepsilon, \beta}(g) &: x \in X \mapsto -\varepsilon \cdot \log \int \exp\left( \frac{g(y) - c(x, y)}{\varepsilon} \right) \mathrm{d}\beta(y), \\ \mathcal{S}_{\varepsilon, \alpha}(f) &: y \in Y \mapsto -\varepsilon \cdot \log \int \exp\left( \frac{f(x) - c(x, y)}{\varepsilon} \right) \mathrm{d}\alpha(x). \end{aligned} \quad (11)$$

The sequence $(f_s, g_s)$ converges to the optimal dual potentials $(f^*, g^*)$, recovering the optimal EOT plan $\pi^*$ via:

$$\forall\, x, y \in X \times Y, \quad \frac{\mathrm{d}\pi^*}{\mathrm{d}\alpha\mathrm{d}\beta}(x, y) = \exp\left( \frac{f^*(x) + g^*(y) - c(x, y)}{\varepsilon} \right).$$

However, the intermediate $\pi_s$ obtained by using $(f_s, g_s)$ instead of $(f^*, g^*)$ in the previous formula are not valid couplings as they do not exactly satisfy the marginal constraints. For discrete measures of size N, the number of iterations required to reduce $\mathrm{Err}(\pi_s)$ to a threshold $\delta$ scales as $s = O(\delta^{-2} \cdot (\log(\mathrm{N}) + \|c\|_\infty / \varepsilon))$ (Altschuler et al., 2017), where $\|c\|_\infty = \max(c) - \min(c)$ and:

$$\mathrm{Err}(\pi_s) = \left\| \int \mathrm{d}\pi_s(\cdot, y) - \alpha \right\|_1 + \left\| \int \mathrm{d}\pi_s(x, \cdot) - \beta \right\|_1.$$

Therefore, while the convergence rate is not strongly influenced by the input size N, it depends on the entropic regularization: keeping $\varepsilon$ sufficiently large is crucial to ensure fast computations.

## A.2. Review of the Computational Gromov-Wasserstein Literature

This section provides an overview of existing GW solvers, including both unregularized and entropic approaches. We also discuss algorithms designed for GW extensions such as Fused (Vayer et al., 2020) and Unbalanced (Séjourné et al., 2021) Gromov-Wasserstein, as these frameworks offer a broader perspective on current computational challenges. To maintain clarity, we adopt the notation from Section 2 for general measures and Section 5 for finite discretizations.

### A.2.1. UNREGULARIZED GROMOV-WASSERSTEIN SOLVERS

**Global Optimization Methods.** GW is closely related to the quadratic assignement problem (QAP), which is known to be NP-hard (Burkard et al., 1998). Consequently, finding a global minimum is computationally expensive. Recent works have proposed relaxations to make this global search tractable: (Mula & Nouy, 2024) formulate GW as a generalized moment problem that can be truncated at the desired degree of precision, while (Chen et al., 2024) propose a semi-definite relaxation solvable in polynomial time. Alternatively, (Ryner et al., 2023) exploit the structure of squared Euclidean costs to design a cutting plane algorithm that efficiently explores the set of admissible solutions. Despite these advances, global methods remain slow and are limited to inputs containing at most a few hundred points.

**Conditional Gradient Descent (Franke-Wolfe).** By contrast, local minima can be obtained in polynomial time using first-order methods. The standard approach is the Conditional Gradient Descent (or Franke-Wolfe) algorithm (Vayer et al., 2019a), whose updates are given by:

$$\pi_{t+1} = (1 - \gamma) \cdot \pi_t + \gamma \cdot \tilde{\pi}_t \quad \text{where} \quad \tilde{\pi}_t = \operatorname*{argmin}_{\pi \in \mathcal{M}(\alpha, \beta)} \int \nabla_{\pi_t} \mathcal{L} \cdot \mathrm{d}\pi, \quad (12)$$

where $\gamma \in [0, 1]$ is either fixed or selected by line search. Each step is equivalent to a linear optimal transport problem with cost $c = \nabla_{\pi_t} \mathcal{L}$, effectively transforming GW into a sequence of OT subproblems solvable in cubic time. For finite measures, these subproblems take the form:

$$\tilde{\pi}^{t+1} = \underset{\pi \in \mathcal{M}(\alpha,\beta)}{\operatorname{argmin}} \sum C_{ij}^t \pi_{ij} \quad \text{where} \quad C_{ij}^t = \sum_{kl} (c_{\mathcal{X}}(x_i, x_k) - c_{\mathcal{Y}}(y_j, y_l))^2 \pi_{kl}^t \; .$$

When the base costs $c_X$ and $c_Y$ are CNT, the loss $\mathcal{L}$ is concave: it allows the step size $\gamma$ to be set to 1, ensuring convergence without needing to tune this parameter. (Maron & Lipman, 2018).

**Cheaper Gromov-Wasserstein Variants.** Many practical applications only require a quantification of similarity between $\alpha$ and $\beta$ rather than an exact matching. In such cases, cheaper alternatives to the full GW problem may suffice. This category includes SLICED GW (Vayer et al., 2019b), QUANTIZED-GW (Chowdhury et al., 2021), MINIBATCH-GW (Fatras et al., 2021) and LINEAR-GW (Beier et al., 2022), as well as GW-inspired distances for Gaussian mixture models (Salmona et al., 2023). While these heuristics can be computed in quadratic or even linear time, they rely on heavy approximations and are less suitable for tasks requiring precise point-wise correspondences such as shape matching.

A.2.2. ENTROPIC GROMOV-WASSERSTEIN SOLVERS

**Exact Entropic Solvers.** Entropic regularization, introduced by (Solomon et al., 2016; Peyré et al., 2016), allows GW to be solved using the efficient Sinkhorn algorithm. The standard solver, ENTROPIC-GW, is a Projected Gradient Descent (PGD) in the KL-divergence geometry. This method, widely adopted in the literature, uses updates of the form:

$$\pi_{t+1} = \underset{\pi \in \mathcal{M}(\alpha,\beta)}{\operatorname{argmin}} \left( \int \nabla_{\pi_t} \mathcal{L} \cdot \mathrm{d}\pi + \varepsilon \mathrm{KL}\left(\pi \| \alpha \otimes \beta\right) \right).$$

These updates are similar to the Franke-Wolfe steps of Eq. (12) and can be viewed as a GW adaptation of the *soft assign* method proposed in (Gold & Rangarajan, 2002). Focusing on Unbalanced GW, (Séjourné et al., 2021) also propose a bilinear relaxation of the EGW loss which can be solved by alternate minimization:

$$\pi_{t+1} = \underset{\pi \in \mathcal{M}(\alpha,\beta)}{\operatorname{argmin}} \left[ \int \left( \|x - x'\|^2 - \|y - y'\|^2 \right)^2 \mathrm{d}\pi(x, y)\mathrm{d}\pi_t(x', y') + \varepsilon \mathrm{KL}\left(\pi \| \alpha \otimes \beta\right) \right].$$

For CNT base costs, this relaxation is tight and its minimum coincides with the true EGW solution. In the finite case, this technique is equivalent to ENTROPIC-GW and consists in solving a sequence of EOT problems:

$$\pi^{t+1} = \underset{\pi \in \mathcal{M}(\alpha,\beta)}{\operatorname{argmin}} \left( \sum C_{ij}^t \pi_{ij} + \varepsilon \mathrm{KL}\left(\pi \| \alpha \otimes \beta\right) \right) \quad \text{where} \quad C_{ij}^t = \sum_{kl} (c_{\mathcal{X}}(x_i, x_k) - c_{\mathcal{Y}}(y_j, y_l))^2 \pi_{kl}^t \; . \tag{13}$$

Using the Sinkhorn algorithm, these steps can be computed in $\mathcal{O}(\mathrm{NM})$ time. The overall complexity, however, is dominated by the computation of the cost matrix $C^t$ that takes $\mathcal{O}(\mathrm{N}^2\mathrm{M}^2)$ time. In practice, the same solution can be obtained with $C_{ij}^t = -2\sum_k c_{\mathcal{X}}(x_i, x_k) \sum_l c_{\mathcal{Y}}(y_j, y_l)\pi_{kl}^t$ instead (Peyré et al., 2016), so that the overall running time becomes cubic. When costs are squared norms, (Scetbon et al., 2022) further reduce this computation to $\mathcal{O}(\mathrm{NMD})$; similarly, the dual solver DUAL-GW of (Rioux et al., 2024) achieves quadratic complexity and is guaranteed to converge.

**Accelerated Approximations.** Apart from the squared norm case, solving EGW remains expensive due to the computation of the matrix $C^t$. The algorithm complexity can still be improved thanks to several approximation strategies introduced in the last few years. SAMPLED-GW (Kerdoncuff et al., 2021) computes $C^t$ using a random subset of indices $k$ and $l$ in Eq. (13), while SPARSE-GW (Li et al., 2023) applies a sparsity mask to fasten both cost matrix computations and Sinkhorn iterations. Alternatively, QUADRATIC-LOWRANK-GW (Scetbon et al., 2022) approximates the input matrices $(C_{ij}^{\mathcal{X}}) = (c_{\mathcal{X}}(x_i, x_j))$ and $(C_{ij}^{\mathcal{Y}}) = (c_{\mathcal{Y}}(y_i, y_j))$ via low-rank factorization to accelerate the estimation of $C^t$. These three algorithms reach a quadratic running time, improving significantly the scalability of the original EGW solvers. With LOWRANK-GW, (Scetbon et al., 2022) combine the low-rank cost reduction of QUADRATIC-LOWRANK-GW with a low rank constraint on the transport plan, further reducing computation time to $\mathcal{O}(\mathrm{N} + \mathrm{M})$ but generating quantization artifacts in the transport plan. Finally, ULOT (Unsupervised Learning of Optimal Transport) (Mazelet et al., 2025) trains neural networks to predict optimal couplings, removing the optimization burden at inference time – although its training phase keeps a cubic complexity.

**Proximal Solvers.** A last strategy consists in adding a proximal term to the GW loss rather than an entropic regularization (Xu et al., 2019a;b; Kerdoncuff et al., 2021). The updates of PROXIMAL-GW are given by:

$$\pi_{t+1} = \operatorname*{argmin}_{\pi \in \mathcal{M}(\alpha,\beta)} \left( \int \nabla_{\pi_t} \mathcal{L} \cdot \mathrm{d}\pi + \varepsilon \mathrm{KL}\left(\pi \| \pi_t\right) \right).$$

This approach ensures the convergence of $(\pi_t)$ to the minimum of the *unregularized* GW problem while remaining Sinkhorn-compatible. Indeed, in the finite case, it can be solved using:

$$\pi^{t+1} = \operatorname*{argmin}_{\pi \in \mathcal{M}(\alpha,\beta)} \left( \sum C_{ij}^t \pi_{ij} + \varepsilon \mathrm{KL}\left(\pi \| \alpha \otimes \beta\right) \right) \quad \text{where} \quad C_{ij}^t = -2 \sum_{kl} c_{\mathcal{X}}(x_i, x_k) c_{\mathcal{Y}}(y_j, y_l) \pi_{kl}^t - \varepsilon \log(\pi_{ij}^t).$$

This alternative is attractive when exact, unregularized GW solution are required. However, it has other limitations. The proven convergence rates only apply to finite measures and degrade as the number of points increases (Xie et al., 2020). The sequential dependency of the proximal term (due to the $\log(\pi^t)$ term) complicates its integration with the advanced scalability techniques described in Section 5, such as lazy tensor manipulation with KeOps or multiscaling, which are essential for large-scale applications. Finally, in many machine learning contexts, the entropic term acts as a regularizer against noisy input, making EGW preferable to the unregularized GW formulation.

### A.2.3. CHOICE OF BASELINES

Despite the wide variety of Gromov-Wasserstein solvers available, only entropic-regularized methods (Section A.2.2) scale to inputs containing thousands of points, and only accelerated approximations can handle tens of thousands. Consequently, we focus our benchmarks on two main competitors: ENTROPIC-GW, the reference solver for exact EGW problems, and QUADRATIC-LOWRANK-GW, the current state-of-the-art for approximate solving. We discarded heuristic methods and linear-time approximations as they introduce artifacts that are unsuited to our shape matching applications (Figure 4g and h). Finally, we excluded other potential baselines for the following reasons:

- SAMPLED-GW relies on stochastic sampling: due to this randomness, the solver does not converges effectively which prevents reliable comparisons of convergence speed.

- PROXIMAL-GW addresses a distinct problem formulation, and does not scale to $N > 10^4$ points.

- Methods such as SPARSE-GW or ULOT rely on different computational backends or address different settings, making fair comparisons difficult to perform.

Note that although we did not include them in our quantitative benchmarks, we still discuss the convergence properties of SAMPLED-GW and PROXIMAL-GW in Section D.1. Finally, we stress that the quadratic-time methods excluded from our benchmarks were originally validated on inputs with at most a few thousand nodes. Therefore, while we do not provide a direct performance comparison against all pre-existing GW solvers, our experiments demonstrate scalability to problems whose sizes are 10 to 100 times larger than those addressed in prior works.

# B. Detailed Implementation

We detail here the pseudo-code of the main routines used in our method.

## B.1. Sinkhorn Algorithms

**Standard Sinkhorn.** Algorithm 2 details the standard version of the Sinkhorn algorithm. It implement the discrete counterpart of the Sinkhorn operators defined in Equation (11), adressing the discrete EOT problem:

$$\min_{\pi \in \mathbb{R}^{N \times M}} \left[ \sum_{ij} C_{ij} \pi_{ij} + \varepsilon \sum_{ij} \pi_{ij} \log\left(\frac{\pi_{ij}}{a_i b_j}\right) \right] \quad \text{subject to} \quad \sum_{j} \pi_{ij} = a_i \quad \text{and} \quad \sum_{i} \pi_{ij} = b_j,$$

where $C \in \mathbb{R}^{N \times M}$ is a cost matrix and $a \in \mathbb{R}^N, b \in \mathbb{R}^M$ are two positive vectors summing to 1. From the final dual potentials $(f^{n_{\text{inner}}}, g^{n_{\text{inner}}})$, we recover the approximate optimal plan using Equation (1):

$$\forall\, i, j, \quad \pi_{ij} = a_i b_j \exp((f_i^{n_{\text{inner}}} + g_j^{n_{\text{inner}}} - C_{ij})/\varepsilon).$$

---

**Algorithm 2** SINKHORN.

---

**Parameters:** $\varepsilon > 0$, $n_{\text{inner}} > 0$.
**Inputs:** $a \in \mathbb{R}^N, b \in \mathbb{R}^M, C \in \mathbb{R}^{N \times M}$.
Initialize $f^0 \in \mathbb{R}^N, g^0 \in \mathbb{R}^M$
**for** $n = 0$ to $n_{\text{inner}} - 1$ **do**
   $f^{n+1} \leftarrow -\varepsilon \cdot [\log \sum_j \exp(\log(b_j) + (g_j^n - C_{ij})/\varepsilon)]_i$
   $g^{n+1} \leftarrow -\varepsilon \cdot [\log \sum_i \exp(\log(a_i) + (f_i^{n+1} - C_{ij})/\varepsilon)]_j$
**end for**
**Return** $f^{n_{\text{inner}}}, g^{n_{\text{inner}}}$

---

**Symmetrized Sinkhorn.** Algorithm 3 presents the symmetrized variant of Sinkhorn introduced in (Feydy, 2020, Algorithm 3.4). This variant ensures that the resulting pair $(f^{n_{\text{inner}}}, g^{n_{\text{inner}}})$ remains the same if the roles of $\alpha$ and $\beta$ are exchanged with each other. As discussed in Section D.1, we also notice a drastic convergence speed-up when using symmetrized Sinkhorn in EGW solvers.

---

**Algorithm 3** SINKHORN-SYMM.

---

**Parameters:** $\varepsilon > 0$, $n_{\text{inner}} > 0$.
**Inputs:** $a \in \mathbb{R}^N, b \in \mathbb{R}^M, C \in \mathbb{R}^{N \times M}$.
Initialize $f^0 \in \mathbb{R}^N, g^0 \in \mathbb{R}^M$
**for** $n = 0$ to $n_{\text{inner}} - 1$ **do**
   $f^{n+1} \leftarrow \frac{1}{2}(f^n - \varepsilon \cdot [\log \sum_j \exp(\log(b_j) + (g_j^n - C_{ij})/\varepsilon)]_i)$
   $g^{n+1} \leftarrow \frac{1}{2}(g^n - \varepsilon \cdot [\log \sum_i \exp(\log(a_i) + (f_i^n - C_{ij})/\varepsilon)]_j)$
**end for**
**Return** $f^{n_{\text{inner}}}, g^{n_{\text{inner}}}$

---

**Warm-Start Initialization.** The number of iterations needed to reach convergence is highly sensitive to the initialization of the potentials $(f^0, g^0)$. Since EGW solvers involve a sequence of successive Sinhorn calls, we employ a warm-start strategy: the optimal potentials $(f, g)$ from one optimization step are used to initialize the Sinkhorn loop of the next step. This greatly reduces the number of iterations required for convergence.

**Sinkhorn Annealing.** In order to get sharp transport plans, a common strategy consists is to decrease the regularization parameter $\varepsilon$ at every Sinkhorn iteration. As proven in (Chizat, 2024), using a squared root decay $\varepsilon_n = \varepsilon_0/\sqrt{n}$ guarantees the convergence of the reconstructed transport plans towards the unregularized OT solution. We use this annealing scheme to apply the transfer texture of Figure 1, only in the last step of the EGW solver.

## B.2. Kernel matrices computations.

**Centered Kernels From Costs.** Algorithm 4 computes the kernel matrix $K$ associated with a CNT cost matrix $(C_{ij}) = (c(x_i, x_j))$ using the equation of Proposition 2.3:

$$k(x, x') = \big(c(x, x_0) + c(x_0, x') - c(x, x')\big) / 2 . \tag{14}$$

The procedure includes a centering step (line 3) which ensures that the embeddings are centered in the kernel space. This centering also ensures that the output is independent of the choice of base point $x_0$; we can therefore take 1 as base index in line 2 of the algorithm without inducing any bias.

---

**Algorithm 4** KERNEL

> **Inputs:** $C \in \mathbb{R}^{N \times N}, a \in \mathbb{R}^N$.
> $K \leftarrow [(C_{i1} + C_{1j} - C_{ij})/2]_{ij}$
> $K \leftarrow [K_{ij} - \sum_i a_i K_{ij} - \sum_j a_j K_{ij} + \sum_{ij} a_i a_j K_{ij}]_{ij}$
> **Return** $K$

---

**Kernel PCA.** The Kernel PCA of a centered kernel matrix $K \in \mathbb{R}^{N \times N}$ is computed from its eigendecomposition. Let $\Lambda_{1:D} \in \mathbb{R}^{D \times D}$ denote the diagonal matrix containing the D largest eigenvalues of $K$ and $V_{1:D} \in \mathbb{R}^{N \times D}$ the corresponding eigenvectors. The Kernel PCA of $K$ in dimension D is then given by $X_i = V_{i,1:D}\sqrt{\Lambda_{1:D}} \in \mathbb{R}^D$ for all $i \in \{1, \dots, N\}$.

When the cost $c$ is given by a simple mathematical formula, e.g. an analytical function of the Euclidean norm, the kernel of Equation (14) admits a simple formula as well and the kernel matrix $K$ can be implemented as a *symbolic tensor* with the KeOps library. Combined with the function eigsh of the scipy library, the truncated PCA of $K$ can thus be obtained without explicitly storing the full matrix. This allows us to perform Kernel PCAs on $N > 10^4$ points without memory overflows.

## B.3. Exact Kernel Embeddings with KERNEL-GW

Rather than relying on approximate embeddings, the alternate minimization of Equation (5) can be implemented exactly using the *kernel trick*. If $k_{\mathcal{X}}$ and $k_{\mathcal{Y}}$ are the kernels obtained from $c_{\mathcal{X}}$ and $c_{\mathcal{Y}}$ by Proposition 2.3, we have:

$$\langle \varphi(x)\psi(y)^\top, \varphi(x')\psi(y')^\top \rangle_{\mathrm{HS}} = k_{\mathcal{X}}(x, x')k_{\mathcal{Y}}(y, y') .$$

Denoting by $\Gamma = \sum_{ij} \varphi(x_i)\psi(y_j)^\top \pi_{ij}$ the cross-correlation of a given transport plan $\pi \in \mathbb{R}^{N \times M}$, the cost $c_\Gamma$ becomes:

$$c_\Gamma(x, y) = -4 \cdot k_{\mathcal{X}}(x, x)k_{\mathcal{Y}}(y, y) - 16 \cdot \sum_{k,l} k_{\mathcal{X}}(x, x_k)k_{\mathcal{Y}}(y, y_l)\pi_{kl} . \tag{15}$$

Note that for the sake of simplicity, we include the factor 8 that was factored out in our statement of Theorem 3.2. Algorithm 5 thus implements the optimization scheme without handling $\Gamma$ explicitly (where $C^X, C^Y$ are the cost matrices of $(x_1, \dots, x_N)$ and $(y_1, \dots, y_M)$). KERNEL-GW provides few computational benefits compared to ENTROPIC-GW, as both methods store a large matrices of size $N \times M$. Yet, it will help us to isolate the specific effects of kernelization on solver convergence in Section D.1, discarding the approximation errors induced by CNT-GW and LOWRANK-GW.

## B.4. Adaptive Sinkhorn Iterations.

In Algorithm 6, we present the adaptive variant of Algorithm 5 where the fixed iteration count $n_{\mathrm{inner}}$ is replaced by an adaptive schedule. While an initial value $n_{\mathrm{inner}}^0$ is still required, the algorithm automatically increases precision when needed: therefore, this hyperparameter can be set to a small arbitrary value. Since that EGW-LOSS$(\pi^t)$ is guaranteed to decrease at each step when $\pi^t$ corresponds to the true Sinkhorn solution, any violation of this decrease indicates a lack of Sinkhorn convergence: therefore, the adaptive method is guaranteed to converge to a EGW local minimum for $t \to +\infty$ independently of the choice $n_{\mathrm{inner}}^0$ (although this initial value can influence which local minimum will be attained).

Adaptive scheduling can also be applied to CNT-GW. In this case, we use the squared Euclidean norm of the embedding space as the base cost for the computations of EGW-LOSS, since it can be computed more efficiently.

---

**Algorithm 5** KERNEL-GW.

---

**Parameters:** $\varepsilon > 0$, $n_{\text{outer}} > 0$, $n_{\text{inner}} > 0$.
**Inputs:** $a \in \mathbb{R}^N$, $b \in \mathbb{R}^M$, $C^X \in \mathbb{R}^{N \times N}$, $C^Y \in \mathbb{R}^{M \times M}$.
$K^X, K^Y \leftarrow$ KERNEL$(C^X, a)$, KERNEL$(C^Y, b)$
Initialize $\pi^0$
**for** $t = 0$ to $n_{\text{outer}} - 1$ **do**
$\quad C \leftarrow [-4 \cdot K^X_{ii} K^Y_{jj} - 16 \cdot \sum_{kl} K^X_{ik} K^Y_{jl} \pi^t_{kl}]_{ij}$
$\quad f^{t+1}, g^{t+1} \leftarrow$ SINKHORN-SYMM$_{\varepsilon, n_{\text{inner}}}(a, b, C)$
$\quad \pi^{t+1} = [a_i b_j \exp((f^{t+1}_i + g^{t+1}_j - C_{ij})/\varepsilon)]_{ij}$
**end for**
**Return** $\pi^{n_{\text{outer}}}, f^{n_{\text{outer}}}, g^{n_{\text{outer}}}$

---

**Algorithm 6** KERNEL-GW with adaptive Sinkhorn iterations.

---

**Parameters:** $\varepsilon > 0$, $T > 0$, $n^0_{\text{inner}} > 0$.
**Inputs:** $a \in \mathbb{R}^N$, $b \in \mathbb{R}^M$, $C^X \in \mathbb{R}^{N \times N}$, $C^X \in \mathbb{R}^{M \times M}$.
$K^X, K^Y \leftarrow$ KERNEL$(C^X, a)$, KERNEL$(C^Y, b)$
Initialize $\pi^0$
**for** $t = 0$ to $n_{\text{outer}} - 1$ **do**
$\quad C \leftarrow [-4 \cdot K^X_{ii} K^Y_{jj} - 16 \cdot \sum_{kl} K^X_{ik} K^Y_{jl} \pi^t_{kl}]_{ij}$
$\quad f^{t+1}, g^{t+1} \leftarrow$ SINKHORN-SYMM$_{\varepsilon, n^t_{\text{inner}}}(a, b, C)$
$\quad \pi^{t+1} = [a_i b_j \exp((f^{t+1}_i + g^{t+1}_j - C_{ij})/\varepsilon)]_{ij}$
$\quad$**if** EGW-LOSS$(\pi^{t+1}) >$ EGW-LOSS$(\pi^t)$ **then**
$\quad\quad n^{t+1}_{\text{inner}} \leftarrow 2 \cdot n^t_{\text{inner}}$
$\quad$**else**
$\quad\quad n^{t+1}_{\text{inner}} \leftarrow n^t_{\text{inner}}$
$\quad$**end if**
**end for**
**Return** $\pi^{n_{\text{outer}}}, f^{n_{\text{outer}}}, g^{n_{\text{outer}}}$

---

### B.5. Implementation of MULTISCALE-GW

Algorithm 7 directly adapts the multiscale technique of (Feydy, 2020) to CNT-GW. The COARSEN function is implemented using K-Means clustering; the coarse points $X^c \in \mathbb{R}^{N^c \times D}$ are the K-Means centroids and the weights $a^c$ are the aggregated weights of the clusters:

$$a^c_i = \sum_{j \in \mathcal{C}_i} a_j \quad \text{where} \quad \mathcal{C}_i = \left\{ j = 1, \ldots, N \text{ s.t. } i = \operatorname*{argmin}_{k=1,\ldots,N^c} \|X_j - X^c_k\|^2 \right\}.$$

The Sinkhorn potentials $(f, g)$ are upsampled from coarse to fine using the block-wise interpolation described in Algorithm 8, and are used together with the coarse dual solution $\Gamma^c$ to initialize the large-scale solver.

---

**Algorithm 7** MULTISCALE-GW.

---

**Parameters:** $\varepsilon > 0$, $D > 0$, $\rho \in [0, 1]$.
**Inputs:** $a \in \mathbb{R}^N$, $b \in \mathbb{R}^M$, $X \in \mathbb{R}^{N \times D}$, $Y \in Y^{M \times E}$.
$(X^c, a^c), (Y^c, b^c) \leftarrow$ COARSEN$(X, a, k = \lceil \rho N \rceil)$, COARSEN$(Y, b, k = \lceil \rho M \rceil)$
$\Gamma^c, f^c, g^c \leftarrow$ CNT-GW$(a^c, b^c, X^c, Y^c)$
$f, g \leftarrow$ INTERPOLATE$(g^c, X, Y^c, b^c)$, INTERPOLATE$(f^c, Y, X^c, a^c)$
$\Gamma, f, g \leftarrow$ CNT-GW$(a, b, X, Y \mid \Gamma^0 = \Gamma^c, (f^0, g^0) = (f, g))$
**Return** $\Gamma, f, g$

---

---

**Algorithm 8** INTERPOLATE.

---

**Inputs:** $g^c \in \mathbb{R}^{M^c}, X \in \mathbb{R}^{N \times D}, Y^c \in R^{M^c \times D}, b^c \in R^{M^c}$.
$C \leftarrow [-4 \|X_i\|^2 \|Y_j^c\|^2 - 16 \sum_{rs} X_{ir} Y_{js}^c \Gamma_{rs}]_{ij}$
$f \leftarrow -\varepsilon \cdot [\log \sum_j \exp(\log(b_j^c) + (g_j^c - C_{ij})/\varepsilon)]_i$
**Return** $f$

---

## B.6. EGW Gradient Computations.

We finally detail the computation of EGW gradients with respect to the input locations $x_i$ and $y_j$. Specifically, we seek:

$$G_i^X = \nabla_{x_i} \mathrm{GW}_\varepsilon(\alpha, \beta) \quad \text{and} \quad G_j^Y = \nabla_{y_j} \mathrm{GW}_\varepsilon(\alpha, \beta) \quad \text{where} \quad \alpha = \sum_i a_i \delta_{x_i} \quad \text{and} \quad \beta = \sum_i b_j \delta_{y_j}.$$

We could compute them by differentiating through the whole EGW solver using PyTorch's autograd: however, this method would be costly in time and memory and would not scale on large point clouds. Therefore, we need more efficient methods to perform these computations. Assuming that the gradient is well defined, we rather apply the envelope theorem to write:

$$\nabla_{x_i} \mathrm{GW}_\varepsilon(\alpha, \beta) = \nabla_{x_i} C(\varphi_\sharp \alpha, \psi_\sharp \beta) + \nabla_{x_i} \mathrm{OT}_\varepsilon^{\Gamma^*}(\varphi_\sharp \alpha, \psi_\sharp \beta).$$

The main complexity lies in the presence of the kernel embeddings $\varphi$ and $\psi$ that we need to differentiate through. We provide two techniques to perform these computations: the first one, based on the KERNEL-GW algorithm, outputs exact EGW gradient while the second one provides approximate gradients based using CNT-GW.

**Exact Gradient Computation** The kernel trick provides an exact method to compute EGW gradients. First, the constant can be differentiated automatically with PyTorch using the explicit formula:

$$C(\varphi_\sharp \alpha, \psi_\sharp \beta) = \sum_{ik} c_{\mathcal{X}}(x_i, x_k)^2 a_i a_k + \sum_{jl} c_{\mathcal{Y}}(y_j, y_l)^2 b_j b_l - 4 \sum_i k_{\mathcal{X}}(x_i, x_i) a_i \sum_j k_{\mathcal{Y}}(y_j, y_j) b_j,$$

To differentiate the EOT loss, we use the method of (Feydy et al., 2019) recalled in Algorithm 10 (where the apostrophes indicate tensors whose auto-differentiation is enabled) and the fact that $c_{\Gamma^*}(x_i, y_j)$ is explicitly computable using Equation (15). The whole implementation of EGW-GRAD is given in Algorithm 11, where KERNELGW outputs both the optimal coupling $\pi$ and the optimal potentials $(f, g)$ of the last Sinkhorn step.

---

**Algorithm 9** EOT-DUALLOSS.

---

**Parameters :** $\varepsilon > 0$.
**Inputs:** $a \in \mathbb{R}^N, b \in \mathbb{R}^M, f \in \mathbb{R}^N, g \in \mathbb{R}^M, C \in \mathbb{R}^{N \times M}$.
$f \leftarrow -\varepsilon \cdot [\log \sum_j \exp(\log(b_j) + (g_j - C_{ij})/\varepsilon)]_i$
$g \leftarrow -\varepsilon \cdot [\log \sum_i \exp(\log(a_i) + (f_i - C_{ij})/\varepsilon)]_j$
$L \leftarrow \sum_i a_i f_i + \sum_j b_j g_j$
**Return** $L$

---

**Algorithm 10** EOT-GRAD.

---

**Parameters :** $\varepsilon > 0$.
**Inputs:** $a \in \mathbb{R}^N, b \in \mathbb{R}^M, x_i \in \mathcal{X}, y_j \in \mathcal{Y}$.
$f, g \leftarrow \text{SINKHORN}_\varepsilon(a, b, C = [c(x_i, y_j)]_{ij})$
$x_i', y_j' \leftarrow x_i, y_j$ with auto-diff. activated
$L' \leftarrow \text{EOT-DUALLOSS}(a, b, f, g, C' = [c(x_i', y_j')]_{ij})$
$G^X, G^Y \leftarrow$ Gradient of $L'$ w.r.t. $x_i', y_j'$
**Return** $G^X, G^Y$

---

---

**Algorithm 11** EGW-GRAD.

---

**Parameters :** $\varepsilon > 0$.
**Inputs:** $a \in \mathbb{R}^N, b \in \mathbb{R}^M, x_i \in \mathcal{X}, y_j \in \mathcal{Y}$.
$P, f, g \leftarrow$ KERNEL-GW$_\varepsilon(a, b, C^X = [c_\mathcal{X}(x_i, x_k)]_{ik}, C^Y = [c_\mathcal{Y}(y_j, y_l)]_{jl})$.
$x'_i, y'_j \leftarrow x_i, y_j$ with auto-diff. activated
$L' \leftarrow$ EOT-DUALLOSS$(a, b, f, g, C' = [-4 \cdot k_\mathcal{X}(x'_i, x'_i) k_\mathcal{Y}(y'_j, y'_j) - 16 \cdot \sum_{kl} k_\mathcal{X}(x'_i, x_k) k_\mathcal{Y}(y'_j, y_l) \pi_{kl}]_{ij})$
$L'_0 \leftarrow \sum_{ik} c_\mathcal{X}(x'_i, x'_k)^2 a_i a_k + \sum_{jl} c_\mathcal{Y}(y'_j, y'_l)^2 b_j b_l - 4 \sum_i k_\mathcal{X}(x'_i, x'_i) a_i \sum_j k_\mathcal{Y}(y'_j, y'_j) b_j$
$G^X, G^Y \leftarrow$ Gradient of $L' + L'_0$ w.r.t. $(x'_i), (y'_j)$
**Return** $G^X, G^Y$

---

**Approximation with Kernel PCA.** The previous method outputs exact gradients, but it scales quadratically in memory. To further improve computational complexity, we provide a method relying on Kernel PCA instead: the principle is to differentiate through kernel embeddings, then differentiate through CNT-GW using the same technique as before.

Although we cannot differentiate through Kernel PCA directly, we use the following strategy. Let us denote by $\Lambda_1, \ldots, \Lambda_D \in \mathbb{R}$ and $V_1, \ldots, V_D \in \mathbb{R}^N$ the eigendecomposition of the centered kernel matrix $K_X = [k_X(x_i, x_j)]_{ij}$. The *kernel projection map* $\Pi_\mathcal{X} : \mathcal{X} \mapsto \mathbb{R}^D$ sends any point $x \in \mathcal{X}$ to its kernel embedding by interpolating the embeddings of $x_1, \ldots, x_N$:

$$\Pi_\mathcal{X}(x) = \left[ \frac{1}{\sqrt{\Lambda_d}} \sum_{i=1}^N k_\mathcal{X}(x, x_i) V_{di} \right]_{d=1,\ldots,D}.$$

This map is differentiable with respect to $x$, with:

$$\nabla_x \Pi_\mathcal{X} = \frac{1}{\sqrt{\Lambda_d}} \sum_{i=1}^N \nabla_x k_\mathcal{X}(x, x_i) V_{di}.$$

We can therefore use $\nabla_{x_i} \Pi_\mathcal{X}$ as an approximation of the differentiation of the Kernel PCA with respect to $x_i$. The corresponding code is provided in Algorithm 12.

---

**Algorithm 12** EGW-GRAD (with Kernel PCA).

---

**Parameters :** $\varepsilon > 0, D > 0, E > 0$.
**Inputs:** $a \in \mathbb{R}^N, b \in \mathbb{R}^M, x_i \in \mathcal{X}, y_j \in \mathcal{Y}$.
$X, V_{1:D}^X, \Lambda_{1:D}^X \leftarrow$ KERNEL-PCA$(x_i, k_\mathcal{X}, D)$ with eigendecomposition
$Y, V_{1:E}^Y, \Lambda_{1:E}^Y \leftarrow$ KERNEL-PCA$(x_j, k_\mathcal{Y}, E)$ with eigendecomposition
$\Gamma, f, g \leftarrow$ CNT-GW$(a, b, X, Y)$
$x'_i, y'_j \leftarrow x_i, y_j$ with auto-diff. activated
$X'_i \leftarrow$ KERNEL-PROJ$(x'_i, x_i, V_{1:D}^X, \Lambda_{1:D}^X)$
$Y'_j \leftarrow$ KERNEL-PROJ$(y'_j, y_j, V_{1:E}^Y, \Lambda_{1:E}^Y)$
$L' \leftarrow$ EOT-DUALLOSS$(a, b, f, g, C' = [-4 \cdot \|X'_i\|^2 \|X'_j\|^2 - 16 \cdot \sum_k X'_{ik} \sum_l (Y'_{jl} \Gamma_{kl})]_{ij})$     with $C'$ as symbolic matrix
$L'_0 \leftarrow \sum_{ik} \|X'_i - X'_k\|^4 a_i a_k + \sum_{jl} \|Y'_j - Y'_l\|^4 b_j b_l - 4 \cdot \sum_i \|X'_i\|^2 a_i \sum_j \|Y'_j\|^2 b_j$
$G^X, G^Y \leftarrow$ Gradient of $L' + L'_0$ w.r.t. $(x'_i), (y'_j)$
**Return** $G^X, G^Y$

---

**Algorithm 13** KERNEL-PROJ.

---

**Inputs:** $x \in \mathcal{X}, x_i \in \mathcal{X}, V_{1:D} \in \mathbb{R}^{N \times D}, \Lambda_{1:D} \in \mathbb{R}^D$.
$X \leftarrow [\frac{1}{\sqrt{\Lambda_d}} \sum_i k_X(x, x_i) V_{di}]_{d=1,\ldots,D}$
**Return** $X$

---

# C. Proofs

We first recall several definitions necessary to the proofs of this paper.

**Pushforward.** Given a measure $\alpha \in \mathcal{M}(\mathcal{X})$ and a map $\Psi : X \longrightarrow Y$, the *pushforward* of $\alpha$ by $\Psi$ (dentoed $\Psi_\sharp \alpha$) is the only measure of $\mathcal{M}(Y)$ such that for all continuous function $h \in \mathcal{C}(Y)$, $\int h(y)\mathrm{d}(\Psi_\sharp \alpha)(y) = \int h(\Psi(x))\mathrm{d}\beta(x)$.

**Weak convergence of measures.** A sequence of measures $(\alpha_n) \in \mathcal{M}(\mathcal{X})$ converges weakly to $\alpha \in \mathcal{M}(\mathcal{X})$ (denoted $\alpha_n \rightharpoonup \alpha$) if for any continuous function $f \in \mathcal{C}(\mathcal{X})$, $\int f \mathrm{d}\alpha_n \to \int f \mathrm{d}\alpha$.

**Hilbert-Schmidt norm and scalar product.** Let $\Gamma \in \mathbf{H}$ be a HS operator. Its HS norm is equal to $\|\Gamma\|_{\mathrm{HS}}^2 := \sum_i \|\Gamma e_i\|_{\mathcal{H}_Y}^2$, where $(e_i)$ is an orthonormal basis of $\mathcal{H}_\mathcal{X}$ (this definition is independent of the choice of $(e_i)$). If $x, x' \in \mathcal{H}_X$ and $y, y' \in \mathcal{H}_Y$, the HS scalar product satisfies $\langle xy^\top, x'y'^\top \rangle_{\mathrm{HS}} = \langle x, x' \rangle_{\mathcal{H}_X} \langle y, y' \rangle_{\mathcal{H}_Y}$.

## C.1. Proofs of Section 2 and 3

**Proposition 2.1.** *Let $c_\mathcal{X}$ and $c_\mathcal{Y}$ be two symmetric functions with non-negative values such that:*

$$c_\mathcal{X}(x, x') = 0 \Leftrightarrow x = x' \quad \text{and} \quad c_\mathcal{X}(y, y') = 0 \Leftrightarrow y = y' \ .$$

*Then, $GW(\alpha, \beta) = 0$ if and only if $\alpha$ and $\beta$ are isometric, i.e. there exists an application $I : \mathcal{X} \longrightarrow \mathcal{Y}$ that pushes $\alpha$ onto $\beta$ such that for all $x$ and $x'$ in the support of $\alpha$:*

$$c_\mathcal{X}(x, x') = c_\mathcal{Y}(I(x), I(x')) \ .$$

*Proof.* This is result have been proven in (Mémoli, 2011, Theorem 5.1) when $c_\mathcal{X}, c_\mathcal{Y}$ are metric distances, and their proof does not use the triangular inequality. We will recall their demonstration here, showing that it generalizes to our setting.

If $\alpha$ and $\beta$ are isometric, then $GW(\alpha, \beta)$ is trivially equal to 0. Reciprocally, let $\alpha \in \mathcal{M}_1^+(\mathcal{X})$, $\alpha \in \mathcal{M}_1^+(\mathcal{Y})$ such that $GW(\alpha, \beta) = 0$. There exists a $\pi \in \mathcal{M}(\alpha, \beta)$ such that:

$$\int (c_\mathcal{X}(x, x') - c_\mathcal{Y}(y, y'))^2 \mathrm{d}\pi(x, y) \mathrm{d}\pi(x', y') = 0, \quad i.e. \quad \forall (x, y), (x', y') \in \mathrm{Supp}\,(\pi), \quad c_\mathcal{X}(x, x') = c_\mathcal{Y}(y, y'). \quad (16)$$

Let $x \in \mathcal{X}$ and $y, y' \in \mathcal{Y}$. If $(x, y), (x, y') \in \mathrm{Supp}\,(\pi)$, then Equation (16) implies that $c_\mathcal{Y}(y, y') = c_\mathcal{X}(x, x) = 0$, and $y = y'$ by hypothesis on $c_\mathcal{Y}$: therefore, for each $x \in \mathrm{Supp}\,(\alpha)$, there is a unique $y \in \mathrm{Supp}\,(\beta)$ such that $(x, y) \in \mathrm{Supp}\,(\pi)$. Reciprocally, for each $x \in \mathrm{Supp}\,(\alpha)$ there is a unique $y \in \mathrm{Supp}\,(\beta)$ such that $(x, y) \in \mathrm{Supp}\,(\pi)$. This proves the existence of a bijection $I : \mathrm{Supp}\,(\alpha) \longrightarrow \mathrm{Supp}\,(\beta)$ such that $\mathrm{Supp}\,(\pi) = \{(x, I(x)), x \in \mathcal{X}\}$.

From Equation (16), we immediately have that all $(x, I(x)), (x', I(x')) \in \mathrm{Supp}\,(\pi)$ satisfy $c_\mathcal{X}(x, x') = c_\mathcal{Y}(I(x), I(x'))$. Moreover, using the fact that $y = I(x)$ for almost every $(x, y) \in \mathrm{Supp}\,(\pi)$, we write:

$$\text{For all } h \in \mathcal{C}(\mathcal{Y}), \quad \int h(y)\mathrm{d}\beta(y) = \int h(y)\mathrm{d}\pi(x, y) = \int h(I(x))\mathrm{d}\pi(x, y) = \int h(\Psi(x))\mathrm{d}\beta(y).$$

Therefore, $\beta = I_\sharp \alpha$, i.e. $I$ pushes $\alpha$ onto $\beta$, which proves the result. $\square$

To prove the main EGW decomposition theorem, we need the following lemma:

**Lemma C.1.** *Let $\mathcal{X}$ and $\mathcal{Y}$ be Hilbert spaces and $c_\mathcal{X}, c_\mathcal{Y}$ be the corresponding squared Hilbert norms. Let $\alpha, \beta$ be compactly supported probability measures over $\mathcal{X}$ and $\mathcal{Y}$. The GW loss decomposes as:*

$$\text{For all } \pi \in \mathcal{M}(\alpha, \beta), \quad \mathcal{L}(\pi) = C(\alpha, \beta) - 8 \cdot \left\| \int xy^\top \mathrm{d}\pi(x, y) \right\|_{\mathrm{HS}}^2 - 4 \cdot \int \|x\|^2 \|y\|^2 \cdot \mathrm{d}\pi(x, y),$$

*where $C(\alpha, \beta)$ is a constant that only depends on the marginals $\alpha$ and $\beta$:*

$$C(\alpha, \beta) = \int \|x - x'\|^4 \, \mathrm{d}\alpha(x)\mathrm{d}\alpha(x') + \int \|y - y'\|^4 \, \mathrm{d}\beta(y)\mathrm{d}\beta(y') - 4 \cdot \int \|x\|^2 \, \mathrm{d}\alpha(x) \int \|y\|^2 \, \mathrm{d}\beta(y).$$

*Proof.* We follow the same computations as (Zhang et al., 2024). Let $\pi \in \mathcal{M}(\alpha, \beta)$. We first develop:

$$\mathcal{L}(\pi) = \int \|x - x'\|^4 \, d\pi(x,y)d\pi(x',y') + \int \|y - y'\|^4 \, d\pi(x,y)d\pi(x',y') - 2\int \|x - x'\|^2 \|y - y'\|^2 \, d\pi(x,y)d\pi(x',y').$$

Since $\pi \in \mathcal{M}(\alpha, \beta)$, we have $\int f(x)d\pi(x,y) = \int f \cdot \alpha$ and $\int g(y)d\pi(x,y) = \int g \cdot \beta$ for any measurable functions $f$ and $g$ on $\mathcal{H}_X$ and $\mathcal{H}_Y$. Therefore:

$$\int \|x - x'\|^4 \, d\pi(x,y)d\pi(x',y') = \int \|x - x'\|^4 \, d\alpha(x)d\alpha(x'),$$

$$\int \|y - y'\|^4 \, d\pi(x,y)d\pi(x',y') = \int \|y - y'\|^4 \, d\beta(y)d\beta(y').$$

Moreover, for any $x, x' \in \mathcal{H}_X$ and $y, y' \in \mathcal{H}_Y$, and using the equality $\langle x, x' \rangle \langle y, y' \rangle = \langle xy^\top, x'y'^\top \rangle_{\mathrm{HS}}$:

$$\|x - x'\|^2 \|y - y'\|^2 = \|x\|^2 \|y\|^2 + \|x'\|^2 \|y'\|^2 + \|x'\|^2 \|y\|^2 + \|x\|^2 \|y'\|^2 + 4 \langle xy^\top, x'y'^\top \rangle_{\mathrm{HS}}$$
$$- 2\left( \langle x, x' \rangle (\|y\|^2 + \|y'\|^2) + \langle y, y' \rangle (\|x\|^2 + \|x'\|^2) \right). \quad (17)$$

The four first terms can be simplified as follow:

$$\int \|x\|^2 \|y'\|^2 \, d\pi(x,y)d\pi(x',y') = \int \|x'\|^2 \|y\|^2 \, d\pi(x,y)d\pi(x',y') = \int \|x\|^2 \, d\alpha(x) \int \|y\|^2 \, d\beta(y),$$

$$\int \|x\|^2 \|y\|^2 \, d\pi(x,y)d\pi(x',y') = \int \|x'\|^2 \|y'\|^2 \, d\pi(x,y)d\pi(x',y') = \int \|x\|^2 \|y\|^2 \, d\pi(x,y).$$

Moreover, by bilinearity of $\langle \cdot, \cdot \rangle_{\mathrm{HS}}$:

$$\int \langle xy^\top, x'y'^\top \rangle_{\mathrm{HS}} \, d\pi(x,y)d\pi(x',y') = \left\langle \int xy^\top d\pi(x,y), \int x'y'^\top d\pi(x',y') \right\rangle_{\mathrm{HS}} = \left\| \int xy^\top d\pi(x,y) \right\|_{\mathrm{HS}}^2.$$

Note that by the Cauchy-Scharwz inequality, this (Bochner) integral is well-defined. Finally, since $\alpha$ is centered:

$$\int \langle x, x' \rangle \|y\|^2 \, d\pi(x,y)d\pi(x',y') = \int \left\langle x, \int x' d\alpha(x') \right\rangle \|y\|^2 \, d\pi(x,y) = 0.$$

Using the same computations on the other terms, we obtain:

$$\int \left( \langle x, x' \rangle (\|y\|^2 + \|y'\|^2) + \langle y, y' \rangle (\|x\|^2 + \|x'\|^2) \right) d\pi(x,y)d\pi(x',y') = 0.$$

Putting everything together in Equation (17), we finally obtain the desired formula. $\qquad \square$

**Theorem 3.2.** *Let $\alpha \in \mathcal{M}_1^+(\mathcal{X})$ and $\beta \in \mathcal{M}_1^+(\mathcal{Y})$ be two probability distributions with compact supports on topological spaces $\mathcal{X}$ and $\mathcal{Y}$, endowed with definite CNT costs $c_\mathcal{X}$ and $c_\mathcal{Y}$. Let $\Phi(x) = (\varphi(x), \frac{1}{2}\|\varphi(x)\|^2)$ and $\Psi(y) = (\psi(y), \frac{1}{2}\|\psi(y)\|^2)$ denote their respective GW-embeddings, as in Definition 3.1. Then, for any temperature $\varepsilon \geq 0$, the EGW problem of Eq. (3) is equivalent to:*

$$GW_\varepsilon(\alpha, \beta) = C(\alpha, \beta) + 8 \min_{\Gamma \in \mathbf{H}} \min_{\pi \in \mathcal{M}(\alpha, \beta)} \mathcal{F}(\Gamma, \pi),$$

*where $C(\alpha, \beta)$ is an additive constant and:*

$$\mathcal{F}(\Gamma, \pi) := \|\Gamma\|_{\mathrm{HS}}^2 + \varepsilon KL(\pi\|\alpha \otimes \beta) - 2\int \langle \overline{\Gamma}, \Phi(x)\Psi(y)^\top \rangle_{\mathrm{HS}} d\pi(x,y).$$

*Proof of Theorems 3.2 to 3.4.* This is a consequence of Lemma C.1 combined with the following equality:

$$-\left\| \int xy^\top d\pi(x,y) \right\|_{\mathrm{HS}}^2 = \min_{\Gamma \in \mathbf{H}} \left( \|\Gamma\|_{\mathrm{HS}}^2 - 2\left\langle \Gamma, \int xy^\top d\pi(x,y) \right\rangle_{\mathrm{HS}} \right),$$

since $f : \Gamma \mapsto \|\Gamma\|^2 - 2 \langle \Gamma, \int xy^\top \mathrm{d}\pi(x,y) \rangle_{\mathrm{HS}}$ is a quadratic function with minimum attained at $\Gamma = \int xy^\top \mathrm{d}\pi(x,y)$. We only need to apply this formula to the GW-embeddings $\Phi(x)$ and $\Psi(y)$ to obtain the results (Theorem 3.3 being a consequence of the convexity of $f$). Finally, Theorem 3.4 is a direct reformulation of Theorem 3.2 using the fact that:

$$\langle \overline{\Gamma}, \Phi(x)\Psi(y)^\top \rangle_{\mathrm{HS}} = \langle \Phi(x), \overline{\Gamma}(\Psi(y)) \rangle_{\mathcal{H}_\mathcal{X}} = \langle \overline{\Gamma}^\top (\Phi(x)), \Psi(y) \rangle_{\mathcal{H}_\mathcal{Y}}.$$

□

*Remark* C.3. In the proofs below, we will also use the following, equivalent form of Theorem 3.2:

$$\mathrm{GW}_\varepsilon(\alpha, \beta) = C(\varphi_\sharp \alpha, \psi_\sharp \beta) + \min_{\Gamma \in \mathbf{H}} \left( 8 \cdot \|\Gamma\|_{\mathrm{HS}}^2 + \mathrm{OT}_\varepsilon^\Gamma(\varphi_\sharp \alpha, \psi_\sharp \beta) \right), \tag{18}$$

where $\mathrm{OT}_\varepsilon^\Gamma$ is the entropic optimal transport for the cost $c_\Gamma(x,y) = -16 \langle \Gamma, xy^T \rangle_{\mathrm{HS}} - 4 \|x\|^2 \|y\|^2$. This corresponds to the form in which (Rioux et al., 2024; Zhang et al., 2024) stated the dual formula for squared Euclidean costs.

## C.2. Proofs of Section 4.1

**Theorem 4.1.** *Let $c$ be a continuous cost (not necessarily vanishing on the diagonal) on a bounded domain. Then, $c$ is CNT if and only if $\mathrm{S}_\varepsilon \geq 0$ for any $\varepsilon > 0$.*

*Proof.* The main result in (Feydy et al., 2019) is that if the cost is such that $e^{-c(x,y)/\varepsilon}$ is a positive definite kernel for all $\varepsilon$, then $\mathrm{S}_\varepsilon \geq 0$ for all $\varepsilon$. When the cost $c$ is CND, this condition is satisfied. The converse implication follows by taking the limit when $\varepsilon \to +\infty$. Indeed, we show below that $\lim_{\varepsilon \to +\infty} \mathrm{OT}_\varepsilon(\alpha, \beta) = \int_{\mathcal{X} \times \mathcal{Y}} c(x,y) \alpha \otimes \beta$ which implies that

$$\mathrm{S}_\varepsilon(\alpha, \beta) = \int_{\mathcal{X} \times \mathcal{Y}} c(x,y) \mathrm{d}\alpha(x) \mathrm{d}\beta(y) - \frac{1}{2} \left( \int_{\mathcal{X} \times \mathcal{X}} c(x,x') \mathrm{d}\alpha(x) \mathrm{d}\alpha(x') + \int_{\mathcal{Y} \times \mathcal{Y}} c(y,y) \mathrm{d}\beta(y) \otimes \mathrm{d}\beta(y') \right). \tag{19}$$

The right-hand side is the maximum mean discrepancy between the two measures $\alpha, \beta$ with respect to the cost $-\frac{1}{2} c(x,y)$. Since this quantity is nonnegative for all differences of probability measures, it implies that the cost $-\frac{1}{2} c(x,y)$ is a conditionally positive kernel.

To show that the entropic transport cost limit when $\varepsilon \to +\infty$, we remark that, by testing the corresponding variational problem with $\pi(x,y) = \alpha(x) \otimes \beta(y)$, the KL term vanishes and therefore

$$\mathrm{OT}_\varepsilon(\alpha, \beta) \leq \int_{\mathcal{X} \times \mathcal{Y}} c(x,y) \mathrm{d}\alpha(x) \mathrm{d}\beta(y). \tag{20}$$

Since the cost is nonnegative (and thus the transport cost), we have

$$\mathrm{KL}\left(\pi \| \alpha \otimes \beta\right) \leq \frac{1}{\varepsilon} \int_{\mathcal{X} \times \mathcal{Y}} c(x,y) \mathrm{d}\alpha(x) \mathrm{d}\beta(y). \tag{21}$$

By Pinsker's inequality, this implies the convergence of $\pi_\varepsilon(x,y)$ to $\alpha \otimes \beta$ w.r.t. the TV norm. Consequently, when $\varepsilon \to +\infty$, we have $\int_{\mathcal{X} \times \mathcal{Y}} c(x,y) \mathrm{d}\alpha(x) \mathrm{d}\beta(y) + o(1) \leq \mathrm{OT}_\varepsilon(\alpha, \beta)$. □

**Lemma C.4.** *When the base costs are CNT, the operator $\mathrm{GW}_\varepsilon : \mathcal{M}_1^+(\mathcal{X}) \times \mathcal{M}_1^+(\mathcal{Y}) \longrightarrow \mathbb{R}$ is continuous for the weak convergence of measures.*

*Proof.* Since the embeddings provided by Theorem 2.2 are continuous, it is sufficient to prove the result when $\mathcal{X}$ and $\mathcal{Y}$ are compact subsets of two Hilbert spaces and when $c_\mathcal{X}, c_\mathcal{Y}$ are the corresponding squared Hilbert norms. We will proceed it by showing both the upper and lower semi-continuity of $\mathrm{GW}_\varepsilon$ for the weak convergence of measures.

For every $\Gamma \in \mathbf{H}$, the function $x, y \longmapsto c_\Gamma(x,y)$ is Lipschitz on $\mathcal{X} \times \mathcal{Y}$. Therefore, the function $\alpha, \beta \mapsto \mathrm{OT}_\varepsilon^\Gamma(\alpha, \beta)$ is continuous for the weak convergence (Feydy et al., 2019, Proposition 2). Since $GW_\varepsilon(\alpha, \beta)$ is the infimum of continuous functions, it is upper semi-continuous for the weak convergence.

Conversely, let $\alpha_n \rightharpoonup \alpha \in \mathcal{M}_1^+(\mathcal{X})$ and $\beta_n \rightharpoonup \beta \in \mathcal{M}_1^+(\mathcal{Y})$. Let $\pi_n \in \mathcal{M}(\alpha_n, \beta_n)$ be a sequence of optimal plans for $GW_\varepsilon(\alpha_n, \beta_n)$. By compacity of $\mathcal{M}(\alpha_n, \beta_n)$ for the weak convergence, there is an extraction $(\alpha_m, \beta_m)$ of $(\alpha_n, \beta_n)$ such that $\pi_m \rightharpoonup \pi \in \mathcal{M}(\alpha, \beta)$, and $\pi_m \otimes \pi_m \rightharpoonup \pi \otimes \pi$. By continuity of $x, x', y, y' \mapsto (\|x - x'\|^2 - \|y - y'\|^2)^2$:

$$\lim_{m \to +\infty} \int \left( \|x - x'\|^2 - \|y - y'\|^2 \right)^2 \mathrm{d}\pi_m(x, y)\mathrm{d}\pi_m(x', y') = \int \left( \|x - x'\|^2 - \|y - y'\|^2 \right)^2 \mathrm{d}\pi(x, y)\mathrm{d}\pi(x', y').$$

By lower semi-continuity of $\mu, \nu \to \mathrm{KL}\,(\mu \| \nu)$, we also have $\liminf \mathrm{KL}\,(\pi_m \| \alpha_m \otimes \beta_m) \geq \mathrm{KL}\,(\pi \| \alpha \otimes \beta)$, so:

$$\liminf \mathrm{GW}_\varepsilon(\alpha_m, \beta_m) \geq \int \left( \|x - x'\|^2 - \|y - y'\|^2 \right)^2 \mathrm{d}\pi(x, y)\mathrm{d}\pi(x', y') + \varepsilon \mathrm{KL}\,(\pi \| \alpha \otimes \beta).$$

By definition of EGW, the right term is always upper-bounded by $\mathrm{GW}_\varepsilon(\alpha, \beta)$. Hence, any sequence of $\mathrm{GW}_\varepsilon(\alpha_n, \beta_n)$ contain a subsequence whose limit inferior is larger than $GW_\varepsilon(\alpha, \beta)$, which proves $\liminf \mathrm{GW}_\varepsilon(\alpha_n, \beta_n) \geq \mathrm{GW}_\varepsilon(\alpha, \beta)$ and the lower-semi continuity of $\mathrm{GW}_\varepsilon$. $\qquad\square$

**Theorem 4.2.** *If $c_\mathcal{X}$ and $c_\mathcal{Y}$ are CNT costs, then $\mathrm{SGW}_\varepsilon(\alpha, \beta) \geq 0$ for any distributions $\alpha$, $\beta$ and temperature $\varepsilon > 0$. If $\alpha$ and $\beta$ are isometric, then $\mathrm{SGW}_\varepsilon(\alpha, \beta) = 0$.*

*Proof.* From (Van Gaans, 2003, Proposition 4.4), we know that the set of finitely supported probability measures over $\mathcal{X}$ (resp. $\mathcal{Y}$) is dense in $\mathcal{M}_1^+(\mathcal{X})$ (resp. $\mathcal{M}_1^+(\mathcal{Y})$). Therefore, we can restrict to the case where $\alpha$ and $\beta$ have finite support, and conclude on the general case using the continuity of EGW stated in Lemma C.4. Since finite subsets of $\mathcal{H}_\mathcal{X}$ and $\mathcal{H}_\mathcal{Y}$ belong to finite-dimensional subspaces of $\mathcal{H}_\mathcal{X}$ and $\mathcal{H}_\mathcal{Y}$, we can further restrict ourselves to measures lying on finite-dimensional euclidean spaces: from now, we set $\alpha \in \mathcal{M}_1^+(\mathbb{R}^\mathrm{D})$ and $\beta \in \mathcal{M}_1^+(\mathbb{R}^\mathrm{E})$ as two centered measures for the squared euclidean norm, with $c_\mathcal{X}, c_\mathcal{Y}$ the squared Euclidean norms of $\mathbb{R}^\mathrm{D}$ and $\mathbb{R}^\mathrm{E}$.

In finite dimension, Hilbert-Schmidt operators correspond to matrices of $\mathbb{R}^{\mathrm{D} \times \mathrm{E}}$: therefore, each $\Gamma \in \mathbf{H}$ admits a singular value decomposition (SVD) of the form $\Gamma = U^T \Sigma V$, where $\Sigma \in \mathcal{D}_\mathrm{R}^+(\mathbb{R})$ is diagonal with positive coefficients and $U \in \mathcal{O}_{\mathrm{D},\mathrm{R}}(\mathbb{R}), V \in \mathcal{O}_{\mathrm{E},\mathrm{R}}(\mathbb{R})$ are orthogonal matrices (with $\mathrm{R} \geq \max(\mathrm{D}, \mathrm{E})$). Equation (18) rewrites as:

$$GW_\varepsilon(\alpha, \beta) = C(\alpha, \beta) + \min_{\Sigma \in \mathcal{D}_\mathrm{R}^+(\mathbb{R})} \left[ 8 \cdot \|\Sigma\|_F^2 + \min_{U \in \mathcal{O}_{\mathrm{D},\mathrm{R}}(\mathbb{R}), V \in \mathcal{O}_{\mathrm{E},\mathrm{R}}(\mathbb{R})} \mathrm{OT}_\varepsilon^\Sigma(U_\sharp \alpha, V_\sharp \beta) \right], \qquad (22)$$

where $\mathrm{OT}_\varepsilon^\Sigma(\alpha, \beta)$ is the EOT for the cost $c_\Sigma(x, y) = -4 \|x\|^2 \|y\|^2 - 16 x^T \Sigma y$. Denoting by $(\Sigma^*, U^*, V^*)$ the optimal matrices in Equation (22) (and using the fact that $(\Sigma^*, U^*, U^*)$ and $(\Sigma^*, V^*, V^*)$ are in the feasible set of $\mathrm{GW}_\varepsilon(\alpha, \alpha)$ and $\mathrm{GW}_\varepsilon(\beta, \beta)$ for the new formulation of Equation (22)) we obtain:

$$\begin{aligned}
\mathrm{GW}_\varepsilon(\alpha, \beta) &= C(\alpha, \beta) + 8 \cdot \|\Sigma^*\|_F^2 + \mathrm{OT}_\varepsilon^{\Sigma^*}(U_\sharp^* \alpha, V_\sharp^* \beta), \\
\mathrm{GW}_\varepsilon(\alpha, \alpha) &\leq C(\alpha, \alpha) + 8 \cdot \|\Sigma^*\|_F^2 + \mathrm{OT}_\varepsilon^{\Sigma^*}(U_\sharp^* \alpha, U_\sharp^* \alpha), \\
\mathrm{GW}_\varepsilon(\beta, \beta) &\leq C(\beta, \beta) + 8 \cdot \|\Sigma^*\|_F^2 + \mathrm{OT}_\varepsilon^{\Sigma^*}(V_\sharp^* \beta, V_\sharp^* \beta).
\end{aligned} \qquad (23)$$

Writing $\mathrm{S}_\varepsilon^{\Sigma^*}(U_\sharp^* \alpha, V_\sharp^* \beta) = \mathrm{OT}_\varepsilon^{\Sigma^*}(U_\sharp^* \alpha, V_\sharp^* \beta) - \frac{1}{2}(\mathrm{OT}_\varepsilon^{\Sigma^*}(U_\sharp^* \alpha, U_\sharp^* \alpha) + \mathrm{OT}_\varepsilon^{\Sigma^*}(V_\sharp^* \beta, V_\sharp^* \beta))$, this implies:

$$\mathrm{SGW}_\varepsilon(\alpha, \beta) \geq C(\alpha, \beta) - \frac{1}{2}(C(\alpha, \alpha) + C(\beta, \beta)) + \mathrm{S}_\varepsilon^{\Sigma^*}(U_\sharp^* \alpha, V_\sharp^* \beta).$$

The positivity of the constant $C(\alpha, \beta) - \frac{1}{2}(C(\alpha, \alpha) + C(\beta, \beta))$ comes from the following identity:

$$\begin{aligned}
C(\alpha, \beta) - \frac{1}{2}(C(\alpha, \alpha) + C(\beta, \beta)) &= 2 \left( \int \|x\|^2 \mathrm{d}\alpha(x) \right)^2 + 2 \left( \int \|y\|^2 \mathrm{d}\beta(y) \right)^2 - 4 \int \|x\|^2 \mathrm{d}\alpha(x) \int \|y\|^2 \mathrm{d}\beta(y) \\
&= 2 \left( \|x\|^2 \mathrm{d}\alpha(x) - \int \|y\|^2 \mathrm{d}\beta(y) \right)^2 \geq 0.
\end{aligned}$$

Finally, that $\mathrm{S}_\varepsilon^{\Sigma^*}(U_\sharp^* \alpha, V_\sharp^* \beta)$ is positive by recalling that $-c_{\Sigma^*}$ is a scalar product of GW-embeddings:

$$-c_{\Sigma^*}(x, y) = 4 \|x\|^2 \|y\|^2 + 16 x^T \Sigma^* y = \left\langle (2 \|x\|^2, 4\sqrt{\Sigma^*}x), (2 \|y\|^2, 4\sqrt{\Sigma^*}y) \right\rangle.$$

which implies the positivity of the kernel $k_\varepsilon(x, y) = \exp(-c_{\Sigma^*}(x, y)/\varepsilon)$. We are in the setting of (Feydy et al., 2019, Theorem 1), which proves the positivity of $\mathrm{S}_\varepsilon^{\Sigma^*}(U_\sharp^* \alpha, V_\sharp^* \beta)$. The positivity of $\mathrm{SGW}_\varepsilon(\alpha, \beta)$ follows. $\qquad\square$

*Remark* C.5. The hypotheses of (Feydy et al., 2019) are not completely satisfied, since the kernel $k_\varepsilon$ is not necessarily universal. However, the universality of $k_\varepsilon$ is only used to prove the definiteness of the Sinkhorn divergence, and the positivity of $k_\varepsilon$ is sufficient to show the positivity of $S_\varepsilon^{\Sigma^*}(U_\sharp^*\alpha, V_\sharp^*\beta)$. It also shows that the lack of definiteness of EGW (evidenced in Proposition C.6) is a consequence of the non-universality of $k_\varepsilon$, which happens when the embedding $x \mapsto (2\|x\|^2, 4\sqrt{\Sigma^*}x)$ is non-injective. In particular, if $\Sigma^*$ is full-rank, then the kernel $e^{((4\|x\|^2\|y\|^2+16x^\top\Sigma^*y)/\varepsilon}$ is universal on compact domains (since polynomial functions are contained in the corresponding RKHS), and thus $S_\varepsilon^{\Sigma^*}$ is definite: $S_\varepsilon^{\Sigma^*}(U_\sharp^*\alpha, V_\sharp^*\beta) = 0$ if and only if $U_\sharp^*\alpha = V_\sharp^*\beta$, and the latter implies that $\alpha$ and $\beta$ are isometric since $U^*$ and $V^*$ are orthogonal matrices.

**Proposition C.6.** *Let $\alpha$ and $\beta$ be uniform measures on the spheres $\mathbb{S}^D$ and $\mathbb{S}^E$ of dimensions $D \neq E$, while $c_{\mathcal{X}}$ and $c_{\mathcal{Y}}$ are the squared norms of $\mathbb{R}^D$ and $\mathbb{R}^E$, respectively. Then, $\alpha$ and $\beta$ are not isometric but there exists an $\varepsilon_0 > 0$ such that $\mathrm{SGW}_\varepsilon(\alpha, \beta) = 0$ for every $\varepsilon > \varepsilon_0$.*

*Proof.* The function $\pi \mapsto \mathrm{KL}(\pi\|\alpha \otimes \beta)$ is strictly convex: therefore, when $\varepsilon$ is sufficiently large, the function $\mathcal{L}_\varepsilon : \pi \mapsto \mathcal{L}(\pi) + \varepsilon\mathrm{KL}(\pi\|\alpha \otimes \beta)$ is strictly convex as well, and $\mathcal{L}_\varepsilon$ admits a unique minimum on $\mathcal{M}(\alpha, \beta)$. By choice of $c_{\mathcal{X}}$ and $c_{\mathcal{Y}}$, $\mathcal{L}_\varepsilon(\pi)$ is invariant by rotation of the marginals of $\pi$: its unique minimizer must be radially symmetric, and the only coupling satisfying this constraint is $\alpha \otimes \beta$.

Similarly, when $\varepsilon$ is sufficiently large, the optimal plans of $GW_\varepsilon(\alpha, \alpha)$ and $GW_\varepsilon(\beta, \beta)$ are $\alpha \otimes \alpha$ and $\beta \otimes \beta$. Using these optimal plans, the value of $GW_\varepsilon(\alpha, \beta) - \frac{1}{2}(GW_\varepsilon(\alpha, \alpha) + GW_\varepsilon(\beta, \beta))$ can be explicitly determined, and a direct computation shows that $\mathrm{SGW}_\varepsilon(\alpha, \beta) = 0$. $\qquad\square$

**Proposition 4.3.** *Let $\varepsilon > 0$, $\alpha \in \mathcal{M}_1^+(\mathbb{R}^D)$, $\beta \in \mathcal{M}_1^+(\mathbb{R}^E)$ with supports of diameter $R, R'$, and $c_{\mathcal{X}}, c_{\mathcal{Y}}$ the squared norm of $\mathbb{R}^D$ and $\mathbb{R}^E$. Let $\lambda_\alpha, \lambda_\beta$ be the smallest eigenvalues of $\Sigma_\alpha$ and $\Sigma_\beta$. There are constants $C(D, R)$ and $C(E, R')$ such that, if:*

$$\varepsilon \max(1, \log(1/\varepsilon)) \leq \max\left(C(D, R)\cdot\lambda_\alpha^2, \ C(E, R')\cdot\lambda_\beta^2\right),$$

*then $\alpha$ and $\beta$ are isometric if and only if $\mathrm{SGW}_\varepsilon(\alpha, \beta) = 0$.*

*Proof.* To prove the result, we proceed by contraposition. Let $\alpha \in \mathcal{M}_1^+(\mathbb{R}^D)$, and let assume the existence of $\beta \in \mathcal{M}_1^+(\mathbb{R}^E)$ non isometric to $\alpha$ such that $\mathrm{SGW}_\varepsilon(\alpha, \beta) = 0$. We will show that it implies an inequality involving $\varepsilon$ and the smallest eigenvalue of $\Sigma_\alpha = \int xx^T \mathrm{d}\alpha$.

For the nullity of SGW to hold, the inequalities of Equation (23) must be equalities: if $\Gamma^*$ is the optimal dual matrix for $GW_\varepsilon(\alpha, \beta)$, denoting by $\Gamma^* = U^{*T}\Sigma^*V^*$ its singular value decomposition, we must have:

$$GW_\varepsilon(\alpha, \alpha) = C(\alpha, \alpha) + 8 \cdot \|\Sigma^*\|_F^2 + \mathrm{OT}_\varepsilon^{\Sigma^*}(U_\sharp^*\alpha, U_\sharp^*\alpha),$$

and $\Gamma_\alpha^* = U^{*T}\Sigma^*U^*$ must be an optimal dual matrix for $GW_\varepsilon(\alpha, \alpha)$.

From Theorem 4.2, $\mathrm{SGW}_\varepsilon(\alpha, \beta) = 0$ also implies $S_\varepsilon^{\Sigma^*}(U_\sharp^*\alpha, V_\sharp^*\beta) = 0$. As explained in Remark C.5, if $\Sigma^*$ is a positive definite matrix, then the nullity of $\mathrm{SGW}_\varepsilon$ would mean the isometry of $\alpha$ and $\beta$. Therefore, $\Sigma^*$ cannot be definite: the matrix $\Sigma^*$ must have a null eigenvalue, and the matrix $\Gamma_\alpha^*$ has at least one null eigenvalue as well.

Meanwhile, (Zhang et al., 2024, Proposition 1) provides an estimate of the difference between the unregularized and entropic GW values, giving a bound of the form:

$$GW_\varepsilon(\alpha, \alpha) = |GW_\varepsilon(\alpha, \alpha) - GW(\alpha, \alpha)| \leq \widetilde{C}(D, R)\cdot\varepsilon\max(1, \log(1/\varepsilon)).$$

In the original result of (Zhang et al., 2024), the constant depends on the moments of $\alpha$ rather than its radius; however, since $\alpha$ is bounded, we can bound its moments by functions of $R$ and simply make the constant depend on $D$ and $R$.

Let $\pi^*$ denote the optimal plan of $GW_\varepsilon(\alpha, \alpha)$ associated to the auto-correlation matrix $\Gamma_\alpha^*$. Using Lemma C.1, we get:

$$GW_\varepsilon(\alpha, \alpha) = C(\alpha, \alpha) - 8 \cdot \left\|\int xx'^T\mathrm{d}\pi^*\right\|_F^2 - 4 \cdot \int \|x\|^2\|x'\|^2\mathrm{d}\pi^* \leq \widetilde{C}(D, R)\cdot\varepsilon\max(1, \log(1/\varepsilon)).$$

A direct computation shows that $C(\alpha, \alpha) = 8 \left\| \int xx^T \mathrm{d}\alpha \right\|_F^2 + 4 \int \|x\|^4 \mathrm{d}\alpha$, and:

$$8 \cdot \left( \left\| \int xx^T \mathrm{d}\alpha \right\|_F^2 - \left\| \int xx'^T \mathrm{d}\pi^* \right\|_F^2 \right) + 4 \cdot \left( \int \|x\|^4 \mathrm{d}\alpha - \int \|x\|^2 \|x'\|^2 \mathrm{d}\pi^* \right) \leq \widetilde{C}(\mathrm{D}, R) \cdot \varepsilon \max(1, \log(1/\varepsilon)).$$

By the Cauchy-Schwartz inequality, $\int \|x\|^2 \|x'\|^2 \mathrm{d}\pi^* \leq \int \|x\|^4 \mathrm{d}\alpha$, and we finally obtain:

$$8 \cdot \left( \|\Sigma_\alpha\|_F^2 - \|\Gamma_\alpha^*\|_F^2 \right) \leq \widetilde{C}(\mathrm{D}, R) \cdot \varepsilon \max(1, \log(1/\varepsilon)). \tag{24}$$

Moreover, $\Sigma_\alpha$ and $\Gamma_\alpha^*$ are symmetric semi-definite matrices. Therefore, given $(e_i)$ an orthonormal eigenvector basis of $\Gamma_\alpha^*$ (by increasing eigenvalue), we have:

$$\|\Gamma_\alpha^*\|_F^2 = \sum_{i=1}^{\mathrm{D}} (e_i^T \Gamma_\alpha^* e_i)^2 \quad \text{and} \quad \|\Sigma_\alpha\|_F^2 \geq \sum_{i=1}^{\mathrm{D}} (e_i^T \Sigma_\alpha e_i)^2.$$

From the rearranging inequality, we also have:

$$\text{For any } i \geq 1, \quad e_i^T \Gamma_\alpha^* e_i = \int (e_i^T x)(e_i^T x') \mathrm{d}\pi \leq \int (e_i^T x)(e_i^T x) \mathrm{d}\alpha = e_i^T \Sigma_\alpha e_i.$$

Finally, since $\Gamma_\alpha^*$ admits at least one null eigenvalue, we have $e_1^T \Gamma_\alpha^* e_1 = 0$, and $e_1^T \Sigma_\alpha e_1 \geq \lambda_\alpha$ where $\lambda_\alpha$ is the smallest eigenvalue of $\Sigma_\alpha$. Putting everything together, we obtain the following inequality:

$$\|\Gamma_\alpha^*\|_F^2 = \sum_{i=2}^{\mathrm{D}} (e_i^T \Gamma_\alpha^* e_i)^2 \leq \sum_{i=2}^{\mathrm{D}} (e_i^T \Sigma_\alpha e_i)^2 \leq \|\Sigma_\alpha\|_F^2 - (e_1^T \Sigma_\alpha e_1)^2 \leq \|\Sigma_\alpha\|_F^2 - \lambda_\alpha^2.$$

Plugging this result in Equation (24), we finally prove that $\varepsilon \max(1, \log(1/\varepsilon)) \geq C(\mathrm{D}, R)\lambda_\alpha^2$ (with $C(\mathrm{D}, R) = 8/\widetilde{C}(\mathrm{D}, R)$). Applying the same proof on $\beta$ yields $\varepsilon \max(1, \log(1/\varepsilon)) \geq C(\mathrm{E}, R')\lambda_\beta^2$, proving the contraposition of Proposition 4.3. $\square$

### C.3. Proofs of Section 4.2

**Proposition 4.4** (Primal descent). *Couplings $(\pi_t)$ obtained by alternate minimization from Eq. (5) also satisfy Eq. (6), with a monotonic decrease of the EGW loss at every step.*

*Proof.* To simplify the notations, we identify the base points $x \in \mathcal{X}$ and $y \in \mathcal{Y}$ with their GW-embeddings $\phi(x)$ and $\psi(y)$.

The gradient of $\mathcal{L}$ at any $\tilde{\pi} \in \mathcal{M}(\mathcal{X} \times \mathcal{Y})$ is $\nabla_{\tilde{\pi}} \mathcal{L} : x, y \mapsto 2 \cdot \int (c_\mathcal{X}(x, x') - c_\mathcal{Y}(y, y'))^2 \mathrm{d}\tilde{\pi}(x', y')$. Therefore:

$$\text{For all } \pi, \tilde{\pi} \in \mathcal{M}(X \times Y), \quad \int \nabla_{\tilde{\pi}} \mathcal{L} \cdot \mathrm{d}\pi = 2 \cdot \int (c_X(x, x') - c_Y(y, y'))^2 \mathrm{d}\pi(x, y)\mathrm{d}\tilde{\pi}(x', y').$$

When both $\pi$ and $\tilde{\pi}$ belong to the feasible set $\mathcal{M}(\alpha, \beta)$, the computations of Lemma C.1 can be adapted to show that:

$$\int \nabla_{\tilde{\pi}} \mathcal{L} \cdot \mathrm{d}\pi = 2 \cdot C(\alpha, \beta) - 16 \cdot \left\langle \int xy^\top \mathrm{d}\pi, \int x'y'^\top \mathrm{d}\tilde{\pi} \right\rangle_{\mathrm{HS}} - 4 \int \|x\|^2 \|y\|^2 \cdot \mathrm{d}\pi(x, y) - 4 \int \|x\|^2 \|y\|^2 \cdot \mathrm{d}\tilde{\pi}(x, y),$$

where the last term is independent of $\pi$. Therefore, for any $t > 0$, since $\Gamma_t = \int xy^\top \mathrm{d}\pi_t$ and $\pi_{t+1} = \operatorname{argmin}_{\pi \in \mathcal{M}(\alpha, \beta)} \mathrm{OT}_\varepsilon^{\Gamma_t}(\alpha, \beta)$ (with $\mathrm{OT}_\varepsilon^{\Gamma_t}$ defined in Remark C.3), we have:

$$\begin{aligned} \pi_{t+1} &= \operatorname*{argmin}_{\pi \in \mathcal{M}(\alpha, \beta)} \left( -16 \cdot \left\langle \int xy^\top \mathrm{d}\pi, \Gamma_t \right\rangle_{\mathrm{HS}} - 4 \int \|x\|^2 \|y\|^2 \cdot \mathrm{d}\pi(x, y) + \varepsilon \mathrm{KL}(\pi \| \alpha \otimes \beta) \right) \\ &= \operatorname*{argmin}_{\pi \in \mathcal{M}(\alpha, \beta)} \left( \int \nabla_{\pi_t} \mathcal{L} \cdot \mathrm{d}\pi + \varepsilon \mathrm{KL}(\pi \| \alpha \otimes \beta) \right), \end{aligned}$$

which proves the desired equivalence. $\square$

**Proposition 4.5** (Dual descent). *Operators $\Gamma_t \in \mathbf{H}$ obtained by alternate minimization from Eq. (5) also satisfy:*

$$\Gamma_{t+1} = \Gamma_t - (1/2) \cdot \nabla U_\varepsilon(\Gamma_t)$$
$$\text{where} \quad U_\varepsilon : \Gamma \mapsto \min_{\pi \in \mathcal{M}(\alpha,\beta)} \mathcal{F}(\Gamma, \pi) \tag{7}$$

*derives from the objective $\mathcal{F}$ defined in Theorem 3.2.*

*Proof.* We first explicit the dual function:

$$\text{For all } \Gamma \in \mathbf{H}, \quad U_\varepsilon(\Gamma) = \|\Gamma\|_{\mathrm{HS}}^2 + (1/8) \cdot \mathrm{OT}_\varepsilon^\Gamma(\alpha, \beta).$$

We use (Rioux et al., 2024, Proposition 2) which proves the smoothness of $U_\varepsilon$ and a formula for $\nabla U_\varepsilon(\Gamma)$ in the finite dimensional case. Their proof remains valid in the Hilbert setting[3]; $\nabla U_\varepsilon(\Gamma)$ exists, is continuous, and satisfies:

$$\text{For all } \Gamma \in \mathbf{H}, \quad \nabla U_\varepsilon(\Gamma) = 2 \cdot \Gamma - 2 \cdot \int xy^\top \mathrm{d}\pi_\Gamma \quad \text{where} \quad \pi_\Gamma = \underset{\pi}{\operatorname{argmin}}\, \mathrm{OT}_\varepsilon^\Gamma(\alpha, \beta).$$

As a consequence, since $\pi_{t+1} = \operatorname{argmin}_\pi \mathrm{OT}_\varepsilon^{\Gamma_t}(\alpha, \beta)$ and $\Gamma_{t+1} = \int xy^\top \mathrm{d}\pi_{t+1}$ for all $t > 0$, the previous equation becomes:

$$\nabla U_\varepsilon(\Gamma_t) = 2 \cdot \Gamma_t - 2 \cdot \Gamma_{t+1},$$

which is equivalent to the desired equation. $\square$

**Theorem 4.6.** *Let $(\pi_t, \Gamma_t)$ be the alternate sequence of Eq. (5). Then, $\|\Gamma_t - \Gamma_{t+1}\|_{\mathrm{HS}} \to 0$ and every subsequential limit of $(\Gamma_t)$ is a critical point of $U_\varepsilon$, with:*

$$\sum_{k=0}^{t-1} \|\Gamma_k - \Gamma_{k+1}\|_{\mathrm{HS}}^2 \leq U_\varepsilon(\Gamma_0) - U_\varepsilon(\Gamma_t). \tag{8}$$

*Proof.* We first write:

$$U_\varepsilon(\Gamma) = \min_{\pi \in \mathcal{M}(\alpha,\beta)} f_{\varepsilon,\pi}(\Gamma), \quad \text{where} \quad f_{\varepsilon,\pi} : \Gamma \longmapsto \|\Gamma\|_{\mathrm{HS}}^2 - 2\left\langle \Gamma, \int xy^\top \mathrm{d}\pi \right\rangle_{\mathrm{HS}} + C(\pi),$$

with $C(\pi)$ a constant independent of $\Gamma$. Moreover, $U_\varepsilon(\Gamma_t) = f_{\varepsilon,\pi_{t+1}}(\Gamma_t)$ and $\Gamma_{t+1} = \int xy^\top \mathrm{d}\pi_{t+1}$. Therefore:

$$\begin{aligned}
U_\varepsilon(\Gamma_t) - U_\varepsilon(\Gamma_{t+1}) &\geq f_{\varepsilon,\pi_{t+1}}(\Gamma_t) - f_{\varepsilon,\pi_{t+1}}(\Gamma_{t+1}) \\
&\geq (\|\Gamma_t\|_{\mathrm{HS}}^2 - \|\Gamma_{t+1}\|_{\mathrm{HS}}^2) - 2(\langle \Gamma_t, \Gamma_{t+1} \rangle_{\mathrm{HS}} - \langle \Gamma_{t+1}, \Gamma_{t+1} \rangle_{\mathrm{HS}}) \\
&\geq \|\Gamma_{t+1}\|_{\mathrm{HS}}^2 + \|\Gamma_t\|_{\mathrm{HS}}^2 - 2\langle \Gamma_t, \Gamma_{t+1} \rangle_{\mathrm{HS}} \\
&\geq \|\Gamma_t - \Gamma_{t+1}\|_{\mathrm{HS}}^2.
\end{aligned}$$

By summation, we obtain:

$$\sum_{k=0}^{t-1} \|\Gamma_k - \Gamma_{k+1}\|_{\mathrm{HS}}^2 \leq U_\varepsilon(\Gamma_0) - U_\varepsilon(\Gamma_t). \tag{25}$$

Since the right term converges to a finite value, we must have $\|\Gamma_k - \Gamma_{k+1}\|_{\mathrm{HS}} \to 0$. Since $\Gamma_{t+1} - \Gamma_t = \frac{1}{2}\nabla U_\varepsilon(\Gamma_t)$, we also have $\nabla U_\varepsilon(\Gamma_t) \to 0$: by continuity of $\nabla U_\varepsilon(\Gamma_t)$, all sublimits of $(\Gamma_t)$ are critical points of $U_\varepsilon$. $\square$

*Remark C.7.* The limit $\|\Gamma_k - \Gamma_{k+1}\|_{\mathrm{HS}} \to 0$ implies that the set of subsequential limits of $(\Gamma_t)$ is connected: in particular, if the set of critical points of $U_\varepsilon$ is discrete, then the sequence $(\Gamma_t)$ admits a unique limit. Moreover, since $\Gamma_{t+1} - \Gamma_t = \frac{1}{2}\nabla U_\varepsilon(\Gamma_t)$, we also have a bound on the gradient norms:

$$\sum_{k=0}^{t-1} \|\nabla U_\varepsilon(\Gamma_k)\|_{\mathrm{HS}}^2 \leq \frac{1}{2} \cdot (U_\varepsilon(\Gamma_0) - U_\varepsilon(\Gamma_t)).$$

---

[3]Indeed, the first term is quadratic, therefore smooth. The proof in (Rioux et al., 2024, Proposition 2) uses the implicit function theorem on $\Gamma$, which applies in this infinite dimensional setting of Hilbert space of Hilbert-Schmidt operators.

# D. Additional Experiments

## D.1. Detailed Analysis of Solver Convergence

We now explore convergence properties of EGW solvers. In particular, we thoroughly investigate the differences between classical and kernel-based implementations by focusing on ENTROPIC-GW (Peyré et al., 2016) and KERNEL-GW (Algorithm 5), which are both exact EGW solvers. We mainly show that convergence speed is significantly increased by using symmetrized Sinkhorn (Algorithm 3) rather than standard Sinkhorn in EGW solvers. Symmetrized Sinkhorn with ENTROPIC-GW also introduce instabilities that are greatly attenuated with KERNEL-GW. Finally, adaptive Sinkhorn (Algorithm 6) allows symmetrized KERNEL-GW to converge without requiring to tune $n_{\text{inner}}$ manually, whereas other variants does not benefit from adaptive scheduling. Therefore, combining both symmetrization and kernelization is the best option to fasten convergence while keeping stable optimization, explaining the speed-up of CNT-GW over QUADRATIC-LOWRANK-GW.

### D.1.1. VISUALIZATION OF CONVERGENCE PROPERTIES

**Case of Non-Convergence for Non-CNT Costs.** In Figure 7, we illustrate a counter-example to the convergence of ENTROPIC-GW For the cost function $c_X, c_Y = \|\cdot\|^6$ (which is not conditionally of negative type), we identify a configuration where the algorithm oscillates indefinitely between two distinct states, which we guarantee to never happen with CNT costs.

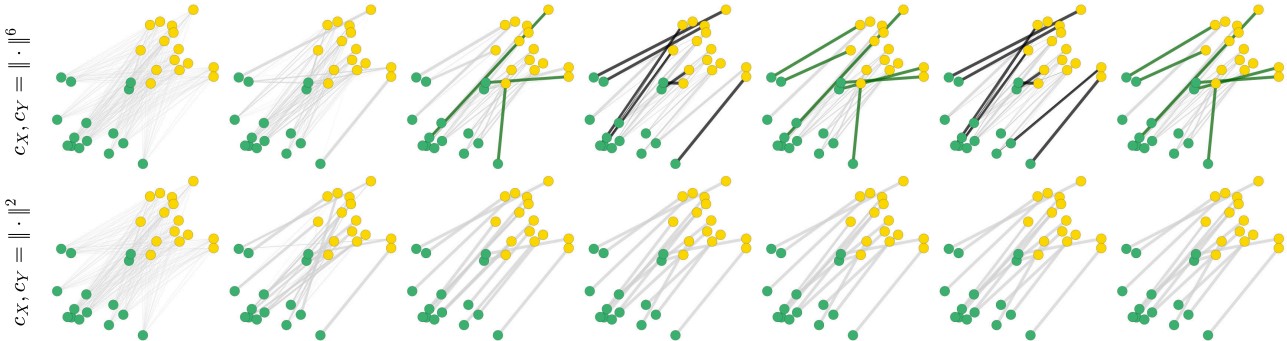

*Figure 7.* Successive transport plans obtained by ENTROPIC-GW. For non CNT costs (e.g., $c_X, c_Y = \|\cdot\|^6$), the algorithm may oscillate indefinitely between distinct transport plans. We prove that this cannot occur when the costs are CNT (e.g., $c_X, c_Y = \|\cdot\|^2$).

**Convergence Trajectories.** We visualize the impact of Sinkhorn symmetrization and the transition from ENTROPICGW to KERNEL-GW by plotting optimization trajectories in the dual space **H**. We take squared norm costs and measures in $\mathbb{R}$ and $\mathbb{R}^2$ so that $\mathbf{H} = \mathbb{R}^{1 \times 2}$ can be displayed in 2D, and we represent the optimization trajectories for different numbers of Sinkhorn iterations $n_{\text{inner}}$. As shown in Figure 8, standard Sinkhorn does not follow the true dual gradient: solver steps are systematically oriented towards the origin, indicating a bias towards the trivial plan $\alpha \otimes \beta$. Symmetrizing Sinkhorn solves this problem, and the solver follows a straight trajectory towards the optimum. When further reducing $n_{\text{inner}}$, an additional bias appears: the solver does not converge to the EGW minimum but stabilizes around a value biased towards the origin. KERNEL-GW corrects this effect: when $n_{\text{inner}}$ is small, Sinkhorn steps remain noisy but gravitate around the true optimum.

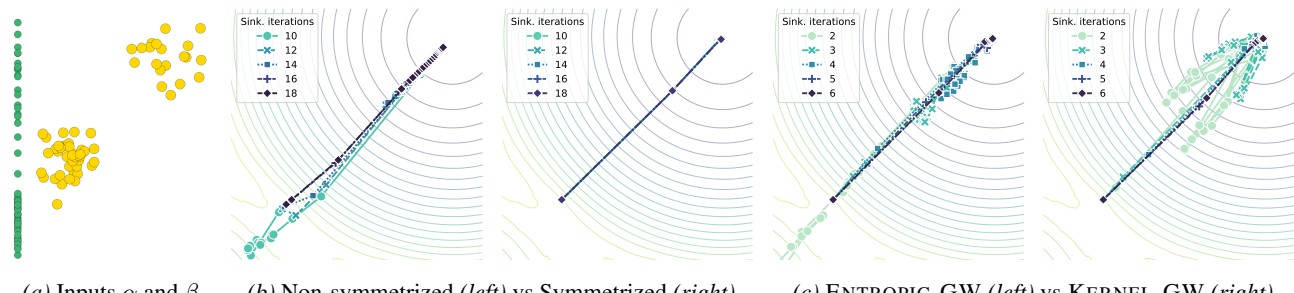

*(a)* Inputs $\alpha$ and $\beta$     *(b)* Non-symmetrized *(left)* vs Symmetrized *(right)*     *(c)* ENTROPIC-GW *(left)* vs KERNEL-GW *(right)*

*Figure 8.* Optimization steps obtained with EGW solvers in different configurations. The dual landscape $\Gamma \mapsto \mathrm{U}_\varepsilon(\Gamma)$ is plot in the background. Solvers are initialized at $\pi = \alpha \otimes \beta$, i.e. $\Gamma = 0$. In the right plot, both solvers use symmetrized Sinkhorn.

### D.1.2. CONVERGENCE SPEED EVALUATION

To evaluate the practical impact of symmetrization and kernelization on convergence, we study the convergence speed of ENTROPIC-GW and KERNEL-GW on different shapes, with standard and symmetrized Sinkhorn. The curves of this section were obtained by sampling $N = 2,000$ points uniformly from the horse shape of the SMAL dataset (Zuffi et al., 2017); we record elapsed time and EGW loss at each solving step. The curves are averaged over 20 random samplings (except for Figure 9 and Figure 10 were 50 samples were computed). We plot the results for different number of Sinkhorn iterations $n_{\text{inner}}$. We also evaluate solvers with adaptive scheduling: starting from a small number of iterations, $n_{\text{inner}}^0 = 5$, we double its value each time the estimated EGW loss increases from one step to the other. We compare ENTROPIC-GW and KERNEL-GW, as well as QUADRATIC-LOWRANK-GW and CNT-GW. Results are given in Figure 9.

Here are our key findings. Methods using standard Sinkhorn updates converge slowly; when the iteration budget $n_{\text{inner}}$ is low, they fail to reach the true global minimum. Applying symmetrization to ENTROPIC-GW accelerates convergence but introduces an instability: the solver tends to diverge for small $n_{\text{inner}}$ and does not benefit effectively from adaptive scheduling. With standard Sinkhorn, KERNEL-GW is slightly more stable than ENTROPIC-GW, as the spikes observed at the beginning of the curves are attenuated; however, it does not make any meaningful difference on convergence time. Finally, the combination of KERNEL-GW with symmetrized Sinkhorn provides the most robust performance. This configuration accelerates and stabilizes convergence across all values of $n_{\text{inner}}$, allowing the adaptive scheduling to function optimally. We obtain similar conclusions with the approximation methods QUADRATIC-LOWRANK-GW and CNT-GW.

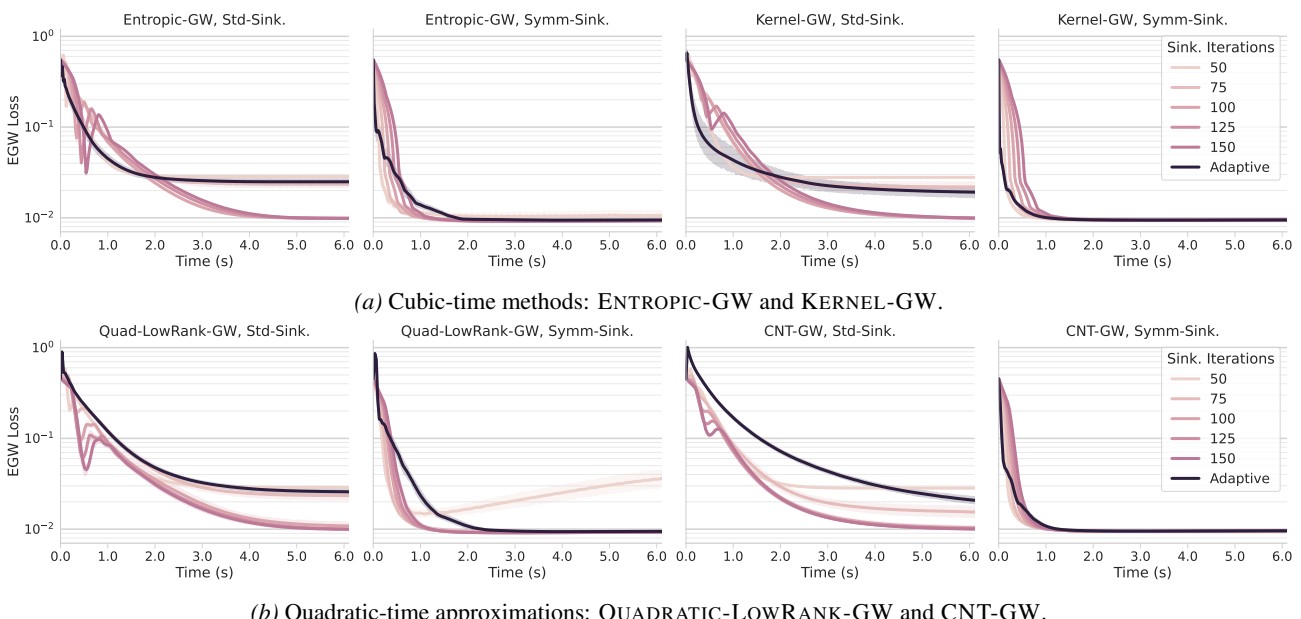

*(a)* Cubic-time methods: ENTROPIC-GW and KERNEL-GW.

*(b)* Quadratic-time approximations: QUADRATIC-LOWRANK-GW and CNT-GW.

*Figure 9.* Evolution of the EGW loss through time in different configurations, for various numbers of Sinkhorn iterations.

**Smaller Sinkhorn Iterations Number.**   These trends are amplified when we further reduce $n_{\text{inner}}$ (Figure 10). Notably, the symmetrized versions of ENTROPIC-GW and QUADRATIC-LOWRANK-GW exhibit important divergence. In contrast, KERNEL-GW and CNT-GW remain stable for $n_{\text{inner}} \geq 30$. When $n_{\text{inner}} \leq 20$, instabilities start to appear but final losses remain closer to the true minimum compared to baselines.

**Stopping Criterion for Sinkhorn Iterations.**   An alternative to a fixed iteration budget is to terminate the Sinkhorn loop dynamically based on marginal error. Specifically, given a threshold $\tau$, we iterate until the weighted $L_1$ error satisfies:

$$\sum_i \left| \sum_j \pi_{ij} - \alpha_i \right| \cdot \alpha_i < \tau \quad \text{and} \quad \sum_j \left\| \sum_i \pi_{ij} - \beta_j \right\| \cdot \beta_j < \tau.$$

Results in Figure 11 show that while symmetrizing Sinkhorn still improves convergence, the kernel implementation does not bring noticeable improvements. Sinkhorn thresholding also appears to be slower than choosing a fixed number of iterations,

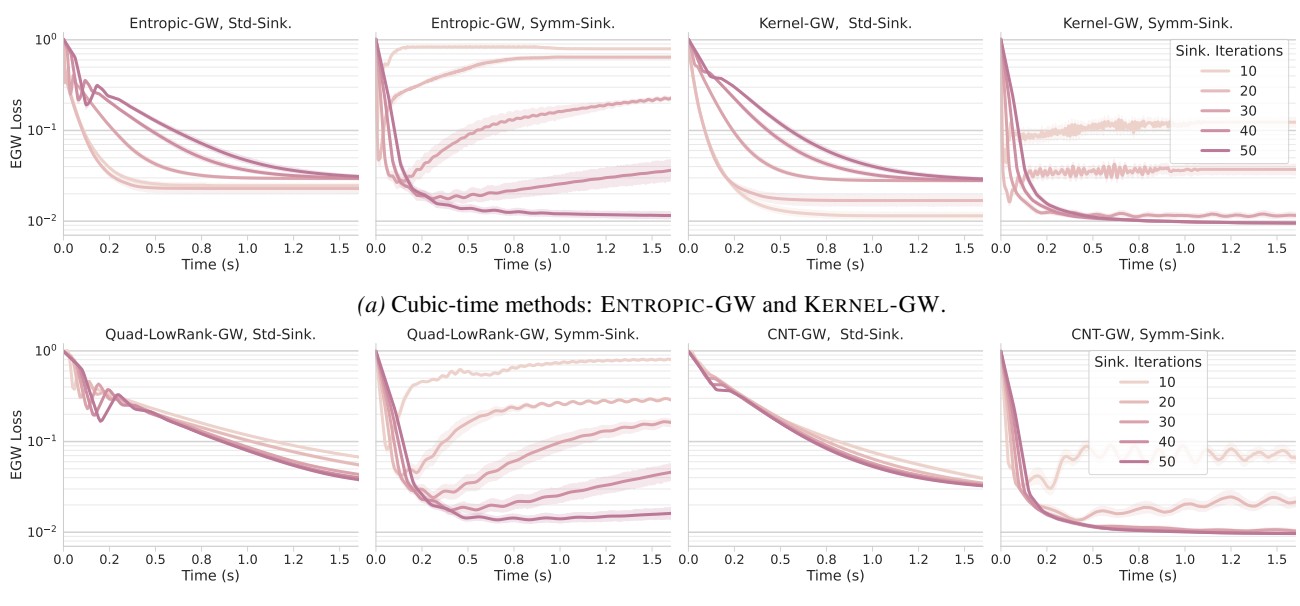

*(a)* Cubic-time methods: ENTROPIC-GW and KERNEL-GW.

*(b)* Quadratic-time approximations: QUADRATIC-LOWRANK-GW and CNT-GW.

*Figure 10.* Convergence curves for smaller values of $n_{\text{inner}}$.

and its sensitivity to the choice of $\tau$ makes this criterion more complex to tune. Therefore, we recommend using a fixed number of iterations, although hybrid strategies (that use both a fixed maximum of iterations and early stopping based on margin error) offer a good alternative.

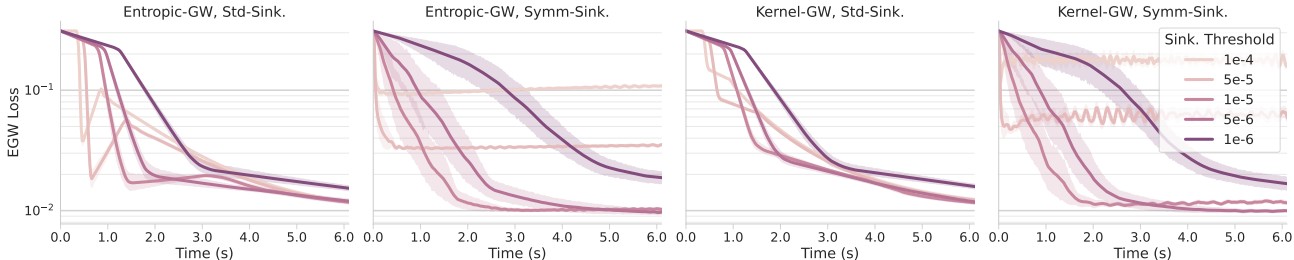

*Figure 11.* Convergence curves using Sinkhorn convergence thresholding.

### D.1.3. COMPARISON WITH OTHER SOLVERS.

We provide convergence curves of other EGW baselines, using the same experimental setup as in the previous section.

**Sampled-GW.** Figure 12a illustrates the behavior of SAMPLED-GW (Kerdoncuff et al., 2021) on the horse shapes. It highlights a fundamental limitation: random sampling induces important fluctuations in the EGW loss, preventing the algorithm from converging to a stationary point. This variability restricts its utility in applications requiring precise matching.

**Dual-GW.** We extend the dual solver of (Rioux et al., 2024) (originally for squared norms) to CNT costs using our Kernel PCA embeddings. As shown in Figure 12b, convergence remains slow, empirically confirming the theoretical convergence rate improvement proven in Theorem 4.6.

**Proximal-GW.** Finally, Figure 12c presents results for PROXIMAL-GW (Xu et al., 2019a) – which outputs exact GW solutions rather than EGW couplings. We also introduce a kernelized variant, PROXIMALKERNEL-GW, based on the same cost matrix as KERNEL-GW. Sinkhorn symmetrization still accelerates convergence in this proximal setting. However, kernel implementation provides no additional benefit here, and adaptive scheduling fails to converge properly.

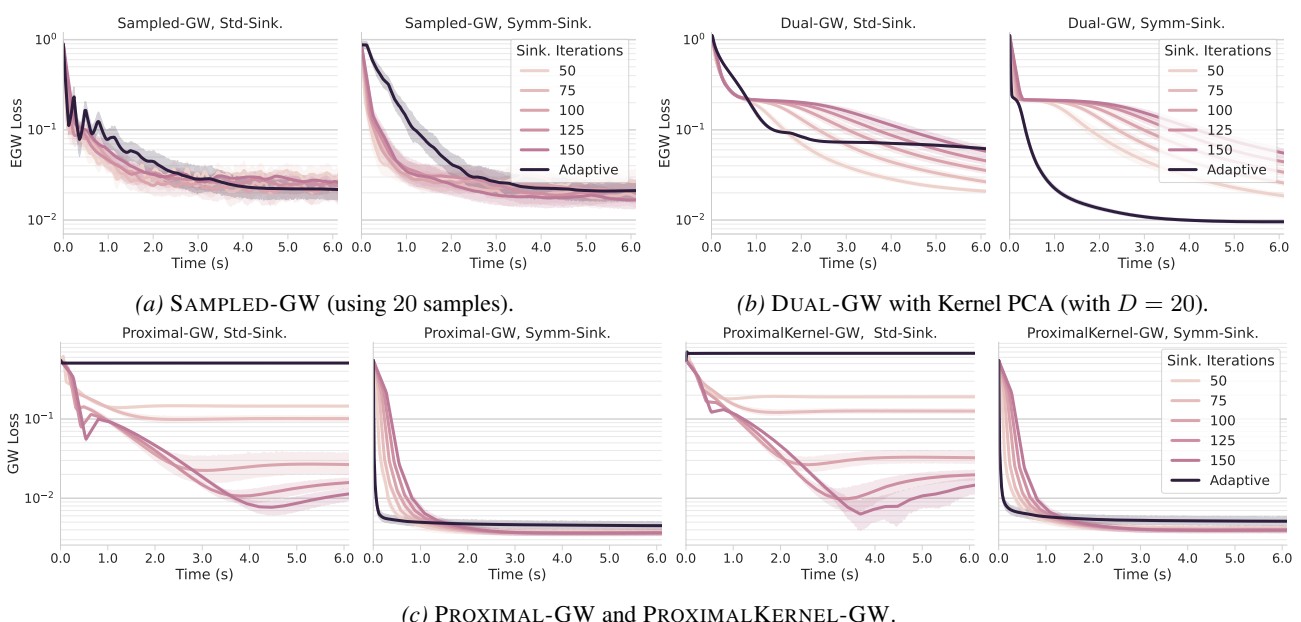

*(a)* SAMPLED-GW (using 20 samples).

*(b)* DUAL-GW with Kernel PCA (with $D = 20$).

*(c)* PROXIMAL-GW and PROXIMALKERNEL-GW.

*Figure 12.* Convergence curves of other solvers with standard or symmetrized Sinkhorn.

### D.1.4. CONVERGENCE ANALYSIS ON SYNTHETIC DATASETS.

**Synthetic Shapes.** We evaluate KERNEL-GW and ENTROPIC-GW on point clouds sampled from a unit sphere and a regular tetrahedron (Figure 13). The tetrahedra case exhibits the same trend as before. However, for spheres, symmetrizing Sinkhorn does not improve convergence speed: the large number of symmetries in the input shapes allows standard Sinkhorn methods to converge quickly even without symmetrization. Yet, run times for symmetrized KERNEL-GW are roughly equivalent to non-symmetrized solvers while symmetrized ENTROPIC-GW is highly unstable when the number of iterations is too small.

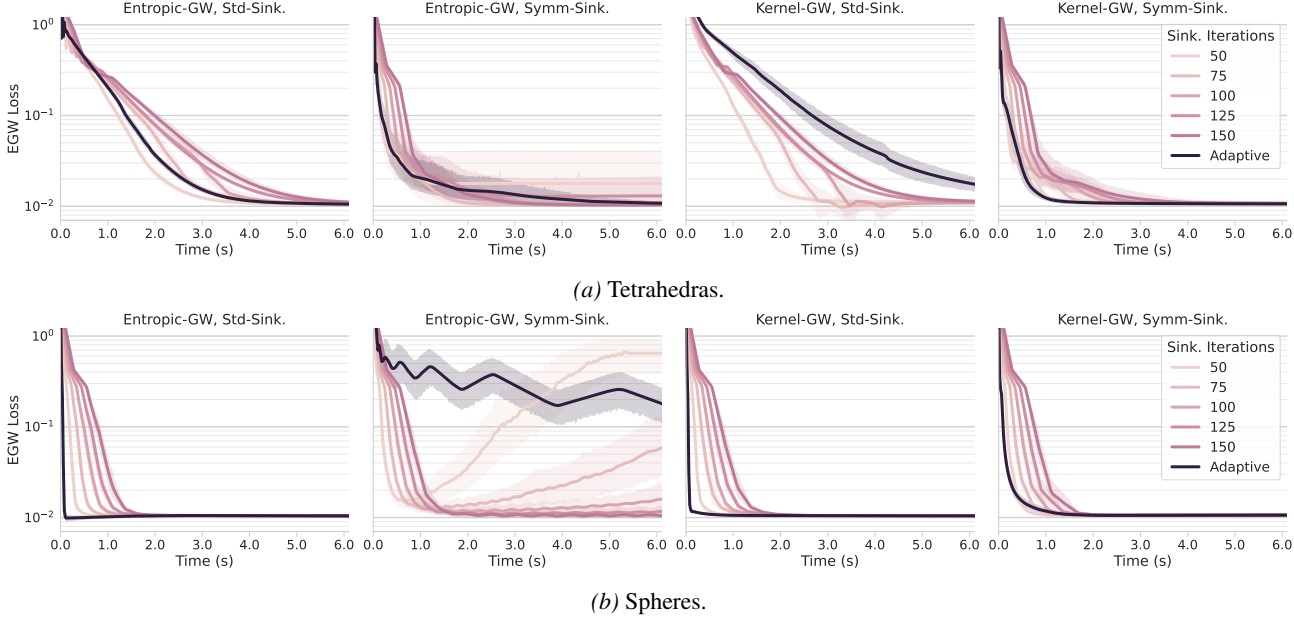

*(a)* Tetrahedras.

*(b)* Spheres.

*Figure 13.* Convergence curves on synthetic surface distributions.

**Gaussian Inputs.** On unstructured Gaussian distributions (Figure 14), KERNEL-GW with symmetrized Sinkhorn clearly outperforms all other implementations, demonstrating superior stability in the absence of geometric structure. This is especially true for 3D distributions, where symmetrized ENTROPIC-GW fails to converge even with $n_{inner} = 150$.

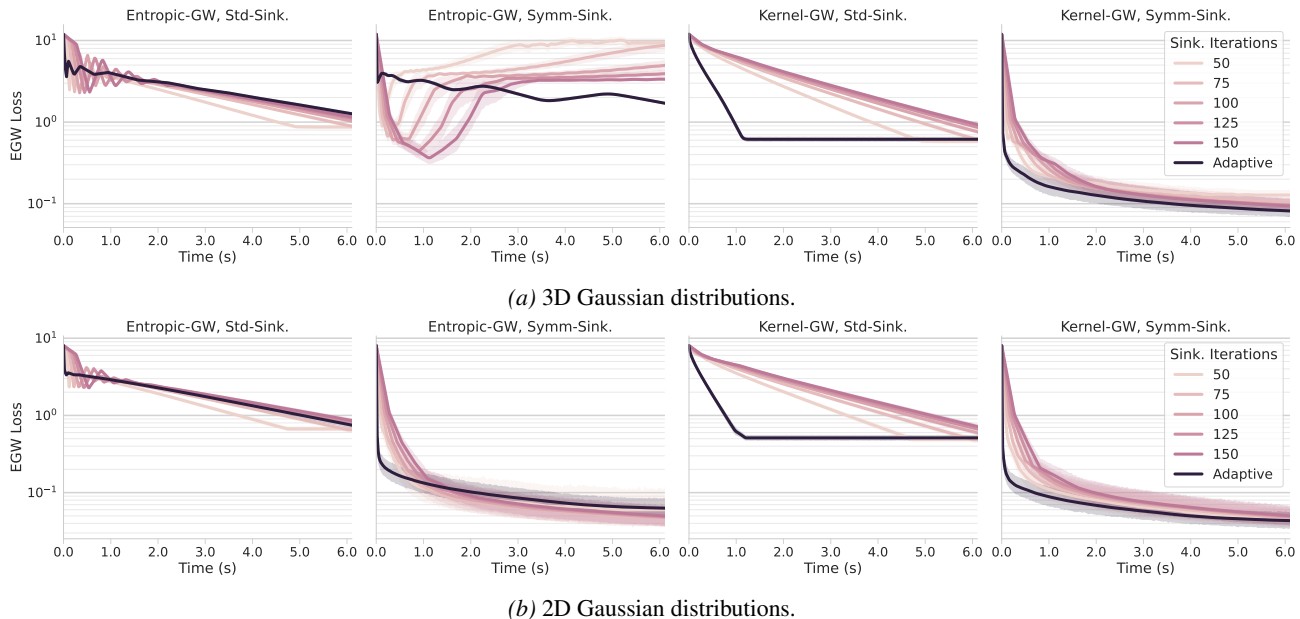

*(a)* 3D Gaussian distributions.

*(b)* 2D Gaussian distributions.

*Figure 14.* Convergence curves on Gaussian samples.

**Distinct Input Distributions.** We finally identify a scenario where our method is less effective than the baseline. When the input shapes differ significantly (Figure 15), particularly when one input is highly symmetric (e.g., Figure 15b), symmetrized KERNEL-GW performs slightly worse than standard ENTROPIC-GW and symmetrization generates oscillations for small Sinkhorn iterations. However, our method remain convergent for sufficiently large iteration numbers, and adaptive Sinkhorn still succeeds to converge to the true optimum. Therefore, our algorithm remains competitive even in this scenario.

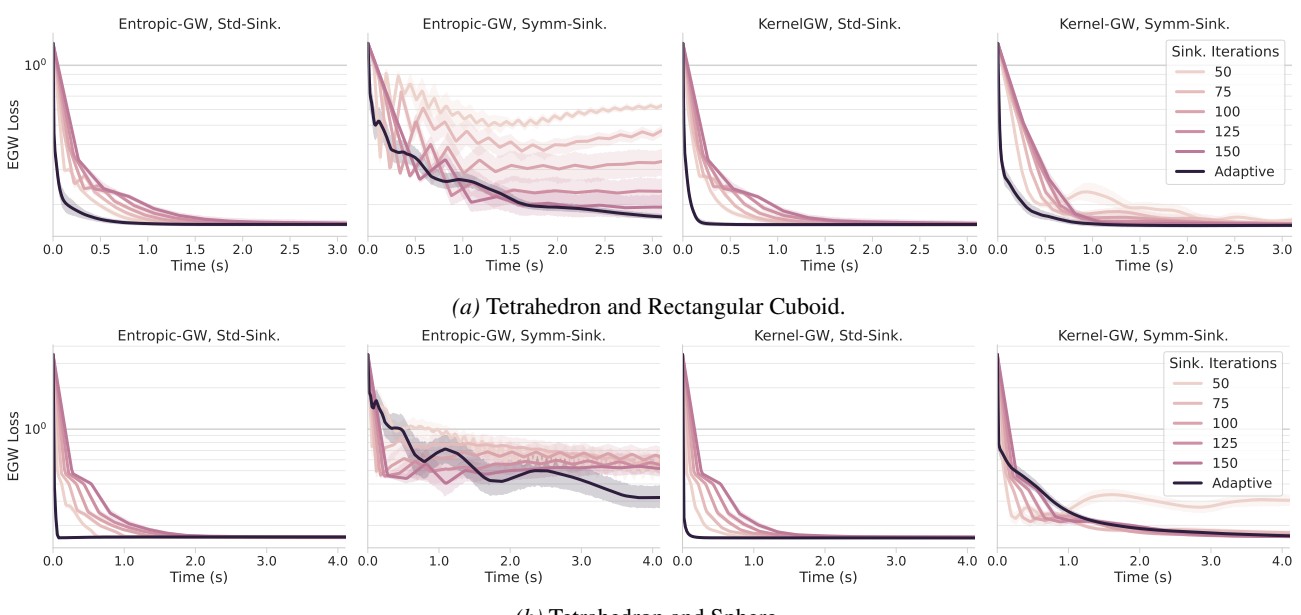

*(a)* Tetrahedron and Rectangular Cuboid.

*(b)* Tetrahedron and Sphere.

*Figure 15.* Convergence curves on distinct surface distributions.

## D.2. Complementary Figures

**Benchmark of EGW Solvers.** Figure 16a displays the convergence times of our solvers and the main baselines as a function of the input size $N = M$. Results were obtained by sampling point clouds uniformly from the horse shape of the SMAL dataset, and highlight the significant performance improvement of our kernel methods over ENTROPIC-GW and QUADRATIC-LOWRANK-GW: KERNEL-GW and CNT-GW achieve a $2 - 4\times$ speed-up over their direct competitors, while MULTISCALE-GW is an order of magnitude faster. Figure 16b shows that all methods output transport plans with equivalent losses, which confirms that the differences of convergence speeds are solely due to solver performances and not to a difference of solution quality [4].

**Impact of the Embedding Dimension on the Quality of the Approximation.** Figure 16c displays the relative loss differences between true optimal plans and approximate ones obtained with CNT-GW and QUADRATIC-LOWRANK-GW, with $N = 2,000$. These results show that CNT-GW maintains an approximation quality equivalent to that of LOWRANK-GW, demonstrating that our computational speed-ups do not compromise the quality of the output. They also imply that approximation errors drop quickly as the embedding dimension increases, $D = 20$ being sufficient to reach a $1\%$ error on the exact EGW loss.

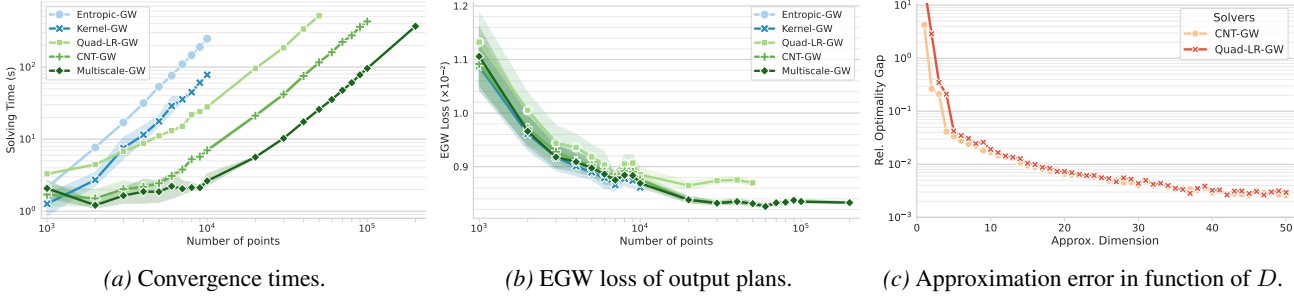

*(a)* Convergence times.            *(b)* EGW loss of output plans.            *(c)* Approximation error in function of $D$.

*Figure 16.* Benchmark of EGW solvers on inputs uniformly sampled on horse surfaces from the SMAL dataset (with $c_X, c_Y = \|\cdot\|_{\mathbb{R}^3}$).

**Visualization of transport plans.** In Figure 17, we represent the optimal plans produced by the different solvers on the hands shapes (since MULTISCALE-GW is an acceleration of CNT-GW, we did not represent its output as it would be identical to CNT-GW). We also visualize in Figure 18 the results of QUADRATIC-LOWRANK-GW and CNT-GW on the other shapes of our benchmark, as they are the only methods able to scale to these inputs. We see that all results are qualitatively similar, confirming the quantitative trend observed in Table 1. Overall, the texture transfers are smooth and provide meaningful correspondences between the sources and targets. Some artifacts can be noticed on the Kids and Vessels examples, on the elongate parts of the shapes: these are well-known in shape matching and require more advanced techniques in order to be solved.

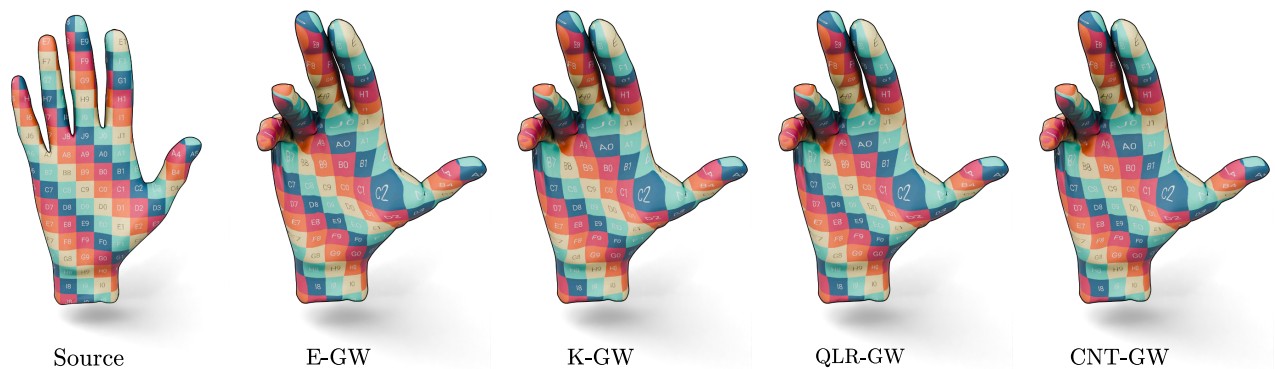

Source            E-GW            K-GW            QLR-GW            CNT-GW

*Figure 17.* GW-based texture transfer between hand shapes.

---

[4]For memory reasons, the true EGW losses could not be computed for $N > 10^4$, and we relied on solver-specific approximations: this explains the gap between CNT-GW/MULTISCALE-GW and QUADRATIC-LOWRANK-GW in this regime.

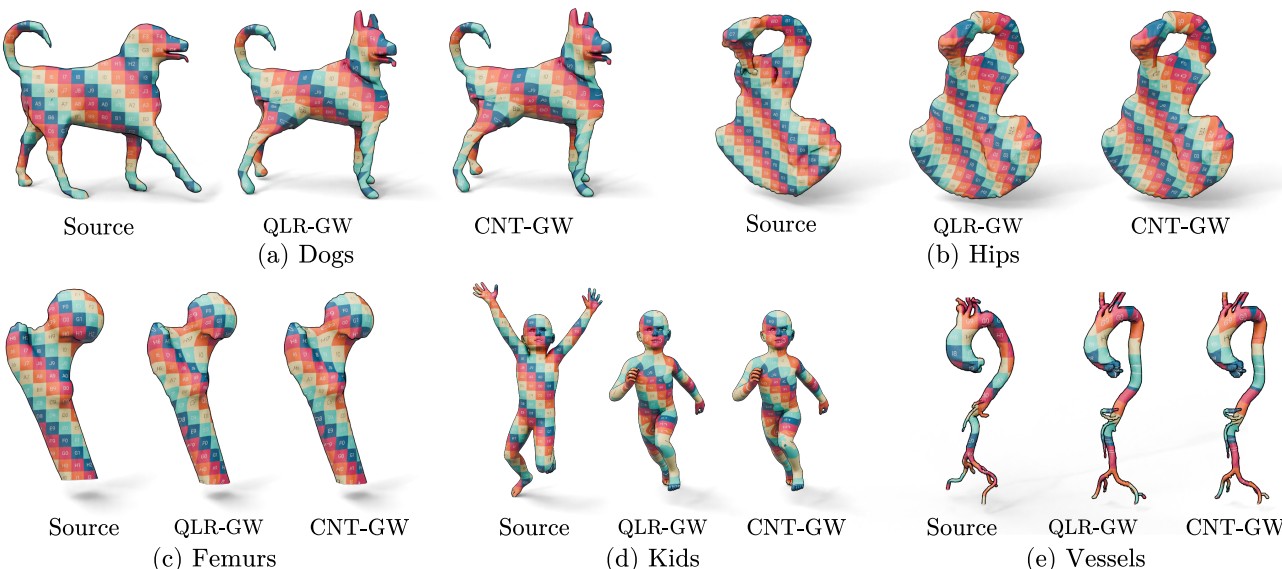

*Figure 18.* GW-based texture transfer between the other shapes used in the benchmarks.

**Impact of Base Costs on EGW Geometry.** We explore the EGW geometries induced by different base costs by displaying a gradient flow between two simple shapes, as we use squared norms, Euclidean norms, root norms and exponential kernels with different radii. The results are given in Figure 19: more global costs tend to preserve the global ordering of the points during the deformation and create discontinuities to reverse the upper bar of the "5" shape. On the other hand, local costs keep the continuity of the shape, and follow non-trivial paths to push the source shape onto the target one. The CNT framework is sufficiently expressive to adjust the balance between these local and global-oriented behaviors.

In Figure 20, we also plot the flows obtained with approximate gradients (Algorithm 12) using a Kernel PCA dimension of $D = 50$. The results are similar to the exact case; however, we also observe several artifacts that highlight the limitations of dimension reduction for precise gradient computations. Note that these artifacts do not modify the asymptotic distribution of the point cloud; despite the approximation, all gradient flows eventually converge to the correct shape.

## E. Experiment Details

**Numerical Backend.** All solvers were implemented in the `PyTorch` framework using single precision float numbers (float32) and run on GPU (Paszke et al., 2017). We used the `KeOps` package to reduce the memory footprint of computations whenever possible. This includes the implementation of CNT-GW as well as the baselines QUADRATIC-LOWRANK-GW, DUAL-GW and SAMPLED-GW. We used truncated PCA decomposition to compute the low-rank approximation of cost matrices in QUADRATIC-LOWRANK-GW. We systematically use ranks of $D = E = 20$ for QUADRATIC-LOWRANK-GW, matching the choices of dimensions made for CNT-GW. We used `KeOps` to compute this truncated PCA, as well as for Kernel PCA in CNT-GW. In our main benchmarks, we implemented our algorithms KERNEL-GW, CNT-GW and MULTISCALE-GW using Symmetrized Sinkhorn (Algorithm 3). We implemented ENTROPIC-GW and QUADRATIC-LOWRANK-GW using standard Sinkhorn (Algorithm 2), which corresponds to the original implementation as introduced by their authors and avoids all numerical instabilities introduced by symmetrized Sinkhorn to these algorithms (Section D.1).

**Libraries.** We also used the `NumPy` (Harris et al., 2020) and `SciPy` (Virtanen et al., 2020) libraries for scientific computing. We relied on `Matplotlib` (Hunter, 2007), `Seaborn` (Waskom, 2021) and Blender for visualizations.

**Computation of Geodesic Embeddings.** To obtain Euclidean approximations of geodesic distances, we implemented the method of (Panozzo et al., 2013). We decimated input meshes, reducing the number of vertices to $N = 2,000$. We computed the exact pairwise geodesic distances on the decimated meshes, and embedded the coarse vertices in $\mathbb{R}^8$ using Multi Dimensional Scaling. We finally interpolated the 8-dimensional Euclidean coordinates of all vertices by solving a quadratic interpolation problem on the original mesh. The Euclidean norm $\|\cdot\|_{\text{geodesic}}$ on this Euclidean embedding approximates true

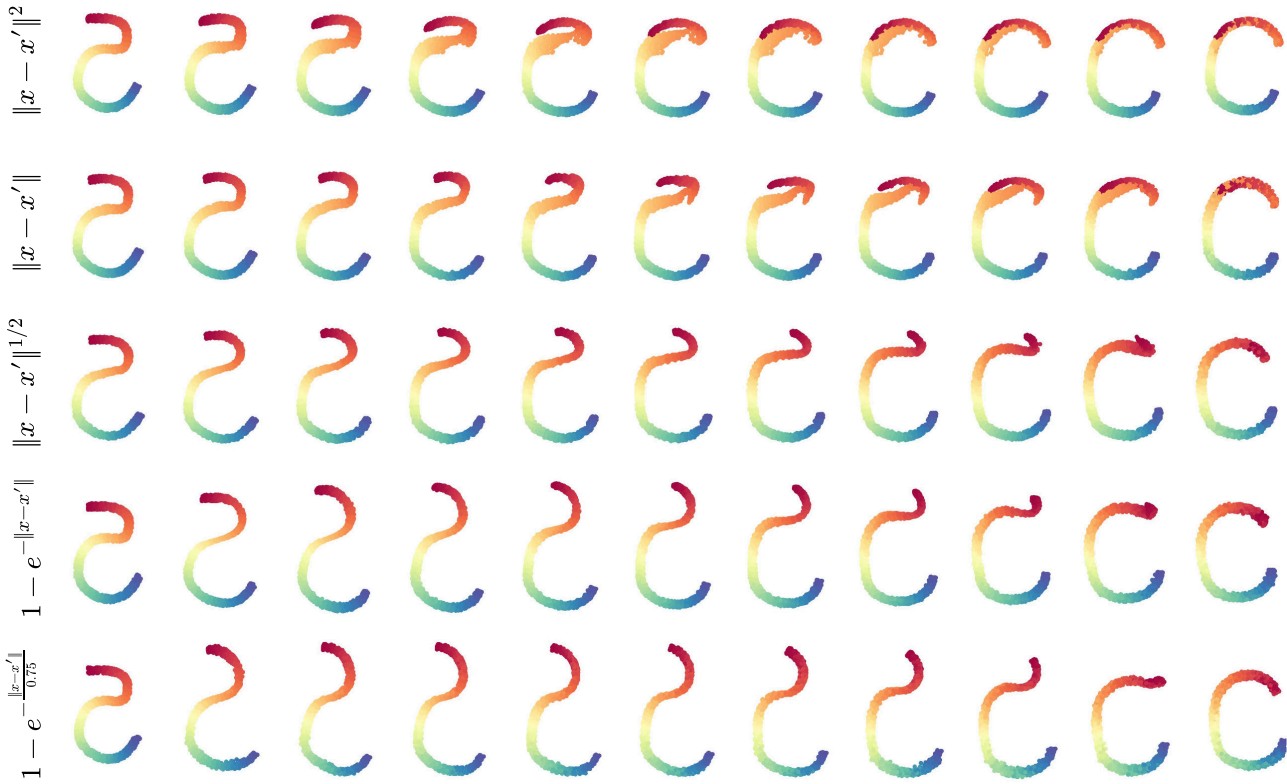

*Figure 19.* Gradient flows $\alpha_{t+\delta t} = \alpha_t - \delta t \nabla \mathrm{SGW}_\varepsilon(\alpha_t, \beta)$ from a reversed "5" shape $\alpha_0$ to a "C" shape $\beta$, using different base costs.

geodesic distances between all pairs of vertices.

**Other Implementation Choices.** For MULTISCALE-GW, we systematically choose a coarsening ratio of $\rho = 0.1$. Unless specified, we initialize all solvers with the trival transport plan $\pi^0 = \alpha \otimes \beta$ (or $\Gamma^0 = 0$ for dual-based solvers). We always take $\alpha$ and $\beta$ as uniformly weighted point clouds (i.e. $a_i = 1/\mathrm{N}$ and $b_j = 1/\mathrm{M}$ for all $i, j$).

**Transport plan visualizations.** To visualize transport plans $\pi$ between 2D or 3D shapes, we either transfer color or texture from the source to the target. For color transfer, we assign an RGB value $C_\mathcal{X}(x) \in \mathbb{R}^3$ to each point $x$ in the source shape. The color at each target point is then defined as the weighted average:

$$\text{For all } y \in \mathcal{Y}, \quad C_\mathcal{Y}(y) = \int C_\mathcal{X}(x) \frac{\mathrm{d}\pi(x, y)}{\mathrm{d}\alpha(x)\mathrm{d}\beta(y)} \mathrm{d}\alpha(x).$$

Texture transfer follows a similar idea and rely on the notion of *UV mapping* in 3D graphics. Each source point $x$ is assigned UV coordinates $UV_\mathcal{X}(x) \in \mathbb{R}^2$, which are then transported to the target via:

$$\text{For all } y \in \mathcal{Y}, \quad UV_\mathcal{Y}(y) = \int UV_\mathcal{X}(x) \frac{\mathrm{d}\pi(x, y)}{\mathrm{d}\alpha(x)\mathrm{d}\beta(y)} \mathrm{d}\alpha(x).$$

Textures are represented as 2D images, with UV coordinates mapping each vertex of a 3D shape to a location in the image. During rendering, these coordinates are interpolated across each triangle of the mesh to produce the final textured surface.

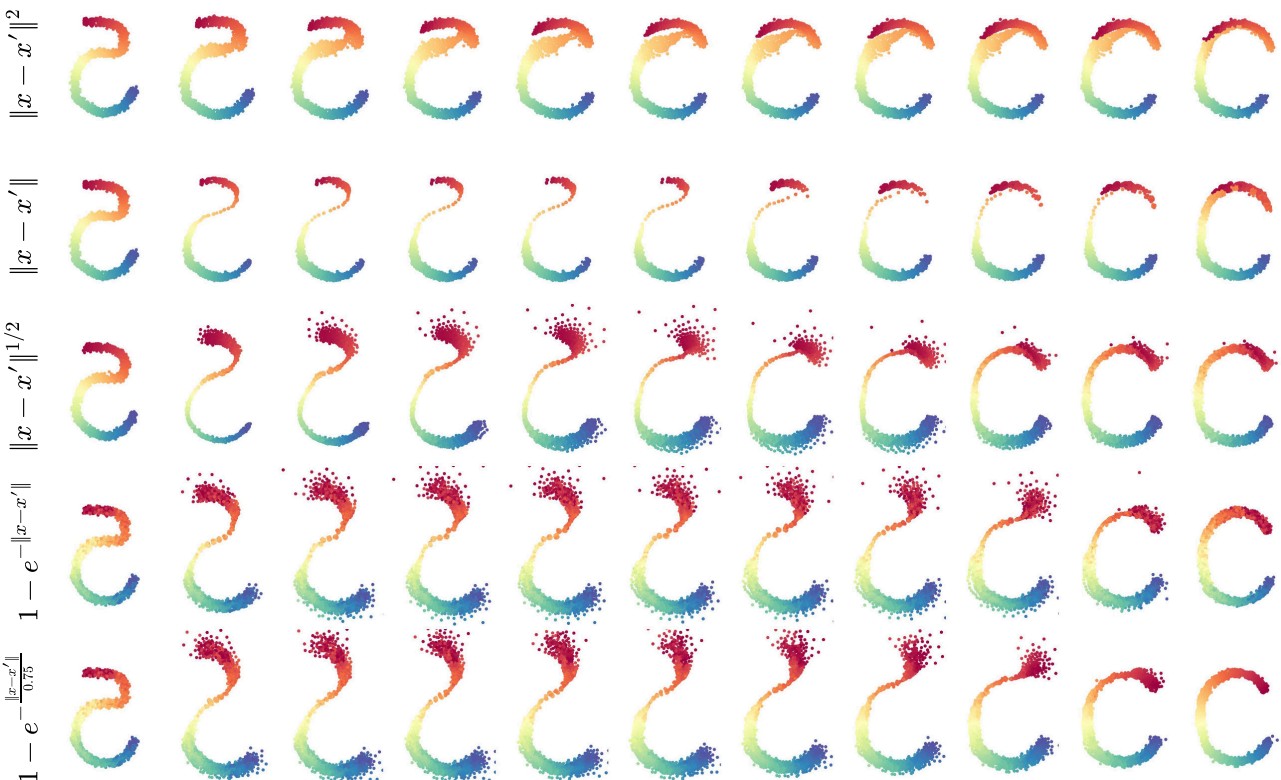

*Figure 20.* Gradient flows with approximate gradients obtained via Kernel PCA ($D = 50$).

