# OpenReview forum: "Gromov-Wasserstein at Scale, Beyond Squared Norms"
_ICML.cc/2026/Conference — ICML 2026 regular_

### Official Review · Reviewer_2R4U · 2026-02-27

**Soundness:** 4
**Presentation:** 3
**Significance:** 3
**Originality:** 4
**Overall Recommendation:** 5
**Confidence:** 4

**Summary:**

This paper introduces a new interpretation of the (possibly regularized) Gromov-Wasserstein (GW) problem between two probability measures as an optimisation problem that alternates between two minimization problems: the linear registration of point clouds and (possibly regularized) optimal transport with a cost function depending on a linear mapping. This problem is formulated for a general class of conditionally of negative type (CNT) costs (including the use of squared norms as in the standard formulation of GW) that enjoy a representation via a feature map that is reminimiscent of what reproducing kernels are for scalar products. The main contributions of the paper are: the proof of such an interpretation showing that computing the GW cost is equivalent to an optimal linear alignement between features space followed by an OT coupling with a ground cost that depends on this optimal registration step; a theoretical study of the entropic bias of regularized GW for the class of CNT costs, and the theoretical study of a new GW solver that uses an alternate minimization scheme exploiting this new formulation of the GW problem;  extensise numerical experiments with comparisons to state-of-the-art computational methods to solve the GW problem that show the superiority of this novel formulation of GW.

**Compliance With Llm Reviewing Policy:**

Affirmed.

**Final Justification:**

I recommend accepting this paper as the authors present an interpretation of the (possibly regularized) Gromov-Wasserstein (GW) problem, supported by both a new algorithm and convincing experimental results.

That said, I suggest revisiting the claim that one of a key contribution is demonstrating that computing the GW cost is equivalent to an optimal linear alignement between features space followed by an OT coupling with a particular ground cost. This assertion should be carefully reconsidered in light of the inclusion of reference [R1].

[R1] O. Sebbouh, M. Cuturi, and G. Peyré. Structured transforms across spaces with cost-regularized optimal transport. In International Conference on Artificial Intelligence and Statistics (pp. 586-594), 2024.

**Key Questions For Authors:**

The main contribution of the paper is to show that GW is equivalent to an optimal alignment of the data by a linear mapping followed by a standard OT problem once the two data sets have been optimally registered. Would it be possible to propose simple deformation models where solving the GW problem recovers an optimal deformation between two probability measures while solving a standard OT fails in computing such a deformation ?

Is it also possible to use the results of the paper to characterize the optimal linear mapping that is obtained by solving the GW problem between two Gaussian supported on spaces of different dimensions ?

Finally, it appears that reference [R1] is missing. Could you explicitly discuss how your contributions differ from or extend the results presented in that reference?

[R1] O. Sebbouh, M. Cuturi, and G. Peyré.  Structured transforms across spaces with cost-regularized optimal transport. In International Conference on Artificial Intelligence and Statistics (pp. 586-594), 2024.

**Limitations:**

Yes

**Strengths And Weaknesses:**

**Soundness :** This paper is vey sound both on the  theoretical analysis of the new GW formulation, the computational aspects (study of a new algorithm based on an alternative scheme with a detailed discussion on its computational cost) and the experimental results. The proposed method is very appropriate with proofs that seem correct under relevant assumptions on the use of CNT costs.

**Presentation :** The paper is clearly written and well structured

**Significance :** The paper addresses an important challenge about the scalability of the GW problem. Moreover, the novel formulation that is proposed opens a new research direction on its connexion with the field of structured data alignment using geometric methods.

**Originality :** The paper provides new insights and a deeper understanding of the GW problem with respect to existing methods. The contributions and novelty of the paper are clearly explained and distinguished from the related literature on the GW problem.

---

> ### Author Rebuttal · Authors · 2026-03-30
>
> We thank the reviewer for its positive review and for pointing out an important missing reference.
>
> ### Deformation models recovered by GW but not OT.
> If one input is obtained from the other by an isometry, then GW will recover the optimal matching whereas OT may fail (as OT is sensitive to rotations and translations). Besides isometries, GW can recover other kinds of deformations that OT cannot: for instance, our Figure 1.c shows that GW can recover an optimal matching between shapes obtained after complex deformations, something that OT cannot obtain. The characterization of which deformations leave GW matching invariant, however, seems rather hard to establish.
>
> ### GW transport plans between Gaussians.
> The GW problem on Gaussian inputs has been extensively studied in [Del]: this article was one of the inspirations of our paper, as the authors rely on a formulation similar to ours. Unfortunately, the squared norm term of the GW-embeddings $\Phi$ and $\Psi$ (that we introduced in page 4) does not preserve Gaussian measures, which makes GW transports between Gaussians hard to characterize. This led the authors to introduce a variant of GW specific to the Gaussian case [Del], for which the theory is easier to develop.
>
> [Del] Delon, Julie, Agnes Desolneux, and Antoine Salmona. "Gromov–Wasserstein distances between Gaussian distributions." Journal of Applied Probability 59.4 (2022): 1178-1198.
>
> ### Relation to reference [R1].
> While the main decomposition of [R1] is similar to ours, we took a different approach to the GW problem: our work focuses on the geometric and algebraic properties of GW while [R1] rather deals with more general matching problems with a strong algorithmic focus. Our contributions are hence distinct from this previous work: we provide an in-depth analysis of the debiasing of GW, provide quantitative convergence guarantees of our alternate minimization scheme, and we introduce a full end-to-end numerical approach to solve GW (dimension reduction, multiscaling, gradient estimation, exploration of local minima). On the other hand, [R1] provides a more generic numerical scheme that applies to a more general class of problems but without our degree of detail on its theoretical and algorithmic properties. Finally, we focused our work on CNT costs while only the squared-Euclidean case is discussed in [R1]. We will discuss this in our paper.
>
> [R1] O. Sebbouh, M. Cuturi, and G. Peyré. Structured transforms across spaces with cost-regularized optimal transport. In International Conference on Artificial Intelligence and Statistics (pp. 586-594), 2024.

---

> > ### Author Rebuttal · Reviewer_2R4U · 2026-04-01
> >
> > I thank the authors for their relevant answers to my comments. I propose to keep my overall recommendation and to accept this paper.

---

### Official Review · Reviewer_Xv5M · 2026-03-07

**Soundness:** 4
**Presentation:** 3
**Significance:** 4
**Originality:** 4
**Overall Recommendation:** 5
**Confidence:** 4

**Summary:**

This paper studies EGW with CNT cost. The main technical contribution is a reformulation of EGW with CNT cost in terms of two variables: a transport plan and a Hilbert-Schmidt linear map between feature spaces induced by CNT embeddings. Under this reformulation, the EGW problem can be solved by alternating between (i) an classic entropic OT problem in a lifted feature space, and (ii) a closed-form update of the linear map. This yields a differentiable solver with linear memory and quadratic time in the number of samples, together with convergence guarantees to critical points. The paper also studies a debiased Sinkhorn-GW divergence and proves nonnegativity and nullity properties in the CNT setting. Numerical results show that the proposed method is more scalable than existing EGW solvers.

**Compliance With Llm Reviewing Policy:**

Affirmed.

**Final Justification:**

My concerns have been fully resolved and I keep my recommendation for acceptance.

**Key Questions For Authors:**

1. Can the authors also report the original GW objective using the transport plan returned by the proposed method and compare with other solvers?

2. Can the authors prove/discuss also the convergence to critical points for the proposed alternating scheme in the non-CNT setting? Even a perturbation-style discussion would strengthen the practical story.

3. The reformulation of GW proposed in this paper reminds me of Lemma 3.3 in Sturm's paper [1]. In short, this lemma says that the GW distance can be viewed as the Wasserstein distance after `optimally' embedding both metric-measure spaces into a common ambient metric space. Is there any relation between this result and the proposed formulation?

[1] Karl-Theodor Sturm "On the geometry of metric measure spaces," Acta Mathematica, Acta Math. 196(1), 65-131, (2006)

**Limitations:**

Yes.

**Strengths And Weaknesses:**

Strengths:

1. The paper contains an interesting reformulation of GW with CNT cost. Recasting EGW as alternating optimization over a transport plan and a linear operator in lifted feature spaces is conceptually elegant.

2. The theory is meaningful. In the CNT setting, the authors provide an exact reformulation, identify the alternating scheme with both primal descent and dual gradient descent, and prove monotone decrease and convergence to critical points. The debiasing result for SGW in the CNT setting is also a valuable plus.

3. The reported memory and runtime improvements are significant, especially at large scale.

Weaknesses:

1. The theoretical results are restricted to CNT costs, but several experiments use non-CNT cost such as the geodesic distances. In those cases, the method is used with approximate kernel-PCA embeddings, but the paper does not provide an analogue of the main theory for the non-CNT case.

2. The empirical section emphasizes convergence time and scalability much more than solution quality. In particular, Table 1 reports timing results, but the main paper does not prominently report the final GW objective values for the returned transport plans. I think the ultimate goal of using the proposed method is to obtain a good approximation for the original GW, not just faster convergence.

---

> ### Author Rebuttal · Authors · 2026-03-30
>
> We would first like to thank the reviewer for its positive review and its suggestions to increase the strength of our article.
>
> ### Comparison of GW objectives between solvers.
> We refer to the answer of reviewer 6mgZ, paragraph "Evaluation of solution quality". In particular, our algorithms provide transport plans of similar quality to concurrent methods while taking less time to converge; we will make this clearer in our article.
>
> ### Extension of results to the non-CNT case.
> Our framework cannot be straightforwardly extended to the non-CNT case, as the reformulation of theorem 3.2 is intrinsically related to Schoenberg's theorem (Theorem 2.2).
> In particular, our alternate scheme is undefined in the non-CNT setting; we also found a counter-example to the convergence of the entropic-GW method (figure 6), which is equivalent to ours in the CNT case. Therefore, it seems impossible to directly extend such convergence results, and we should design new numerical methods to handle non-CNT costs in a robust way: this will be the subject of a future work.
>
> Yet, one can apply our algorithms on CNT approximations of general costs, with quantitative bounds on the approximation error. Indeed, in the discrete setting, the CNT property is directly linked to the spectrum of the kernel matrices $K^X$ and $K^Y$ (defined in section B.2 of the appendix): the cost matrices $C^X$ and $C^Y$ are CNT if and only if all the eigenvalues of $K^X$ and $K^Y$ are positive. By restricting these kernel matrices to their positive eigenspaces, we obtain a CNT approximation of the original cost with a control on the GW objective approximation error:
>
> $$\Delta GW \leq Cste * (R_X * | \lambda^Y_- | + R_Y * | \lambda^X_- |),$$
>
> where $R_X, R_Y$ are the diameters of $X, Y$ and  $\lambda^X_-$, $\lambda^Y_-$ are the smallest negative eigenvalues of $K^X$ and $K^Y$. If the base costs $C^X$ and $C^Y$ are almost CNT, this strategy hence provides accurate estimations of the GW objective while allowing the use of our algorithms. The eigenvalues $\lambda^X_-$ and $\lambda^Y_-$ can be computed using standard linear algebra solvers; they are also a byproduct of the kernel PCA performed by our dimension reduction method so that the approximation bound is actually computable in practice.
>
> ### Relation to Sturm's distance.
> Minimization of  optimal transport costs (e.g. Wasserstein distances) over a set of transformations of the space is common in the shape matching literature, and Sturm's distance (that we abbreviate as S-D in the following) is a typical example. The comparison between GW and S-D has been extensively discussed in [Mem], were the author proved that S-D $>=$ GW (cf. [Mem, Theorem 5.1 (g)]). An important conceptual difference between both formulations is the set of embeddings on which the Wasserstein distance is minimized: S-D takes a minimum over isometric embeddings, whereas our GW formula minimizes over the set of linear transform. This makes S-D more rigid and computationally intractable (as the set of isometries is non-convex and hard to parametrize), while our reformulation of GW involves a minimization over the convex set of linear transforms whose structure is well more adapted to numerical resolution.
>
> [Mem] Mémoli, Facundo. "Gromov–Wasserstein distances and the metric approach to object matching." Foundations of computational mathematics 11.4 (2011): 417-487.

---

> > ### Author Rebuttal · Reviewer_Xv5M · 2026-04-02
> >
> > I would like to thank the authors for their response. My concerns have been fully resolved and I keep my recommendation for acceptance.

---

### Official Review · Reviewer_mJBs · 2026-03-09

**Soundness:** 4
**Presentation:** 3
**Significance:** 4
**Originality:** 4
**Overall Recommendation:** 5
**Confidence:** 4

**Summary:**

This paper addresses the computational bottleneck of the Gromov-Wasserstein (GW) problem, which is known to be computationally demanding even with entropic regularization. The authors shown that when the cost functions are the Conditionally of Negative Type (CNT) class, the EGW problem can be elegantly reformulated. It reduces to find a linear alignment in a lifted Hilbert space (via kernel embeddings) followed by a standard Optimal Transport problem under squared Euclidean distance. Leveraging on this theoretical result, the authors propose a scalable and efficient EGW solver (CNT-GW) and a multiscale variant (MSGW) with linear memory footprint and quadratic time complexity. The proposed solvers can successfully solve the high-dimensional EGM problem with acceptable computational time and memory usage.

**Compliance With Llm Reviewing Policy:**

Affirmed.

**Final Justification:**

I remain positive about the paper and maintain the original score.

**Key Questions For Authors:**

1. How does CNT-GW perform on high-dimensional ML datasets (e.g., word embeddings or single-cell RNA sequences) compared to LowRank-GW? Does the required embedding dimension $D$ scale poorly when the intrinsic dimensionality of the data increases? How should a user determine $D$ for a concrete high-dimensional dataset.

2. In Section 6, you mentioned using 5 pairs of landmarks to initialize $\Gamma\_{0}$ for the FAUST dataset. How sensitive is the final convergence to the choice of these initial landmarks? Can MSGW succeed completely unsupervised on such high-resolution meshes without any prior landmark guidance?

3. Could you elaborate on the computational time overhead of the analytical gradient approximation (Algorithm 12) during a standard neural network training backward pass compared to the forward pass times?

4. The choice of temperature is crucial for the EGW problem. Can you comment on how to choose it for a practical dataset? Should we choose a fixed temperature or use some data driven method to tune this parameter?

**Limitations:**

No. The authors should discuss some limitations of their proposed method.

**Strengths And Weaknesses:**

Strengths:

The manuscript gives nice theoretical results on the EGW problem with CNT costs. For example, it's proved that, under CNT costs, the SGW can correct the entropic bias (Theorem 4.2).

By using Schoenberg's theorem, the authors introduce a new GW-embedding that can be parameterized and translate the original combinatorial matching problem into a more tractable linear registration task combined with standard OT. The latter problem can be efficiently solved with linear memory footprint and quadratic computational time.


Weaknesses:

The experiments heavily focus on 3D shape registration and low-dimensional point clouds. While impressive, GW is widely used in other high-dimensional ML domains (e.g., single-cell data integration, or graph matching). The lack of experiments on these high-dimensional tasks slightly limits the demonstration of its broader ML utility.

The practical speedup heavily relies on the Kernel PCA truncation dimension $D$ (which is set to $D=20$ in the experiments). For highly complex, non-linear, or high-dimensional real-world datasets, $D$ might need to be chosen to be large for capturing enough variance. This could slow down the computational time.

---

> ### Author Rebuttal · Authors · 2026-03-30
>
> We thank the reviewer for its positive feedback and its relevant remarks.
>
> ### Performance on high-dimensional datasets and practical choice of $D$.
> On paper, our algorithms scale as well as competing baselines with respect to the dimension (the time complexity of Cnt-GW and Ms-GW is $O(D N^2 + N D^2)$, equivalent to state-of-the-art solvers). The current limitation to its use on high-dimensional data is our practical implementation, which relies on KeOps, a library specifically optimized for low-dimensional computations: when $D > 100$, KeOps performance tends to drop significantly. Other optimal transport backends suited for high dimensions exist, and we would expect our methods to be at least similar to existing baselines if reimplemented using such backends. An other bottleneck of our current implementation is the impossibility to perform an exact kernel-PCA in high dimension. However, scalable kernel decompositions have been thoroughly studied in the literature, and kernel approximations such as the Nyström method would make our algorithms completely scalable for large $D$. Although we de not have any empirical experiment to evaluate our solver in this setting, it is worth noting that our theoretical results are not specific to low dimension; in particular, our convergence guarantees remain valid in high dimension.
>
> ### Practical choice of $D$.
> A natural criterion to set the kernel PCA dimension $D$ is the explained variance ratio: one can set $D$ such that the low-dimensional approximation explains e.g. $0.9$ of the variance of the original kernel matrix. This ratio can be computed cheaply by dividing the trace of the approximate kernel with the trace of the original one. As an example, in our 3D shape examples, $0.9$ of the variance was always explained by $20$ to $50$ dimensions.
>
> ### Sensitivity to the choice of landmarks and unsupervised setting.
> A rough estimation of landmark positions is sufficient to guide the solver towards the right local minimum, as they are only used to estimate a rough initial cross-correlation matrix $\Gamma$. The experiment made on the "kids" shapes (Fig. 4) suggests that the number of local minima is relatively small on structured shapes, and corresponds to the main symmetries of the input: the role of these landmarks is mainly to discriminate between these symmetries.
>
> ### Unsupervised resolution, without landmarks.
> Since the number of local minima is typically small, sampling several random initialization is a natural way to explore the optimization landscape and to identify the true global minima without supervision. To reduce the computational overhead of this random strategy, the landscape exploration can be done on coarse approximations of the input shapes (in the spirit of the multiscale strategy). This way, it is possible to sample dozens of initializations without a significant increase of the computation time, since solving the coarse-scale problem is much cheaper than working at full resolution. Rather than using random initialization, an other possibility would be to base the tested initializations on the set of symmetries of the inputs, providing a more efficient exploration of the optimization landscape. Finally, in many cases, a trivial initialization ($\Gamma = 0$) is sufficient to reach the true global minimum, although this naive strategy may fail in some complex settings.
>
> ### Gradient overhead.
> The main bottleneck of the GW gradient estimation (algorithm 12) is the GW solver itself: all other steps of the algorithm are negligible in comparison. Therefore, the cost of computing the GW gradient (the backward pass) is very small compared to the cost of solving GW (the forward pass).
>
> ### Choice of temperature parameter.
> The choice of the temperature $\varepsilon$ is indeed crucial; this question goes beyond the GW setting and is an important topic in classical optimal transport as well. Informally, entropic regularization makes the optimal plan blurry, with a blur radius of size $\sim \sqrt{\varepsilon}$ (the square root coming from the square exponent in the GW objective -- cf. equation 2). Therefore, in practical applications, one can choose $\varepsilon$ to be equal to the square of the desired spatial precision. By default, one can choose this spatial precision to be equal to the typical distance between two neighboring points in the inputs, so that the final transport plan will be sharp at the resolution level of the input.

---

> > ### Author Rebuttal · Reviewer_mJBs · 2026-04-02
> >
> > I'd like to thank the authors for addressing all my comments. I remain positive and recommend acceptance.

---

### Official Review · Reviewer_6mgZ · 2026-03-13

**Soundness:** 3
**Presentation:** 3
**Significance:** 3
**Originality:** 3
**Overall Recommendation:** 4
**Confidence:** 3

**Summary:**

While Optimal Transport effectively matches different point sets, it is sensitive to rotations. To fix this, the Gromov-Wasserstein (GW) framework was introduced to minimize distortions, however, it brought a new challenge: non-convexity. This paper addresses that non-convexity by (1) identifying a class of distortion penalties that simplify into an alignment problem in a lifted feature space (namely Conditionally of Negative Type functions), and (2) introducing a differentiable GW solver that reduces time complexity from cubic to quadratic while maintaining a linear memory footprint.

**Compliance With Llm Reviewing Policy:**

Affirmed.

**Final Justification:**

The rebuttal addressed our main concerns, so we decide to keep the scores.

**Key Questions For Authors:**

- would have loved to see more comparisons between Entropic-GW, Kernel-GW, LowRank-GW, CNT-GW, Multiscale-GW on more datasets not only in terms of time (table 1) only the one on SMAL is provided in figure 15. Seems CNT-GW is very comparable to kernel-GW from figures in appendix? but the runtime of Kernel-GW is not reported on table 1 can this be reported too?

**Limitations:**

see questions. Mostly more experiments comparing baselines on real world data and time especially against kernel-GW

**Strengths And Weaknesses:**

# Strengths:
- Paper sets out to adress a real challenge the non convexity in the GW framework and tries to propose a fast differentiable algorithm for some classes of funtions
- clearly written and good overview of existing works
# Weaknesses
- would have loved to see more comparisons between Entropic-GW, Kernel-GW, LowRank-GW, CNT-GW, Multiscale-GW on more datasets not only in terms of time (table 1) only the one on SMAL is provided in figure 15. Seems CNT-GW is very comparable to kernel-GW from figures in appendix? but the runtime of Kernel-GW is not reported on table 1 can this be reported too?

---

> ### Author Rebuttal · Authors · 2026-03-30
>
> We thank the reviewer for its suggestions concerning our experimental evaluations. We will update Table 1 according to these remarks, as below. We added several medical shapes to our benchmark from the MedShapeNet dataset [Li] as well (that provides anatomic structures as 3D meshes). We will also add visualizations of the transport plans obtained on these shapes with the different solvers to clearly show that the results are qualitatively equivalent.
>
> | Shapes     | N, M | EGW (Time) | EGW ($GW_\varepsilon$) | KGW (Time) | KGW ($GW_\varepsilon$) | QLrGW (Time) | QLrGW ($GW_\varepsilon$) | CntGW (Time) | CntGW ($GW_\varepsilon$) | MsGW (Time) | MsGW ($GW_\varepsilon$) |
> | ---------- | ---- | ---------- | ---------------------- | ---------- | ---------------------- |:------------ | ------------------------ | ------------ | ------------------------ | ----------- | ----------------------- |
> | Horses     | 4k   | 81s        | 1.4e-2                 | 16s        | 1.3e-2                 | 15s          | 1.4e-2                   | 3s           | 1.3e-2                   | **2s**      | 1.3e-2                  |
> | Hands      | 10k  | 426s       | 1.3e-2                 | 256s       | 1.3e-2                 | 45s          | 1.3e-2                   | 21s          | 1.3e-2                   | **5s**      | 1.3e-2                  |
> | Dogs       | 36k  | Mem.       | Mem.                   | Mem.       | Mem.                   | 205s         | 1.2e-2                   | 145s         | 1.2e-2                   | **26s**     | 1.2e-2                  |
> | Kids       | 60k  | Mem.       | Mem.                   | Mem.       | Mem.                   | 2,025s       | 2.5e-2                   | 279s         | 2.0e-2                   | **83s**     | 2.0e-2                  |
> | Faust      | 177k | Mem.       | Mem.                   | Mem.       | Mem.                   | -            | -                        | -            | -                        | **355s**    | 1.4e-2                  |
> | Femurs     | 25k  | Mem.       | Mem.                   | Mem.       | Mem.                   | 54s          | 6.3e-3                   | 21s          | 6.5e-3                   | **10s**     | 6.6e-3                  |
> | Hips       | 60k  | Mem.       | Mem.                   | Mem.       | Mem.                   | 479s         | 1.0e-2                   | 342s         | 9.4e-3                   | **89s**     | 9.4e-3                  |
> | VesselTree | 100k | Mem.       | Mem.                   | Mem.       | Mem.                   | 2,444s       | 7.7e-3                   | 891s         | 7.5e-3                   | **196s**    | 7.4e-3                  |
>
> We have changed the name of "LowRank-GW" to "Quadratic-LowRank-GW" (QLrGW) as we realized that it could create some confusion: in [Sce], two algorithms are introduced, a quadratic-time solver that uses low-rank decomposition of cost matrices (corresponding to our "LowRank-GW" terminology) and a linear-time acceleration using low-rank approximation of transport plans as well (that we introduced as "Linear-LowRank-GW" in our discussion of the literature,  section A.2.2 -- paragraph "Accelerated Approximations"). However, the terms "LowRank-GW" refers to the latter in the reference python optimal transport library POT: to avoid any possible confusion, we will align to this terminology, replacing "LowRank-GW" by "Quadratic-LowRank-GW" and "Linear-LowRank-GW" by "LowRank-GW". To justify the focus on the quadratic version of the low-rank method, we will also add visualizations of transport plans obtained with the "Linear" LowRank-GW that evidence the typical "quantization" artifacts of this method that makes it unsuitable for precise matching applications.
>
> [Li] Li, Jianning, et al. "Medshapenet–a large-scale dataset of 3d medical shapes for computer vision." Biomedical Engineering/Biomedizinische Technik 70.1 (2025): 71-90.
>
> [Sce] Scetbon, Meyer, Gabriel Peyré, and Marco Cuturi. "Linear-time Gromov Wasserstein distances using low rank couplings and costs." International Conference on Machine Learning. PMLR, 2022.

---

> > ### Author Rebuttal · Reviewer_6mgZ · 2026-04-04
> >
> > - The authors have substantially addressed our concerns, by providing comprehensive new experiments and running times that help clarify our doubts concerning the performances of  CNT-GW and MsGW with kernel-GW, showing the memory as the main bottleneck which is an interesting result. We will be looking forward to the qualitative figures the authors promised to add in the final version.

---

### Decision · Program_Chairs · 2026-04-30

**Decision:**

Accept (regular)

**Comment:**

This paper proposes a scalable method to compute Gromov Wasserstein distance
with conditionally negative general costs and not only squared norms. It relies
on alternating optimization over a transport plan and a linear operator in lifted feature spaces and entropic
regularization.

Reviewers all found the paper interesting but had a few questions about
positioning wrt existing approaches and the numerical experiments including
comparison with other GW solvers. The authors id a very good response and
clarified all points that were raised by the reviewers. All reviewers
acknowledged the response and found that their concerns were fully resolved. The
paper is clearly of good quality and I recommend acceptance but expect the
authors to include in the final version all the clarifications and new
experiments that they did in the response.